# Targeting ATP2B1 impairs PI3K/Akt/FOXO signaling and reduces SARS-COV-2 infection and replication

Pasqualino de Antonellis [ID][1,2,3,11], Veronica Ferrucci [ID][1,2,3,11], Marco Miceli[1], Francesca Bibbo [ID][1,2], Fatemeh Asadzadeh[1,2,4], Francesca Gorini [ID][2], Alessia Mattivi[5], Angelo Boccia[1], Roberta Russo[1,2], Immacolata Andolfo [ID][1,2], Vito Alessandro Lasorsa [ID][1], Sueva Cantalupo[1], Giovanna Fusco [ID][6], Maurizio Viscardi[6], Sergio Brandi [ID][6], Pellegrino Cerino[6], Vittoria Monaco [ID][1,7], Dong-Rac Choi[8,9], Jae-Ho Cheong [ID][8], Achille Iolascon[1,2], Stefano Amente [ID][2], Maria Monti [ID][1,7], Luca L Fava [ID][5], Mario Capasso[1,2], Hong-Yeoul Kim[9] & Massimo Zollo [ID][1,2,3,4,10] ✉

## Abstract

ATP2B1 is a known regulator of calcium ($Ca^{2+}$) cellular export and homeostasis. Diminished levels of intracellular $Ca^{2+}$ content have been suggested to impair SARS-CoV-2 replication. Here, we demonstrate that a nontoxic caloxin-derivative compound (PI-7) reduces intracellular $Ca^{2+}$ levels and impairs SARS-CoV-2 infection. Furthermore, a rare homozygous intronic variant of *ATP2B1* is shown to be associated with the severity of COVID-19. The mechanism of action during SARS-CoV-2 infection involves the PI3K/Akt signaling pathway activation, inactivation of FOXO3 transcription factor function, and subsequent transcriptional inhibition of the membrane and reticulum $Ca^{2+}$ pumps ATP2B1 and ATP2A1, respectively. The pharmacological action of compound PI-7 on sustaining both ATP2B1 and ATP2A1 expression reduces the intracellular cytoplasmic $Ca^{2+}$ pool and thus negatively influences SARS-CoV-2 replication and propagation. As compound PI-7 lacks toxicity in vitro, its prophylactic use as a therapeutic agent against COVID-19 is envisioned here.

**Keywords** SARS-CoV-2; ATP2B1; Transcription; PI3K/Akt/FOXO; $Ca^{2+}$
**Subject Categories** Membranes & Trafficking; Microbiology, Virology & Host Pathogen Interaction; Signal Transduction

## Introduction

The coronavirus disease 2019 (COVID-19) pandemic has been one of the worst crises of our times, which prompted the urgent need to uncover the mechanisms that have pivotal roles with severe acute respiratory syndrome coronavirus 2 (SARS-CoV-2). SARS-CoV-2 uses multiple approaches to infect its host (Baggen et al, 2021), to evade the host responses that are still poorly understood (Xie and Chen, 2020). SARS-CoV-2 infection shows a wide range of clinical features, which range from asymptomatic to mild and severe, and mainly depends on both host genetic factors and virus–host interactions. Coronaviruses (CoVs) contain positive-sense, single-stranded RNA (~30 kb). Four major categories have been reported, with alphaCoV and betaCoV known to infect humans. Those that can replicate in the lower respiratory tract cause pneumonia, which can be fatal (Tay et al, 2020); they include SARS-CoV, Middle East respiratory syndrome-CoV, and the new SARS-CoV-2. This last CoV belongs to the betaCoV genus (Andersen et al, 2020) and has resulted in pandemic acute respiratory syndrome in humans (i.e., COVID-19 disease). This can progress with pathological alveolar cell syncytia formation to an acute respiratory distress syndrome, generally around 8–9 days after symptom onset (Huang et al, 2020).

Like the other respiratory CoVs, SARS-CoV-2 is transmitted via respiratory droplets, with possible fecal–oral transmission (Huang et al, 2020). When SARS-CoV-2 infects host cells, according to the discontinuous transcription mechanism its full-length positive-sense genomic RNA (gRNA) is used to produce both full-length negative-sense RNA copies (–gRNAs) and subgenomic negative-sense RNAs (–sgRNAs) that act as templates for the synthesis of positive gRNA and sgRNA, respectively. Among those sgRNAs, four encode the structural viral proteins Spike (S), Envelope (E), Membrane (M), and Nucleoprotein (N) (Kim et al, 2020; V'Kovski

[1]CEINGE Biotecnologie Avanzate, Naples 80145, Italy. [2]Dipartimento di Medicina Molecolare e Biotecnologie Mediche (DMMBM), 'Federico II' University of Naples, Naples 80131, Italy. [3]Elysium Cell Bio Ita SRL, Via Gaetano Salvatore 486, 80145 Naples, Italy. [4]European School of Molecular Medicine, SEMM, Naples, Italy. [5]Armenise-Harvard Laboratory of Cell Division, Department of Cellular Computational and Integrative Biology—CIBIO, University of Trento, Trento, Italy. [6]Istituto Zooprofilattico Sperimentale del Mezzogiorno, Naples 80055, Italy. [7]Department of Chemical Sciences, University 'Federico II' University of Naples, Naples 80125, Italy. [8]Department of Surgery, Yonsei University College of Medicine, Seoul, Korea. [9]Elysiumbio Inc., #2007, Samsung Cheil B/D, 309, Teheran-ro, Gangnam-gu, Seoul 06151, Korea. [10]DAI Medicina di Laboratorio e Trasfusionale, 'Federico II' University of Naples, 80131 Naples, Italy. [11]These authors contributed equally: Pasqualino de Antonellis, Veronica Ferrucci. ✉E-mail: massimo.zollo@unina.it

et al, 2021). The most recognized receptor used by SARS-CoV-2 to enter host cells is angiotensin-converting enzyme 2 (ACE2). ACE2 together with the cellular serine protease TMPRSS2 are mainly expressed in lung and intestine, and to a lesser extent in kidney, heart, adipose, brain, and reproductive tissues (Lukassen et al, 2020; Walls et al, 2020; Wrapp et al, 2020). Binding to the host ACE2 receptor is mediated by the viral S protein, which consists of two noncovalently associated subunits: the S1 subunit that binds ACE2, and the S2 subunit that anchors the S protein to the membrane. The S2 subunit also includes a fusion peptide and other machinery necessary to mediate membrane fusion upon infection of a new cell. Furthermore, ACE2 engagement by the virus exposes an additional site internal to the S2 subunit, termed the S2′ site; following ACE2-mediated endocytosis, the S2′ site is cleaved by the transmembrane protease serine 2 (TMPRSS2) at the cell surface, or by cathepsin L in the endosomal compartment (Jackson et al, 2022). This "priming" process triggers the fusion of the viral envelope with cellular membranes, thereby allowing the release of the viral genome into the host cell (Hoffmann et al, 2020; Jackson et al, 2022). Despite data showing that ACE2 is a high-affinity receptor for SARS-CoV-2 (Lan et al, 2020), several lines of evidence have suggested the existence of other factors involved in the priming process.

Calcium ($Ca^{2+}$) signals have long been known to have an essential role during the viral cycle (i.e., virion structure formation, virus entry, viral gene expression, posttranslational processing of viral proteins, and virion maturation and release (Zhou et al, 2009). The role of $Ca^{2+}$ in virus–host cell interactions has been shown for various types of enveloped viruses (e.g., Rubella virus (Dube et al, 2014), Ebola virus (Nathan et al, 2020), including CoVs (Berlansky et al, 2022). In this regard, the depletion of extra- and/or intracellular $Ca^{2+}$ pools were shown to significantly reduce the infectivity of SARS-CoV, thus suggesting that both the plasma membrane and endosomal cell entry pathways (Lai et al, 2017) are regulated by $Ca^{2+}$. Of importance, recent studies have also provided evidence that the use of $Ca^{2+}$ channel blockers (e.g., amlodipine, nifedipine) can reduce mortality from COVID-19 (Crespi and Alcock, 2021), thus further underlining the importance of $Ca^{2+}$ in SARS-CoV-2 infection and replication. Intracellular and organellar $Ca^{2+}$ concentrations are tightly controlled via various pumps, including the calcium pumps ($Ca^{2+}$ ATPases). Among the plasma membrane $Ca^{2+}$ pumps, the plasma membrane $Ca^{2+}$ ATPases (PMCAs) are ATP-driven $Ca^{2+}$ pumps that are ubiquitously expressed in the plasma membrane of all eukaryotic cells. The PMCA proteins are encoded by four genes (ATP2B1-4) with numerous splice variants that modulate their tissue distribution, cellular localization, and functional diversity (Krebs, 2015). Among the ATP2Bs genes, the homozygous deletion of the ATP2B1 gene in mice was shown to give rise to a lethal phenotype, thus suggesting the presence of ATP2B1 essential for life with a role as the housekeeping isoform (Okunade et al, 2004) required for the maintenance of intracellular $Ca^{2+}$. Indeed, ATP2B1 is critical for the maintenance of cytosolic $Ca^{2+}$ concentrations below 300 nM (i.e., at ~100 nM), due to its high affinity for $Ca^{2+}$ (Kd, ~0.2 M), and it represents the major $Ca^{2+}$ efflux pathway in nonexcitable cells (Muallem et al, 1988), with an important role in the regulation of the frequency of $Ca^{2+}$ oscillations (Caride et al, 2001). ATP2B1 activity is regulated by calmodulin (CaM) the $Ca^{2+}$ signaling

protein, which stimulates ATP2B1 activity through its binding to an autoinhibitory domain (Bruce, 2018).

Here, using gene expression analysis, we demonstrate unbalanced $Ca^{2+}$ signaling pathways during SARS-CoV-2 (variants of concern [VOCs]: Delta, Omicron 2 and 5 EG5.1) infection in vitro using both human primary human epithelial cells from nasal brushing and HEK293T cells overexpressing ACE2 on the plasma membrane (HEK293T-ACE2). This occurs mostly due to the deregulation of the ATP2B1 and ATP2A1 (i.e., SERCA1) proteins on the plasma membrane and the endoplasmic reticulum, respectively, thus also clarifying the role of $Ca^{2+}$ in SARS-CoV-2 replication. In this regard, the downregulation of ATP2B1 expression promotes SARS-CoV-2 replication by augmenting the intracellular $Ca^{2+}$ levels, as shown here by increased gene and viral Nucleoprotein (N) levels in infected cells. Upon SARS-CoV-2 infection and replication a modulation of ATP2B1 and ATP2A1 expression is observed which involves PI3K/Akt signaling and inhibition of FOXO3 transcriptional activity. Furthermore, through the search of genetic variants associated with severe disease status we identify a rare (0.038187 global frequency) intronic homozygous polymorphism ("rs111337717"—chr12:89643729, T > C) from the ATP2B1 locus that is positively associated with severity of COVID-19. Of importance, using artificial intelligence screening, we identify a new nontoxic caloxin-derivative compound (PI-7) that: (i) promotes ATP2B1 activity as measured by the reduction of intracellular cellular $Ca^{2+}$ levels, (ii) enhances ATP2B1 and ATP2A1 mRNA and protein levels via FOXO3 transcriptional regulation, (iii) impairs SARS- CoV-2 (VOCs: Delta and Omicron 2 and 5) infection and propagation by negatively affecting the generation of syncytia, and (iv) prevents the release of inflammatory cytokines that are targets of the NF-κB signaling pathway. Together, these data identify a genetic risk factor for severe COVID-19 predisposition, and also report on the potential use of a new nontoxic molecule in the fight against SARS-CoV-2 infection.

## Results

### SARS-CoV-2 infection affects intracellular $Ca^{2+}$ signaling via downregulation of ATP2B1

$Ca^{2+}$ homeostasis has been reported to have an important role during SARS-CoV-2 viral infection (Chen et al, 2019; Zhou et al, 2009). Here, to study the physiological response of ATP2B1 to $Ca^{2+}$ oscillations in the presence of SARS-CoV-2, we generated HEK293T cells that stably express its main receptor, ACE2 (i.e., HEK293T-ACE2 cells (Ferrucci et al, 2022). As a control, we used immunoblotting analyses to verify that overexpression of ACE2 does not alter the subcellular localization of the ATP2B1 protein (Fig. 1A). The presence of ATP2B1 protein only in the membranes enriched fraction obtained from HEK293T-ACE2 cells was confirmed (and in their total protein lysates, as the positive control) (Fig. 1A). Then we examined whether different $Ca^{2+}$ concentrations in culturing media could affect SARS-CoV-2 infection and replication. An equal number of cells (i.e., $0.5 \times 10^6$ of cells) were plated in media containing an increasing amount of $Ca^{2+}$ (i.e., 0 mM, 0.1 mM, and 1 mM) and infected with 0.026 multiplicity of infection (MOI) of SARS-CoV-2 (VOC Delta). QPCR analyses confirmed that the viral E and

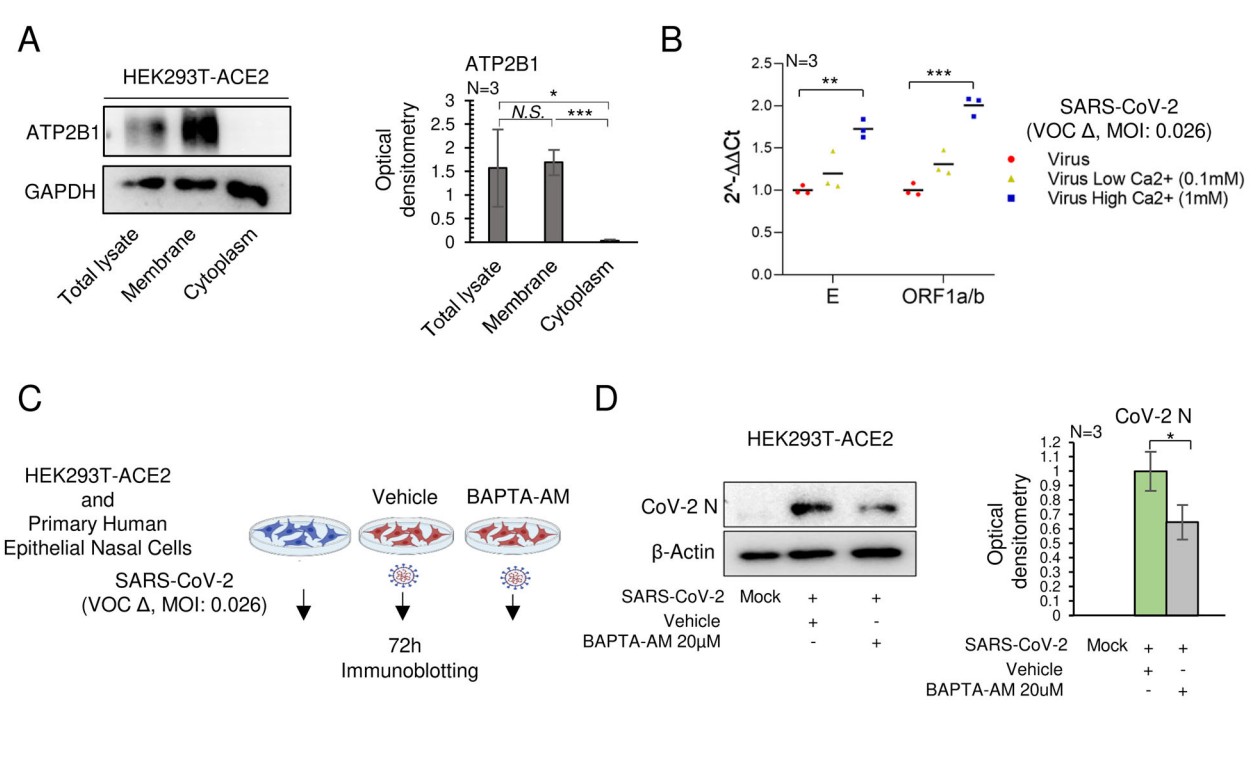

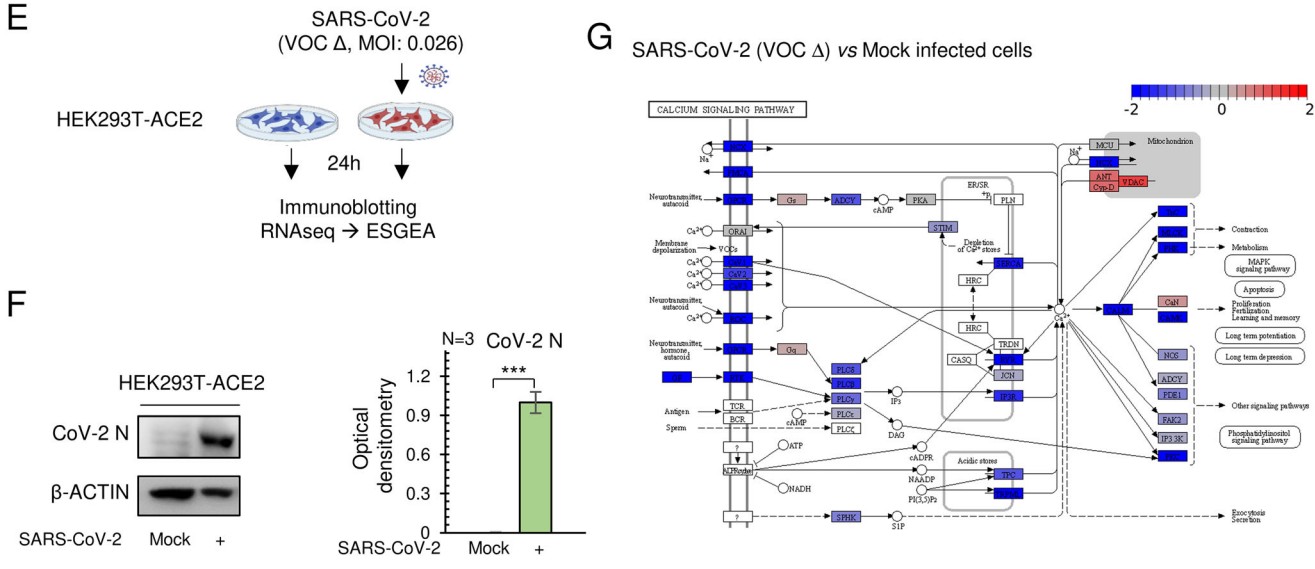

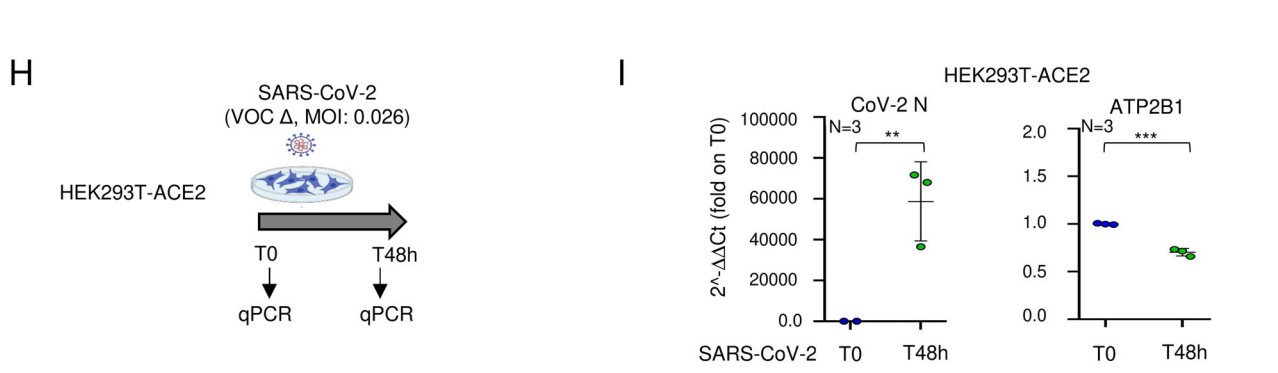

**Figure 1.** Dysregulated expression of $Ca^{2+}$ pumps during SARS-CoV- 2 infection, including ATP2B1.

(A) Left: Representative immunoblotting analysis of cytosolic, membrane, and total protein lysate fractions obtained from HEK293T-ACE2 cells, using antibodies against the ATP2B1 protein. GAPDH is used as the loading control. Right: Densitometric analysis of ATP2B1 band intensities. Data are means ± SD of $N = 3$ biological replicates. Unpaired two-tailed $T$ Student tests, *$P < 0.05$; ***$P < 0.001$, NS not significant. (B) Quantification of mRNA abundance in the presence of increasing amount of $Ca^{2+}$ ($2^{-\Delta\Delta Ct}$) for the viral CoV-2 E and ORF1a/b genes. qPCR analysis of RNA extracted from HEK293T-ACE2 cells infected with SARS-CoV-2 VOC Delta for 24 h. Scattered plots show individual value and mean as indicated by the horizonal black lines of $N = 3$ biological replicates. Unpaired two-tailed $T$ Student tests with Bonferroni correction, **$P < 0.01$; ***$P < 0.001$. (C) Experimental design for the HEK293T-ACE2 cells treated with 20 μM BAPTA-AM and infected with SARS-CoV-2 (VOC Delta at 0.026 MOI) for 72 h. Mock-infected cells are used as control. (D) Left: A representative immunoblotting analysis using antibodies against COV-2-Nucleoprotein (CoV-2 N) on total protein lysates obtained from cells treated with BAPTA-AM at 20 μM concentration. Vehicle-treated cells are used as control. β-Actin is used as the loading control. Right: Densitometric analysis of the CoV-2 N band intensities. Data are represented as means ± SD of $N = 3$ biological replicates. Unpaired two-tailed $T$ Student tests with Bonferroni correction, *$P < 0.05$. (E) Experimental design showing analyses of HEK293T-ACE2 cells infected with SARS-CoV-2 (VOC Delta at 0.026 MOI) for 24 h; mock-treated cells were used as negative control. Total RNA and protein extracted for RNA-seq (Explorative experiment) with ESGEA analyses and immunoblotting. (F) Left: Representative immunoblotting analysis of the CoV-2 N protein in cells treated as in (E). β-Actin is used as the loading control. Right: Densitometric analysis of CoV-2 N. Data are means ± SD of $N = 3$ biological replicates. Unpaired two-tailed $T$ Student tests, ***$P < 0.001$. NS not significant. (G) The "$Ca^{2+}$ signaling by reactome pathway" through RNA-seq analyses from the differentially expressed genes in HEK293T-ACE2 cells treated as in (E). Blue, downregulated (blue) and upregulated (red) genes upon SARS-CoV-2 infection are shown. Red circle indicates PMCA family $Ca^{2+}$ pumps (or ATP2Bs). (H) Experimental design of HEK293T-ACE2 cells infected with SARS-CoV-2 (VOC Delta at 0.026 MOI) for 48 h, RNA extracted at T0 and T48 for qPCR analyses. (I) Quantification of mRNA abundance relative to T0 and T48 ($2^{-\Delta\Delta Ct}$) for the ATP2B1 and CoV-2 N genes in cells treated as in (H). Data are means ± SD of $N = 3$ biological replicates. Unpaired two-tailed $T$ Student tests, **$P < 0.01$, ***$P < 0.001$. Source data are available online for this figure.

ORF1a/b genes were significantly increased in high $Ca^{2+}$ media (Fig. 1B). Previously, it has been reported that a high concentration of gossypol, a known $Ca^{2+}$ regulator, has an additional inhibitory activity in vitro against SARS-CoV-2 RdRp at 50 μM (Wang et al, 2022), whereas at lower concentrations (ranging between 0.2 and 5 μM) is capable of increasing the concentration of intracellular $Ca^{2+}$ by mobilizing intracellular $Ca^{2+}$ pool in a dose-dependent manner and further mobilize the extracellular $Ca^{2+}$ pools (Cheng et al, 2002; Jan et al, 2000). Following these observations, we treated HEK293T-ACE2 cells, in the presence of external $Ca^{2+}$, either with 1 μM or with 5 μM of gossypol. Preincubation with 5 μM of gossypol led to an increase in the efficiency of viral infection/replication as evaluated by increased viral Nucleoprotein (N) and RNA expression in HEK293T-ACE2–infected cells, in comparison with vehicle-treated cells (Fig. EV1A,B). In contrast, $Ca^{2+}$ removals using 20 μM BAPTA-AM a chelator resulted in a reduction of viral N protein levels in both HEK293T-ACE2 (Fig. 1D) and human primary epithelial nasal cells (Fig. EV1C). Taken together, these data uncover a significant role of intracellular $Ca^{2+}$ in SARS-CoV-2 infection and replication in vitro. Then, to better obtain a global picture of the transcriptome landscape in response to the "early phase" of SARS-CoV-2 infection and given the importance of $Ca^{2+}$ homeostasis in SARS-COV-2 infection, we employ gene expression (i.e., RNA-seq) analysis in the HEK293T-ACE2 cellular model upon infection (VOC Delta at 0.026 MOI) for 24 h (see Fig. 1E). Viral infection was confirmed through immunoblotting analyses, which showed the presence of the viral N protein in the HEK293T-ACE2–infected cells, in comparison with mock-infected cells (Fig. 1F). RNA-seq analyses revealed down-regulation of 1742 and upregulation of 34 gene transcripts in SARS-CoV-2 HEK293T-ACE2-infected cells compared to mock-infected cells (i.e., differentially expressed genes [DEGs]; Fig. EV1D). Gene set enrichment analysis on these DEGs showed downregulation of several signaling pathways, where after 24 h of infection, "Calcium signaling" was in the top 20 list (Figs. EV1E and 1G; Appendix Table S1). Within this gene expression picture, we found downregulation of $Ca^{2+}$ pump gene transcripts, including ATP2B (PMCAs) on the cell membrane and SERCAs on the endoplasmic reticulum (Fig. 1F). Moreover, the data obtained in HEK293T-ACE2 cells infected with SARS-CoV-2 for 24 h do not show substantial downregulation of Calcium Release-Activated Channels (CRAC) channels although at

this time, the involvement of CRAC channels cannot be totally excluded (Appendix Table S1). Epithelial lung cells are the most common target of SARS-CoV-2. Acheampong et al (Acheampong et al, 2022) used a single-cell RNA sequencing (RNA-seq) approach to show that a substantial number of the plasma membrane $Ca^{2+}$ ATPases (PMCAs or ATP2B; members of the large family of $Ca^{2+}$ ion pumps) are here found expressed (Fig. EV1F). This analysis revealed that ATP2B1 and ATP2B4 mRNA expression levels are augmented in multiple cell types in the lung parenchyma (i.e., alveolar cells, including macrophages, and in alveolar epithelial cells type I and type II), in contrast to ATP2B2 and ATP2B3 (Fig. EV1F). Here, we further interrogated publicly available datasets obtained from single-nuclei RNA-seq for >116,000 nuclei sequenced from 19 COVID-19 autopsy lungs and seven pre-pandemic controls (Melms et al, 2021) (https://singlecell.broadinstitute.org/single_cell/study/SCP1052/covid-19-lung-autopsy-samples). We found that the ATP2B1-4 (PMCAs) and SERCA pumps (ATP2A1-3) showed distinct fractional and dysfunctional changes across the lungs of patients who died with COVID-19. Of interest, we observed decreased levels of ATP2B1 and ATP2B4 in the lungs of COVID-19 patients, while the levels of ATP2A2 were found to increase (Fig. EV1G). Furthermore, there were undetectable expression levels of ATP2A1, ATP2A3, ATP2B2, and ATP2B3 in the lungs of both COVID-19 patients and controls (Fig. EV1G). These results further confirm those obtained from our in vitro RNA-seq analyses, with downregulation of ATP2B4 and increased levels of ATP2A2 upon SARS-CoV-2 infection in these HEK293T-ACE2 cells (Appendix Table S1). Among these $Ca^{2+}$ pumps, we focused on ATP2B1 because of its pivotal role in the maintenance of $Ca^{2+}$ homeostasis in the cells (Muallem et al, 1988). Using the above public single-nuclei RNA-seq data, we show that ATP2B1 was found reduced in COVID-19 patients compared to controls (i.e., 0.507 vs. 0.513 ATP2B1 expression; Fig. EV1H). However, its expression level was unchanged in the "early phase" of SARS-CoV-2 infection (i.e., after 24 h) in the HEK293T-ACE2 cells (Appendix Table S1). To better dissect the ATP2B1 expression levels during SARS-CoV-2 infection, we used then the HEK293T-ACE2 cell model (Ferrucci et al, 2022). Thus, these cells were infected with viral particles belonging to VOC Delta for 48 h (MOI: 0.026) (Fig. 1H,I). The data obtained from qPCR analyses (see Appendix Table qPCR data) showed upregulated levels

of viral N gene (as the positive control) and downregulated levels of ATP2B1 in the cells infected for 48 h, in comparison to T0 data (fold: 0.7, see Fig. 1I), thus suggesting that the decrease in ATP2B1 levels is not an "early event" during viral infection as also confirmed in lungs from COVID-19 patients (Fig. EV1H). Furthermore, increased amount of $Ca^{2+}$ and reduced amounts of ATP2B1 during SARS-CoV-2 infection are shown to increase viral replication in vitro using the HEK293T-ACE2 cells (Fig. 1B,I). Altogether, these data indicate that SARS-CoV-2 infection affects the $Ca^{2+}$ signaling pathways mainly due to the downregulation of $Ca^{2+}$ pumps on the cell membranes (i.e., ATP2B1).

## Reduced ATP2B1 levels increase SARS-CoV-2 replication in human primary epithelial nasal cells

The link between intracellular $Ca^{2+}$ levels and ATP2B1 was investigated in human primary epithelial cells obtained via nasal brushing from a healthy donor. This was performed to dissect in detail the correlations between ATP2B1 and the intracellular $Ca^{2+}$ pool level during SARS-CoV-2 infection and replication. This cellular model was previously characterized at the molecular level by exome-sequenced technology (deposited on EVA portal— EMBL-EBI; project ID: PRJEB42411; analyses: ERZ1700617) (Ferrucci et al, 2021). To this end, using siRNA approach, we transiently downregulated ATP2B1 (Fig. EV2A–C) in human primary epithelial nasal cells, as verified using immunoblotting analyses performed 48 h after transfection. A downregulation of ATP2B1 expression (Fold 0.4) was observed (Fig. EV2B). In contrast, ATP2B1-overexpressing cells (Fold 1.5) augment of 50% the expression of ATP2B1 (Fig. 2A). We then measured in the same contest the intracellular $Ca^{2+}$ levels using the Fluo3-AM substrate (as described in "Methods"). Overall, these data showed decreased intracellular $Ca^{2+}$ levels in ATP2B1- overexpressing human primary epithelial nasal cells (recorded up to 72 min, with statistical relevance ($P = 0.00365$ Kolmogorov–Smirnov (KS) test; $P = 0.0012$ Fd ANOVA global test between time 24–72 min; ATP2B1-overexpressing cell (black line) vs. E.V. (green line) in the presence of 10 mM $Ca^{2+}$; see Fig. 2B).

In contrast, the downregulation of ATP2B1 in the same cells shows no significant changes in intracellular $Ca^{2+}$ levels (see light-brown and blue lines, Fig. EV2D). At this time, we cannot exclude that the transiently siRNA approaches in primary cell lines were not fully able to impair ATP2B1 expression and for this reason is not potentially affecting the intracellular $Ca^{2+}$ dynamics. Then one other possibility is that $Ca^{2+}$ level can be accordingly adjusted by the activation of other protein pumps here no fully investigated. Thus, other pumps $Ca^{2+}$ dependent can restore the homeostasis of the extra-intracellular $Ca^{2+}$ flux. Indeed, the sequestration of the extracellular $Ca^{2+}$ via EGTA treatment negatively affects the amount of intracellular $Ca^{2+}$ in human primary epithelial nasal cells (see Fig. EV2E). Future studies will clarify the other potential mechanisms responsible for $Ca^{2+}$ homeostasis beyond ATP2B1 downregulation.

The opposite trend between SARS-CoV-2 infection and ATP2B1 levels was then also investigated in this cellular model (i.e., human primary epithelial nasal cells). To this aim, these cells were infected (SARS-CoV-2 viral particles, VOC Delta) for 72 h at 0.026 MOI, with mock-infected cells used as the negative control (Fig. 2C). Immunoblotting analyses showed increased viral N protein levels

upon SARS-CoV-2 infection (72 h; fold: >1.5, see Fig. 2D) and reduced protein levels of ATP2B1 (fold: 0.7; Fig. 2D). Taken together, these data validate the opposite trend observed between viral infection and ATP2B1 expression at the protein level in another cellular model. We next investigated the functional relevance between ATP2B1 expression levels and SARS-CoV-2 infection. To this end, we transiently downregulated ATP2B1 (Fig. EV2A,B), with a control of target specificity using an unrelated ATP2B1 siRNA sequence in HEK293T-ACE2 cells. After 12 h of transfection with siRNA, and infection with SARS-CoV-2 (VOC Delta MOI: 0.026), for an additional 72 h (Fig. 2E). Immunoblotting analyses confirmed increased levels of the viral N protein in cells silenced for ATP2B1 (Figs. 2F and EV2F), thus indicating further that reduced levels of ATP2B1 fosters SARS-CoV-2 replication.

Further, in opposite manner, in an overexpressing status of ATP2B1 upon infection of SARS-CoV-2 (VOC: Omicron, 0.03 MOI) and measuring the expression level of viral structural genes (ORF1a/b and E) and ATP2B1, we saw the reduction of viral gene expression, as featuring the impairment of viral replication (see Fig. 2G).

Overall, the data suggest that SARS-CoV-2 infection requires ATP2B1 downregulation and increased pool of intracellular $Ca^{2+}$ for its propagation in HEK293T and in the airway epithelium human cellular models (Fig. 2H), thus confirming our previous hypothesis.

## An *ATP2B1* intronic polymorphism influences susceptibility for severe COVID-19

As ATP2B1 has been shown to be downregulated in the lungs of COVID-19 patients (Fig. EV1G,H) and here confirmed in our in vitro-infected cellular models (Fig. 1I), we evaluated if potential genetics associations between ATP2B1 locus and the risk of developing severe COVID-19 occurs. To select potential disease causative variants, we downloaded 351 coding variants of ATP2B1 from "The Genome Aggregation Database" (GnomAD v2.1), of these 13 were pathogenic according to their "Functional Analysis through Hidden Markov Models" (FATHMM) prediction scores (Appendix Table S2). However, these variants were extremely rare and were thus excluded from further investigations. The results indicate that loss-of-function mutations of ATP2B1 are rare (observed/expected score = 0.08; data from GnomAD v2.1) and we hypothesized that the ATP2B1 gene related to its activity on regulation of intracellular level of $Ca^{2+}$ pools is essential for the physiological action within the cell (Hegedus et al, 2020). Further we hypothesize that no pathogenetic nucleotide variations within the coding protein region of ATP2B1 can be allowed for the cell to survive. Thus, a second analysis was performed to verify the presence of noncoding variants in the genomic region of the ATP2B1 locus that act as "expression quantitative traits loci" (eQTLs). We used an analytic approach to select candidate functional noncoding variants and single nucleotide polymorphisms (SNPs; Appendix Table S3). We first selected 76 SNPs that were eQTLs for the ATP2B1 gene ($P < 1 \times 10^{-6}$) using the "Genotype-Tissue Expression" (GTex) database (Appendix Table S4). These SNPs were then annotated with prediction functional scores calculated using the "Genome-Wide Annotation of Variants" (GWAVA) tool (Appendix Table S4), and the top six SNPs were

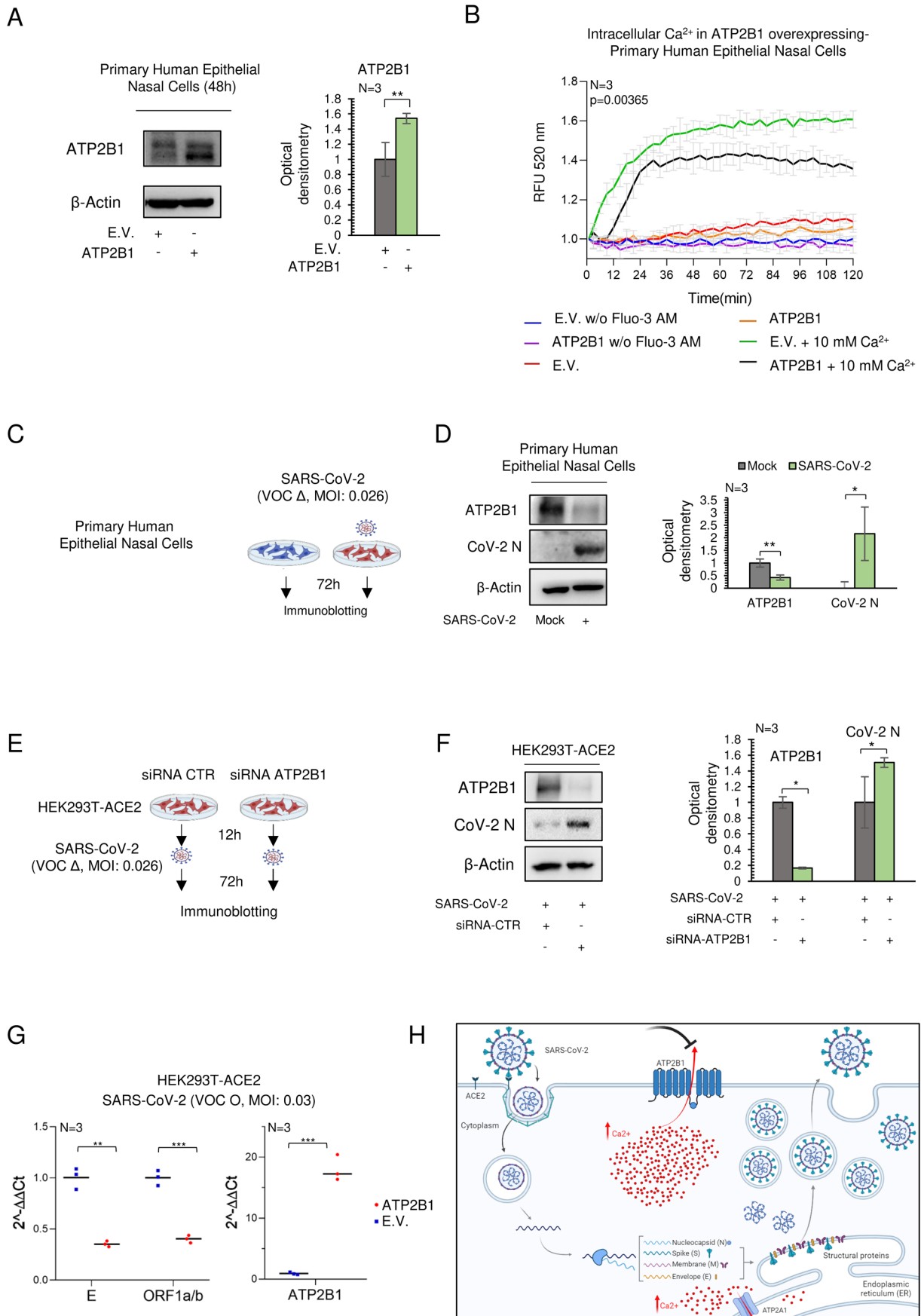

**Figure 2. Reduced ATP2B1 protein levels promote SARS-CoV-2 replication via increasing intracellular Ca²⁺.**

(A) Representative immunoblotting analysis using antibodies against ATP2B1 in human primary epithelial nasal cells transiently overexpressing ATP2B1 human gene (48 h). Cells overexpressing the empty vector (E.V.) were used as negative controls. β-Actin is used as the loading control. Right: Densitometric analysis of ATP2B1. Data are means ± SD of $N = 3$ biological replicates. Unpaired two-tailed $T$ Student tests, **$P < 0.01$. (B) Fluorescence changes relative quantification of intracellular Ca²⁺ by Fluo3-AM for up to 72 min in human primary epithelial nasal cells overexpressing ATP2B1 (48 h after transfection). Cells overexpressing the empty vector [E.V.] used as negative control. Results are expressed as means ± SEM of $N = 3$ biological replicates. Fd ANOVA global $P = 0.0365$, see ATP2B1-overexpressing cell (black line) vs. E.V. (green line) in the presence of 10 mM Ca²⁺. (C) Experimental design showing primary human epithelial nasal cells infected with SARS-CoV-2 (VOC Delta at 0.026 MOI), mock-infected cells used as negative control. After 72 h proteins extracted for immunoblotting. (D) Left: a representative immunoblotting analysis using antibodies against ATP2B1, CoV-2 N in cells as treated in (C). β-Actin is used as the loading control. Right: densitometric analysis of ATP2B1 and CoV-2 N. Data are means ± SD of $N = 3$ biological replicates. Unpaired two-tailed $T$ Student tests, *$P < 0.05$, **$P < 0.01$. (E) Experimental design showing HEK293T-ACE2 cells treated with siRNA against ATP2B1 (or scrambled, CTR) for 12 h, and then infected with SARS-CoV-2 (VOC Delta at 0.026 MOI) for 72 h. (F) A representative immunoblotting analysis using antibodies against ATP2B1 and CoV-2 N from cells treated as in (E). β-Actin is used as the loading control. Right: Densitometric analysis of ATP2B1 and Cov-2 N. Data are means ± SD of $N = 3$ biological replicates. Unpaired two-tailed $T$ Student tests, *$P < 0.05$. (G) Quantification of RNA level measured by qPCR of viral ORF1a/b and E and human ATP2B1 in HEK293-ACE2 transiently transfected with ATP2B1 gene or empty vector E.V. and infected with SARS-COV-2 VOC Omicron (72 h). Data are means ± SD of $N = 3$ biological replicates. Unpaired two-tailed $T$ Student tests, **$P < 0.01$, ***$P < 0.001$. (H) Cartoon representation to illustrate our hypothesis about the role of ATP2B1 during SARS-CoV-2 infection. Downregulation of ATP2B1 increased the intracellular Ca²⁺ levels as observed by the blockage sign (colored black, on top) and resulting on stimulation of SARS-CoV-2 replication with an increased expression of the viral structural genes as represented by enhanced expression of SARS-CoV-2 N protein. Source data are available online for this figure.

selected: rs11105352, rs11105353, rs10777221, rs73437358, rs111337717, and rs2681492. At this time, we excluded rs10777221 (being at the most 5' region in the extragenic region) of the ATP2B1 locus (Fig. EV3A). Linkage disequilibrium (LD) analyses on the remaining SNPs showed that the only variant that is not in linkage disequilibrium was rs111337717 (Figs. 3A and EV3B). This, thus, further indicated that this SNP deserved further analyses as a good candidate for searching for associations to COVID-19 severe and asymptomatic patients. To this end, we tested this SNP (rs111337717; chr12:89643729, T > C) in a cohort of 197 patients affected by severe COVID-19 and 370 asymptomatic cases (D'Alterio et al, 2022). Here, the minor allele "C" of the rs111337717 (NC_000012.12:g.89643729 T > C) with a total allele frequency in global population equal to 0.03992 from gnomAD v4 (https://gnomad.broadinstitute.org/variant/12-89643729-T-C?dataset=gnomad_r4), SNP [(CACATG(T/C)ACATTAT)] was significantly more frequent among severe COVID-19 cases, when compared with asymptomatic individuals ($P = 0.0004$; Table 1), thus suggesting that rs111337717 can be listed among the genetic risk factors for predisposition to severe COVID-19. Of note, alignment sequence analyses of the genomic region flanking rs111337717 [(CACATG(T/C)ACATTAT)] showed high homology identity across different species (Fig. 3B), thus suggesting a potential significant and conserved role during evolution. How this intronic SNP variant influences SARS-CoV-2 infection and propagation will be an issue for future studies. We then ask if this sequence, highly conserved through the evolution, would be a potential DNA binding site for specific transcriptional control factors able to regulate gene expression of the ATP2B1 locus. To this purpose, we query the motif nucleotide elements JASPAR (Sandelin et al, 2004) as available at (https://jaspar.elixir.no/) and we saw the "core" sequence [TG(T/C)ACA] as the human and mouse FOXO3 recognition site (JASPAR CORE transcription factor binding sites databases). Thus, we hypothesized that this site, within the ATP2B1 gene, could be potentially transcriptionally regulated by Forkhead box (FOX) family of transcription factors. Thus, we centered our attention on how the locus is transcriptionally regulated, and we used recent literature data showing "forkhead box O" (FOXO) transcription factors as strong candidates for antiviral responses against SARS-CoV-2. Truly their transcriptional

mechanism of regulation is under an epigenetic control, which in turns regulates those anti-apoptotic and anti-inflammatory pathways, also acting as negative regulators of NF-κB inflammatory signaling (i.e., FOXO3) (Cheema et al, 2021). To this end, we performed in silico analysis of publicly available datasets of single-cell RNA sequencing from 19 COVID-19 autopsy lungs and 7 pre-pandemic controls (Melms et al, 2021); https://singlecell.broadinstitute.org/single_cell/study/SCP1052/covid-19-lung-autopsy-samples). These analyses were performed to define the potential differential expression of FOXOs transcription factors in COVID-19 patients. These data showed higher expression levels in the lung of FOXJ3, FOXK2, FOXN3, FOXO1, FOXO3, and FOXP1. Among these, the expression levels of FOXK2, FOXO3, and FOXP1 were found lower in the lungs of COVID-19 patients, compared to controls (see Fig. 3C), thus showing the same expression levels and trends as observed for ATP2B1 (Fig. EV1G,H). Of note, both HEK293T-ACE2 cells and human epithelial primary nasal cells are not characterized by SNPs in the region of the ATP2B1 locus, as described previously (Fig. EV3A–D). Since FOXO3 shows the same expression trend as ATP2B1, with reduced expression levels in the lung of COVID-19 patients (ATP2B1, Fig. EV1G; FOXO3, Fig. EV3E) we hypothesized that FOXO3 is a candidate transcription factor that would be able to control ATP2B1 expression. We then used genome views of FOXO3 "ChIP-Seq signals" over the ATP2B1 locus using publicly deposited data (Fig. 3D). The normalized FOXO3 signals were visualized with the UCSC genome browser, together with layered CromHMM and H3K4me3 ENCONDE tracks, respectively. The results here indicate that FOXO3 peaks show preferential binding at the promoter and enhancer regions of both the ATP2B1 (Fig. 3D), thus indicating a potential positive transcriptional regulator activity of FOXO3. Of interest, the genome view of FOXO3 "ChIP-Seq signals" over the ATP2A1 locus indicated that FOXO3 peaks also showed binding at the promoter region of ATP2A1 (Fig. EV3F), thus indicating candidate potential activity of FOXO3 as a transcriptional regulator of both the ATP2B1 and ATP2A1 loci. We then investigated whether the predicted functional role also exerts a relevant biological function in our model. To assess the ability of the predicted SNPs in modulating ATP2B1 transcription, genomic regions harboring rs111337717

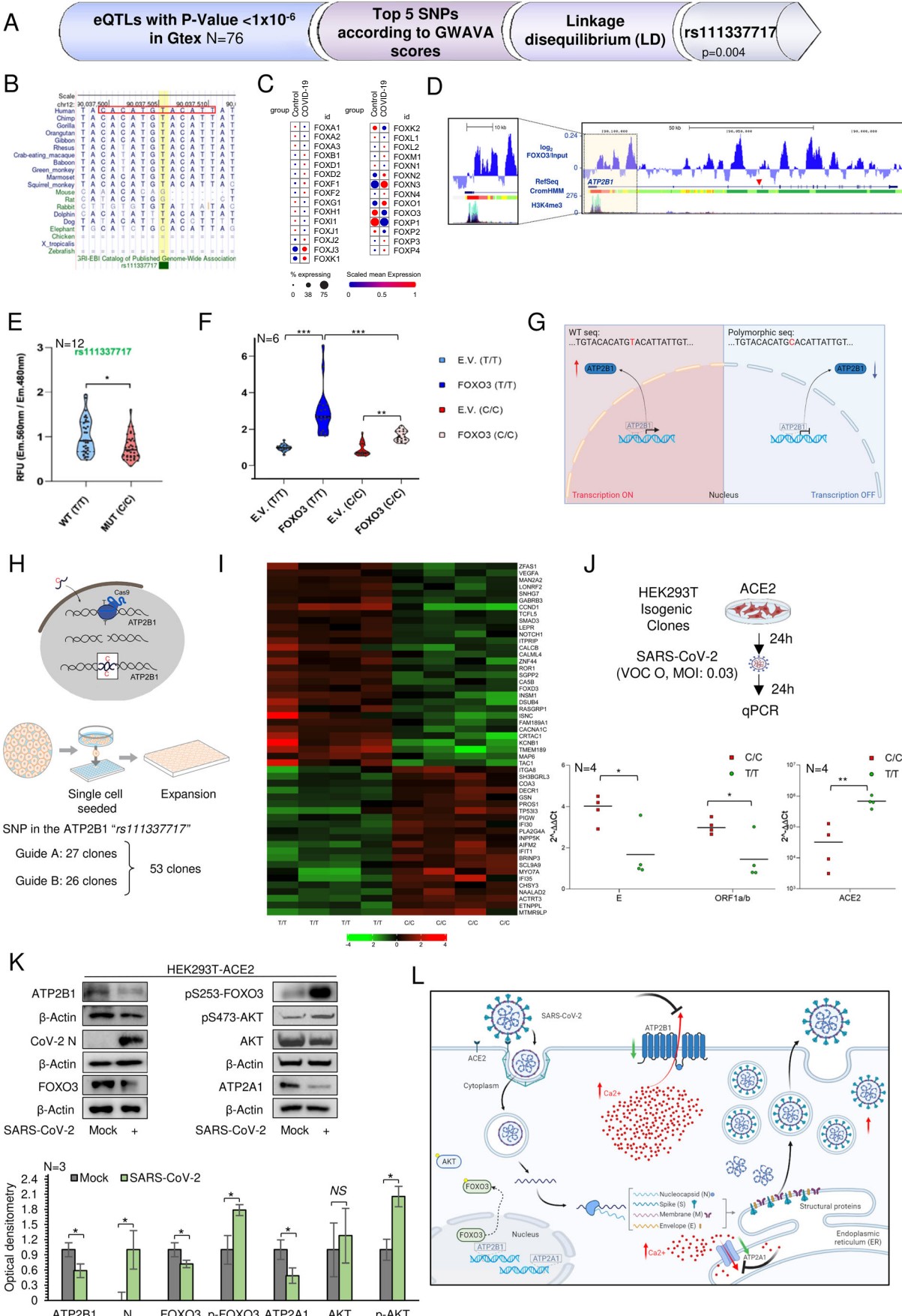

**Figure 3. The homozygous intronic *ATP2B1* variant rs111337717 is responsible for increased SARS-CoV-2 replication in COVID-19 patients via transcriptional regulation of FOXO3.**

(A) In silico analysis pipeline identifying the presence of noncoding variants in the ATP2B1 locus acting as expression quantitative traits (eQTLs) and located in putative elements responsible for transcriptional regulation. EQTLs", GWAVA, and linkage disequilibrium analyses (LD) are used to identify the rs111337717 SNP in the ATP2B1 locus. GTex database (eQTLs): ($P < 1 \times 10^{-6}$) (CIS-eQTL mapping for statistical tools, see (Consortium, 2020); COVID-19 severity vs. asymptomatic patients Fisher test, $P = 0.0004$. (B) Alignment of the sequence genomic region flanking rs111337717 SNP (NC_000012.12: g.89643729 T > C) SNP [(CACATG(T/C)ACATTAT)] shows the conservation through different species sequences through evolution (C) Identification of FOXOs family transcriptional factors (D) CromHMM state segmentation and the H3K4me3 signal (ENCODE), along the ATP2B1 gene in human cells epigenetically analyzed by Genome browser showing accumulation of normalized FOXO3 signal: bright red, promoter; orange and yellow, enhancer; green, transcriptional transition; red arrow, polymorphism- containing region. The expanded view of the highlighted region (left) shows FOXO3 peaks over the ATP2B1 promoter and enhancer regions, as marked by H3K4me3 and CromHMM (red and orange regions), respectively. (E) Luciferase reporter assay for the rs111337717 for both T/T (WT) and C/C (MUT) sequences in ATP2B1 gene as described in Methods section. Data are expressed as means ± SEM of $N = 36$ (n.12 biological replicates with $N = 3$ technical measurements). Unpaired or paired T Student test, *$P < 0.05$. (F) Luciferase reporter assay for the RS111337717 for both "T/T" and "C/C" sequences in HEK293T cells transiently transfected with FOXO3 or Empty Vector (E.V.) for 60 h. Results are expressed as means ± SEM of $N = 18$ ($N = 6$ biological replicates with $N = 3$ technical measurements). One-way ANOVA test in multiple groups comparisons, **$P < 0.01$, ***$P < 0.001$. Statistical details: WT T/T vs. WT T/T FOXO3 $P < 0.0001$; C/C vs. C/C FOXO3 $P = 0.0053$; WT T/T FOXO3 vs. C/C FOXO3 $P < 0.0001$. (G) Right: Cartoon representation to illustrate our hypothesis of the "C/C" sequence in the ATP2B1 intronic region and its transcriptional regulation. (H) Schematic illustration for target T to C genome editing by CRISPR/Cas9 in the rs111337717 region. N. 2 guides (A, B) are used to analyze n.53-edited clones (see "Methods"). (I) Heatmap of top differentially expressed (DE) genes comparing HEK293T (WT T/T) and isogenic generated clones (C/C). Colors represent (green, black, and red) as low, intermediate and high gene expression, respectively. Fold change value +/−2. Statistical Mobin Wald test, $P < 0.005$ Bonferroni corrected of $N = 4$ different isogenic (T/T and C/C) generated clones. $N = 4$ biological replicates. (J) A cartoon showing CRISPR/Cas9 genome editing clones (T/T and C/C) transfected with ACE2 carrying plasmid and infected by SARS-CoV-2. Below: qPCR of viral structural genes (ORF1a/b and E) and ACE2 expression in those ACE2 transiently expressing clones infected by SARS-CoV-2 Omicron 5 (see "Methods"). Scattered plots show individual value and mean as indicated by the horizonal black lines of $N = 4$ biological replicates. Unpaired two-tailed T Student tests, *$P < 0.05$; **$P < 0.004$. Statistical details: CC vs. TT clones: C/C #1; C/C#13, C/C #G; C/C #Q red color vs. T/T WT and T/T #A, T/T #AA, T/T#U green color parental cells. (K) Top: Representative immunoblotting analysis using antibodies against the indicated proteins (ATP2B1, Cov-2 N, FOXO3, pS253-FOXO3, pS473-AKT, ATP2A1) in HEK293T- ACE2 cells infected with SARS-CoV-2 VOC Delta at 0.026 MOI for 72 h. β-Actin is used as the loading control. Mock-infected cells were used as a negative control. Bottom: densitometric analysis of the proteins as above. Data are means ± SD of $N = 3$ biological replicates. Unpaired two-tailed T Student tests, *$P < 0.05$, NS not significant. (L) Cartoon representation to illustrate our hypothesis for downregulation of ATP2B1 during SARS-CoV-2 infection via FOXO3 transcriptional factor. During SARS-CoV-2 infection, while ATP2B1 is downregulated, (see block sign on top) the PI3K/Akt pathway is activated and enhances the phosphorylation of FOXO3, thus excluding its protein entrance in the nucleus (see dashed lines). As a consequence, the expression of the FOXO3 targets, including ATP2B1 and ATP2A1, are also found downregulated, thus increasing the intracellular Ca²⁺ levels and further promoting SARS-CoV-2 replication. Source data are available online for this figure.

**Table 1. Genetic association of rs111337717 and rs116858620 SNP in ATP2B1 gene with severe and asymptomatic cases.**

| rs111337717 | Severe cases | Asymptomatic cases | *P* | OR |
|---|---|---|---|---|
| Genotype | $N = 197$ | $N = 379$ | | |
| TT | 165 (0.84) | 345 (0.93) | | |
| CT | 31 (0.15) | 24 (0.06) | | |
| CC | 1 (0.005) | 1 (0.002) | 0.0005ᵃ | 2.3 |
| Allele | N.394 | N-740 | | |
| T | 361 (0.92) | 714 (0.96) | | |
| C | 33 (0.08) | 26 (0.03) | 0.0004ᵇ | 2.5 |
| rs116858620 | Severe cases | Asymptomatic cases | *P* | OR |
| Genotype | $N = 199$ | $N = 364$ | | |
| TT | 189 (0.95) | 343 (0.94) | | |
| CT | 10 (0.05) | 21 (0.06) | | |
| CC | 0 (0.0) | 0 (0.0) | 0.71 | 0.86 |
| Allele | $N = 398$ | $N = 728$ | | |
| T | 388 (0.97) | 707 (0.97) | | |
| C | 10 (0.03) | 21 (0.03) | 0.71 | 0.86 |

The rs111337717 and rs116858620 SNPs in ATP2B1 gene were tested in a cohort of n.197 patients affected by severe COVID-19 and n.370 asymptomatic cases. The minor allele "C" of rs111337717 (NC_000012.12:g.89643729 T > C) with a frequency of 0.03% in our analyzed cohort; SNP [(CACATG(T/C)ACATTAT)] is significantly more frequent among severe COVID-19 cases when compared with asymptomatic individuals ($P = 0.0004$), thus suggesting rs111337717 SNP could be listed among the genetic risk factor for predisposition to sever COVID-19.
ᵃArmitage trend test: Severe cases vs. Asymptomatic cases.
ᵇFisher's exact test: severe cases vs. Asymptomatic cases.

was cloned upstream of a luciferase reporter gene. Consistent with this hypothesis, the luciferase activities for the allele (T/T) of rs111337717 was higher than the basic pGL4 vector (background) activity. In contrast, luciferase assays for the allele (C/C) SNP rs111337717 was significantly lower confirming our original prediction that the C allele of rs111337717 would alter the binding of FOXO3 in ATP2B1 ($P < 0.05$, Fig. 3E). Moreover, the over-expression of FOXO3 in our luciferase assays increased these differences ($P < 0.001$, Fig. 3F,G). Next, we employed CRISPR/Cas9 technology to develop the disease mutation observed in high-risk patients. Several edited single colonies were picked (Fig. 3H) and expanded followed by DNA sequencing, four clones (C/C #1; C/C #13; C/C #G; C/C #Q) with desired homozygous modifications here identified (Fig. EV3G). RNA-seq analyses revealed that in this model, PI3K/Akt and Ca²⁺ signaling pathways were mainly affected (Appendix Table S5). The transcriptional profiles of $N = 4$ C/C clones were compared then to those of $N = 4$ T/T controls comprising one parental cell line and three unedited post selection HEK293T cell clones, thus taking into account also the effect of the expansion process starting from the single cell ($P < 0.005$, Fig. 3I). We then measured the intracellular Ca²⁺ levels, in the steady state conditions, and found a slight increase of the amount of Ca²⁺ in two clones (Fig. EV3H, $P = 0.0023$ WT vs. C/C#1 and $P = 0.0038$ WT vs. C/C#13; KS test is showing that the two series are significantly different in C/C clones vs. WT T/T cells). The ATP2B1 mRNA level in these edited clones was found moderately decreased as compared to WT cells and unedited clones (Fig. EV3I). This thus suggest that the T > C mutation influences the mRNA levels of ATP2B1 in our genome-edited clones (C/C) containing the SNP variant and identified here. As expected, and confirming our

previous genetic hypotheses linking those to the COVID-19 patient severity, we saw in genome-edited (C/C) clones an increase level of SARS-CoV-2 infection and replication (as measured by ORF1a/b and E structural genes expression) ($P < 0.05$; Fig. 3J). Altogether our results link the severity of COVID-19 in patient genomic data (C/C polymorphism in ATP2B1 locus) to our functional in vitro validation results in our genome-edited (C/C) clones.

To verify further whether FOXO3 and ATP2B1 have the same downregulation trend upon SARS-CoV-2 infection, we infected HEK293T-ACE2 cells with SARS-CoV-2 VOC Delta (MOI 0.026) for 72 h. The immunoblotting data showed downregulated levels of ATP2B1 and FOXO3 in infected cells, with increased levels of phosphorylated (Ser253)-FOXO3 ($P < 0.05$, Fig. 3K). These results indicate an increased levels of the cytosolic inactive phosphorylated-FOXO3 protein, and a reduction in the total FOXO3 and ATP2B1 protein levels following SARS-CoV-2 infection. Of interest, our data also show decreased levels of ATP2A1 upon SARS-CoV-2 infection (Fig. 3K; Appendix Table S1). The expression of ATP2A1 has been previously shown to be reduced upon SARS-CoV-2 infection (Figs. 1I and 2G; Appendix Table S1). The phosphorylation of FOXO3 has been previously shown to be mainly triggered by PI3K/Akt pathway activation, which results in its exclusion from the nucleus and inhibition of the transcriptional activation of its target genes (Brunet et al, 1999; Manning and Cantley, 2007; Stefanetti et al, 2018). Thus, we further investigated the phosphorylation of Akt in the same cells infected with SARS-CoV-2. As expected, we confirmed the literature data showing increased levels of phosphorylated (Ser473)-Akt upon SARS-CoV-2 infection (Khezri et al, 2022) also in this in vitro model (i.e., HEK293T- ACE2 cells, Fig. 3K). Then, in order to validate the functional transcriptional activity of FOXO3, as one of the potential positive transcriptional regulators of ATP2B1 and ATP2A1, we transiently overexpressed a FOXO3-encoding plasmid (containing the coding region of FOXO3; #14937, Addgene) in HEK293T-ACE2 cells. After 48 h from the start of transfection, immunoblotting data showed increased levels of both the FOXO3, ATP2B1 and ATP2A1 proteins (Fig. EV3J), thus suggesting that FOXO3 is a positive regulator of the ATP2B1 and ATP2A1 loci. These data confirm our hypothesis that FOXO3 is a good candidate to transcriptionally activate both membrane (i.e., ATP2B1) and endoplasmic reticulum (i.e., ATP2A1) $Ca^{2+}$ pump expression. Altogether, these data indicate that following SARS-CoV-2 infection, the activation of the PI3K/Akt pathway increases the levels of phosphorylated-FOXO3, thus causing its exclusion from the nucleus, and, as a consequence, the inhibition of their target genes transcription, including ATP2B1 and ATP2A1. This mechanism leads to downregulation of ATP2B1 on plasma membranes and ATP2A1 on endoplasmic reticulum, thus increasing the intracellular $Ca^{2+}$ levels, which enhances SARS-CoV-2 replication, as shown in the model presented in Fig. 3L.

## An ATP2B1 targeting molecule impairs SARS-CoV-2 infection and replication

In the fight against SARS-CoV-2, to identify novel candidate compounds, we used artificial intelligence as a drug design computational tool to model the structure of ATP2B1 (PMCA1)–caloxin 2a1 (as a known ATP2B1 inhibitor; Pande et al, 2011) by docking and energy minimization modeling (Chaudhary et al,

2001) (Fig. EV4A). Thus, a pharmacophore model was built using structures similar to ATP2B1–exodom-2 and caloxin 2a1 (Fig. EV4A). Five pharmacophore features were produced. Among 230 million screened compounds, we selected $30 \times 10^6$ molecules by considering database filtering for solubility and absorption, 7.201 molecules by pharmacophore searches, and 1.028 molecules by docking scoring (Fig. EV4B). Then, from the manually selected the top 22 molecules, two compounds, with the best fit to the docking structure of the aminoacids in the region corresponding to ATP2B1-Caloxin 2a1 (PI-7 and PI-8; Fig. 4A,B) were identified and used for further functional assays. We first assessed the cytotoxicity of these two compounds (PI-7 and PI-8) in terms of cell proliferation and apoptosis in HEK293T-ACE2 cells. The cell proliferation assay (based on the measurement of electrical impedance in real-time; i.e., cells index) was used to determine the half-maximal inhibitory concentrations (i.e., $IC_{50}$) of PI-7 and PI-8. To this aim, we tested escalating doses (from 200 to 1200 μM) of PI-7 and PI-8 on HEK293T-ACE2 cells, and calculated the $IC_{50}$ 48 h after the treatment started ($IC_{50}$ values: PI-7, 580 μM, R2 0.9; PI-8, 336 μM, R2 0.9; Figs. 4C,D and EV4C,D). Of interest, the lower doses of PI-7 and PI-8 (0.1–10 μM) did not alter the cell proliferation rates, as compared to vehicle-treated cells (vehicle: 0.001% dimethylsulfoxide [DMSO]) (Fig. EV4C,D). Similar results were obtained for caspase-3 activation assay as mean of apoptosis. Here, no activation of caspase-3 activity was shown in the HEK293T-ACE2 cells upon treatment with escalating doses of PI-7 and PI-8 (from 1 to 100 μM; Fig. EV4E). These data were further validated by immunoblotting analyses performed on the same treated cells with antibodies against the cleaved fragment of activated caspase-3 (p17 and p19 fragments, see Fig. EV4F). Altogether, these in vitro data exclude the anti-proliferative and the pro-apoptotic actions of both PI-7 and PI-8 compounds. Thus, we evaluated the efficacy of both compounds PI-7 and PI-8 (100 μM) on ATP2B1 pump activity in primary human epithelial nasal cells. The results indicate that PI-7 compound reduced the intracellular $Ca^{2+}$ content as compared to vehicle control (Fig. 4E, $P = 0.0000003063$ by KS test, purple vs. green lines). Conversely, with PI-8 we did not observe changes in concentration of intracellular $Ca^{2+}$ (brown vs. green lines, Fig. 4E, $P = 0.91$ by KS test). Taken together, the data show that compound PI-7 has a better performance in reducing intracellular $Ca^{2+}$. We then investigated in HEK293T cells the effect of decreasing PI-7 concentrations on modulation of $Ca^{2+}$ intracellular dynamics (previously tested at 10 μM of PI-7: Fig. EV4G, $P = 0.0071$ by Fd ANOVA global test between WT vehicle vs. WT 10 μM PI-7; 1 μM of PI-7: Fig. 4F, $P = 0.000787$ by one-way ANOVA and KS test). We notice that the lowest PI-7 concentration, corresponding to 1 μM, substantially decreases intracellular $Ca^{2+}$ in an additional cellular model (i.e., HEK293T cells). Thus, we considered this concentration as representing a lower limit for obtaining sizable physiological effects. This latest condition is responsible for decreased intracellular $Ca^{2+}$ levels required for SARS-CoV-2 infection and replication. For the above reasons, compound PI-7 at 1 μM was used for the following experiments. At this time, we ask how PI-7 will decrease cytoplasmic $Ca^{2+}$. To answer this question, we performed a Cycloheximide (CHX) chase assay to visualize protein-degradation kinetics through CHX treatment over different time points and validation through immunoblotting analyses. We found that ATP2B1 protein half-life is augmented in the treatment with

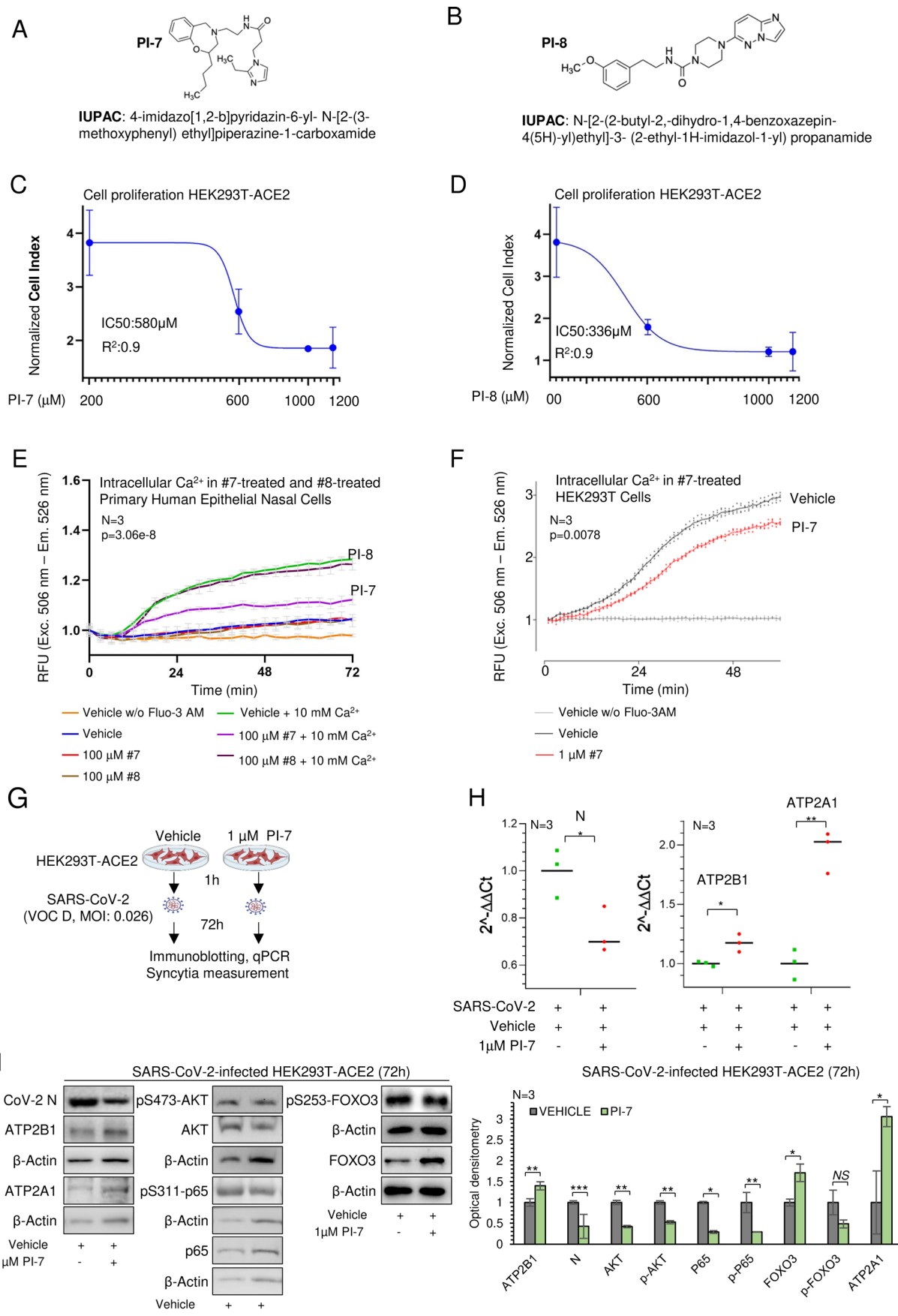

**Figure 4.    ATP2B1 impairment using a caloxin derivative (compound PI-7) impairs SARS-CoV-2 infection by affecting intracellular Ca²⁺ levels.**

(A, B) Molecular structures and IUPAC names of compounds PI-7 (A) and PI-8 (B) selected from the screening. (C, D) Real-time cell proliferation analyses in HEK293T-ACE2 cells (Cell Index; i.e., cell-sensor impedance expressed every 2 min). $IC_{50}$ values are calculated through nonlinear regression analysis, RTCA software vs.1.2.1 (XCELLIGENCE ACEA System application) using the Sigmoidal dose–response (Variable slope) for PI-7 (C) and PI-8 (D). See "Methods" for technical cells handling. Graph generated using Graph Pad Prism 9, with the $IC_{50}$ values given (PI-7: 580 μM, R2 0.9; PI-8: 336 μM, R2 0.9). Data are means ± SD of $N = 3$ biological replicates. (E) Quantification of relative fluorescence changes of Fluo3-AM as a measure of intracellular Ca²⁺ levels. Vehicle-treated cells were used as a negative control. Results are expressed as means ± SEM of $N = 3$ biological replicates. One-way ANOVA and KS test, $P = 3.06E-08$. (F) Quantification of relative fluorescence changes of Fluo3-AM as a measure of intracellular Ca²⁺ levels. Vehicle-treated cells were used as negative control for up to 48 min in HEK293T-ACE2 cells treated with 1 μM of PI-7, vehicle-treated cells were used as negative control. Results are expressed as means ± SEM of $N = 3$ biological replicates. One-way ANOVA and KS test, $P = 0.0007873$. (G) Experimental design showing HEK293T-ACE2 cells treated with PI-7 or Vehicle for 1 h and then infected with SARS-CoV-2 (VOC Delta at 0.026 MOI). After 72 h the cells are lysed or fixed for immunoblotting, qPCR, or immunofluorescence (syncytia measurements), respectively. (H) Quantification of viral Cov-2 N, human ATP2B1 and ATP2A1 gene expression level by qPCR ($2 - \Delta\Delta Ct$) in cells treated as in (G). Scattered plots show the individual values ad mean as indicated by the horizontal black lines of $N = 3$ biological replicates. Unpaired two-tailed T Student tests, *$P < 0.05$, **$P < 0.01$. (I) Representative immunoblotting analysis using antibodies against the proteins (Cov-2 N, ATP2B1, ATP2A1, pS473-AKT, AKT, pS311-p65, p65, pS253-FOXO3, FOXO3) in cells treated as in (G). On the right. Densitometric analysis of the indicated proteins. Data are means ± SD of $N = 3$ biological replicates. Unpaired two-tailed T Student tests, *$P < 0.05$, **$P < 0.01$, ***$P < 0.001$, NS not significant. Source data are available online for this figure.

PI-7 and CHX compared to CHX alone in HEK293T-ACE2 cells (8 vs. 6 h respectively, Fig. EV4H,I). The data presented suggest that PI-7, by enhancing ATP2B1 stability, promotes Ca²⁺ mobilization.

To dissect the intracellular alterations due to treatment with PI-7, we performed a proteomic analysis in the HEK293T-ACE2 cells model upon treatment with 1 μM PI-7 for 24 h (Fig. EV5A). We then generated a network of protein interactions through the "Search Tool for Retrieval of Interacting Genes/Proteins" (STRING) database (Fig. EV5A), with 17 downregulated and 66 upregulated proteins (Appendix Tables S6–8). This network showed that the upregulated proteins take part in the common networks involved in the regulation of metabolic processes and gene expression (Fig. EV5A; Appendix Table S8). Of importance, among those downregulated, we found proteins that are involved in "viral transcription" and "viral processes" (Fig. EV5A in bold; Appendix Table S7). To follow these findings and to weigh the potential antiviral activity of compound PI-7, we treated HEK293T-ACE2 cells with 1 μM PI-7 and then infected with SARS-CoV-2 (VOC Delta) at 0.026 MOI for 72 h (Fig. 4G). QPCR and immunoblotting data show that PI-7 decreased the viral N expression levels (Fig. 4H,I), thus demonstrating inhibition of viral replication (N: qPCR $P < 0.05$, immunoblotting $P < 0.001$). Of note, the data also showed increased ATP2B1 and ATP2A1 RNA and protein levels in PI-7-treated and infected cells (ATP2B1: qPCR $P < 0.05$, Immunoblotting $P < 0.01$; ATP2A1: qPCR $P < 0.01$, immunoblotting $P < 0.05$; Fig. 4H,I).

Taken altogether, the overexpression of ATP2B1 decreases the intracellular Ca²⁺ levels (Fig. 2B) while the treatment with compound PI-7 shows a diminished intracellular Ca²⁺ levels (Fig. 4E,F). In conclusion, the data here are suggesting that compound PI-7 exerts antiviral activity by decreasing the intracellular Ca²⁺ levels, also by promoting its uptake into the endoplasmic reticulum, suggested by the upregulation of ATP2A1.

Then, because of the positive regulation of ATP2B1 and ATP2A1 mediated by FOXO3 (see Figs. 3F and EV3J), we also verified its phosphorylation levels in the same cellular model upon PI-7 treatment and SARS-Cov-2 infection (Fig. 4G). The data show an increased level of total unphosphorylated (transcriptionally active) FOXO3 protein amount ($P < 0.05$, Fig. 4I), and a trend of reduction of inactive phosphorylated (S253)-FOXO3 (not significant (NS), Fig. 4I). Thus, the enhanced transcriptional activity of unphosphorylated FOXO3, due to PI-7 treatment, would also

explain why ATP2B1 and ATP2A1 proteins amount is found restored in the treated cells (Fig. 4H,I). Phosphorylation of FOXO3 and its exclusion from the nucleus, as a consequence of PI3K/Akt pathway activation was already reported by others (Brunet et al, 1999; Manning and Cantley, 2007; Stefanetti et al, 2018). In our previous experiments, FOXO3 was also found inactivated (phosphorylated) during SARS-CoV-2 infection, while Akt pathway is found active ($P < 0.05$, Fig. 3K). We then investigated the phosphorylation status of Akt with SARS-CoV-2 infection, upon PI-7 treatment. Our data show a decrease amount of phosphorylated (Ser473)-AKT upon compound PI-7 treatment during infection ($P < 0.01$, see Fig. 4I). In contrast, the negative regulation of Akt pathway by PI-7 is not observed in the absence of SARS-CoV-2 infection in the same cellular model (Fig. EV5B). This thus confirms that the inhibition of Akt pathway is consequential to PI-7 antiviral activity against SARS-CoV-2.

Of interest, FOXO transcription factors have already been shown to have a role in immune cell maturation and inflammatory cytokines secretion (Cheema et al, 2021). Among the FOXOs, FOXO3 modulates innate immune responses to infections of the airway epithelium through modulation of secretion of several cytokines from immune cells (Xin et al, 2018). This occurs through inhibition of the NF-κB inflammatory pathway, the activation of which is exploited by SARS-CoV-2 (Thompson et al, 2015). Thus, the restoration of the transcriptional activity of FOXO3 by PI-7 (Fig. 4E) might relieve the inflammatory burst following SARS-CoV-2 infection. To test this hypothesis, we investigated NF-κB inflammatory pathway through immunoblotting in the same PI-7-treated and infected cells. Our data show decreased levels of phosphorylated (Ser311)-p65 in PI-7-treated cells, as compared to the vehicle control (p65: $P < 0.05$, p(Ser311)-p65: $P < 0.01$, Fig. 4I). Furthermore, in order to exclude that NF-κB inhibition is only a consequence of a reduced viral infection in these SARS-CoV-2-infected cells, pretreated with compound PI-7, we have also tested the phosphorylation of p65 in non-infected human primary nasal cells with PI-7 for 24 h (Fig. EV5C). Our data show decreased levels of phosphorylated (Ser311)-p65 in PI-7-treated cells in the absence of SARS-CoV-2 (Fig. EV5C), enhancing the efficacy of PI-7 to impair the NF-κB inflammatory pathway. To further confirm the antiviral and anti-inflammatory activity, PI-7 was validated in human primary epithelial nasal cells infected with the latest SARS-CoV-2 variant (VOC: Omicron 2) at 0.04 MOI for 72 h of infection

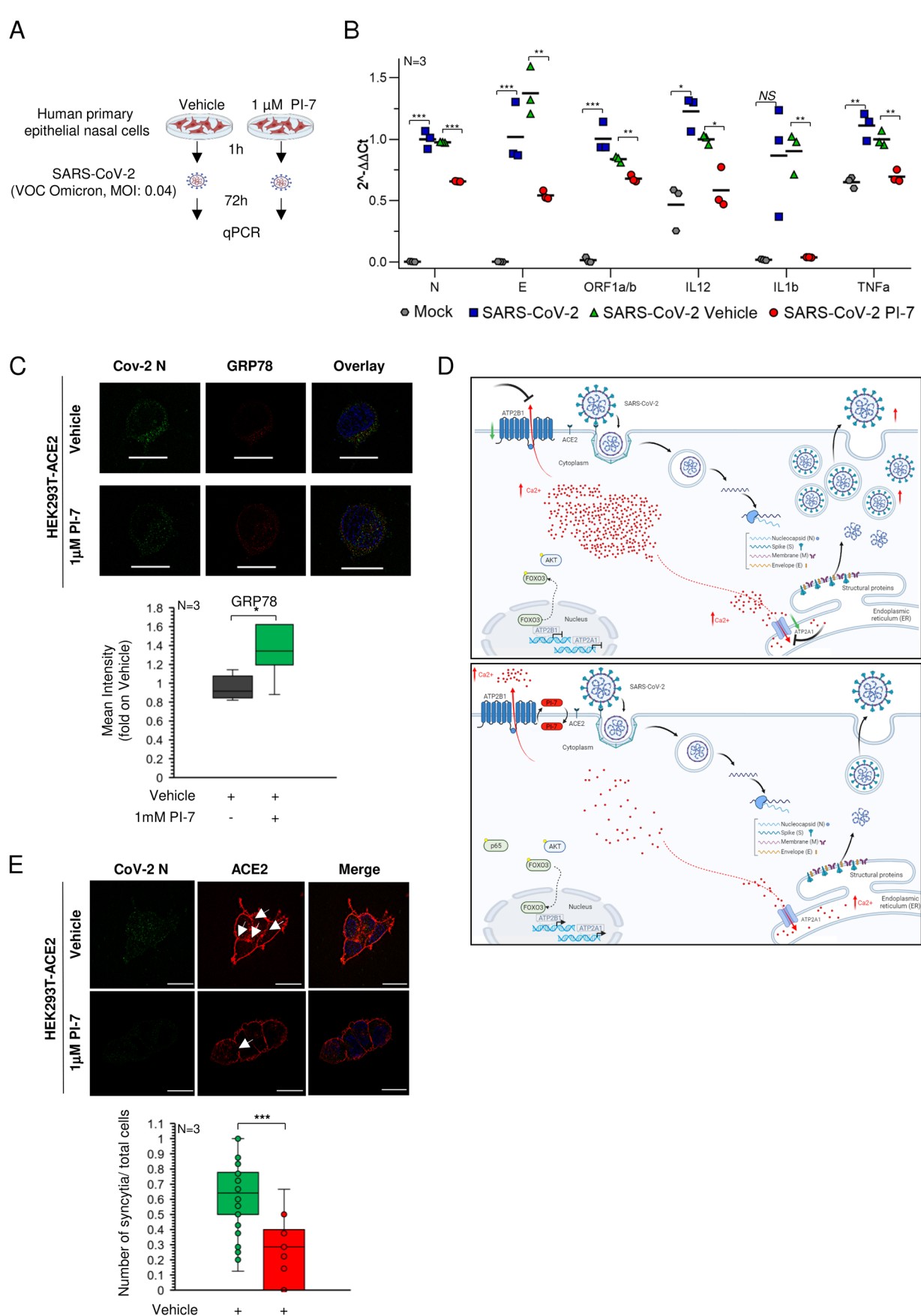

**Figure 5. Compound PI-7 diminishes SARS-CoV-2 replication by affecting syncytia formation and cytokine storm.**

(A) Experimental plan for human primary epithelial cells treated with 1 μM PI-7 or vehicle as control. After 1 h, the cells were infected with SARS-CoV-2 viral particles of VOC Omicron 2 at 0.04 MOI for 72 h. The cells were then used for qPCR analysis. (B) Quantification of viral RNA (N, E, ORF1a/b,) and mRNA for the indicated cytokines ($2^{-\Delta\Delta Ct}$) in cells treated as in (A). The scattered plot shows the individual value and mean as indicated by the horizontal black lines of $N = 3$ biological replicates. Uninfected cells are used as a negative control. Unpaired two-tailed T Student tests and Bonferroni corrected, *$P < 0.05$, *$P < 0.01$, ***$P < 0.001$, NS not significant. (C) On the top: Immunofluorescence staining (IF) with antibodies against viral CoV-2 N (green) and human GRP78 (red) in cells treated ad in Fig. 4G. On the bottom: The graph shows the intensity of fluorescence. SIM$^2$ images are acquired with Zeiss Elyra 7 and processed with Zeiss ZEN software (blue edition). Magnification, ×63. Scale bar, 20 μm. Data are means ± SD of $N = 3$ biological replicates. Data measurements values: vehicle-treated: min=0.820, max=1.143, center=0.915, 1.076 and whiskers=none, bounds of box=0.846–1.076 and percentiles=0.822 ($K = 0.01$) − 1.139 ($K = 0.99$); PI-7 treated: min=0.879 max=1.623, center=1.342 bounds of box=1.193–1.623, whiskers=none, and percentiles= 0.9 ($K = 0.01$) − 2.356($K = 0.99$). Unpaired two-tailed T Student tests, *$P < 0.05$. (D) Cartoon representation to illustrate our hypothesis for the role of ATP2B1 during SARS-CoV-2 infection upon treatment with compound PI-7. (E) Representative IF with antibodies against the CoV-2 viral N protein (green) and human ACE2 (red) in cells treated as in Fig. 4G. White arrows indicate the absence of membranes and syncytia formation. Quantification of the relative proportions of syncytia in >300 cells per condition. The SIM$^2$ image are acquired with Elyra 7 and processed with Zeiss ZEN software (blue edition). Magnification, ×63. Scale bar, 20 μm. Data are means ± SD of $N = 3$ biological replicates. Data measurements values: vehicle-treated min=0.125, max=1, center=0.641, bounds of box=0.5–0.778 and whiskers= 0–1.25, percentiles= 0.064 ($K = 0.01$) − 1.122 ($K = 0.99$); PI-7 treated: min=0.0, max=0.667, center=0.286, bounds of box= 0–0.4, whiskers=none, percentiles= 0 ($K = 0.01$) − 1.653 ($K = 0.99$). Unpaired two-tailed T Student test, ***$P < 0.001$. Source data are available online for this figure.

(Fig. 5A). Our qPCR analyses supported the antiviral activity of PI-7 also against the Omicron 2 SARS-CoV-2 variant, with a confirmed reduction of the viral RNA levels of both N, E, and ORF1a/b in the infected cells (N, E, and ORF1a/b respectively $P < 0.001$, $P < 0.01$, $P < 0.01$, see Fig. 5B). Furthermore, we measured the expression levels of some of the cytokines targeted by NF-κB that take part in the COVID-19 cytokine storm (Hu et al, 2021; Rabaan et al, 2021). The data show that PI-7 can also reduce the SARS-CoV-2 induced cytokines via NF-κB–inhibition in these infected cells (IL-1β, IL-12, TNF-α respectively $P < 0.01$, $P < 0.05$, $P < 0.01$, see Fig. 5B). Altogether, these results show the antiviral activity of compound PI-7 rely on (i) reduced phosphorylated (Ser473)-Akt and phosphorylated (Ser253)-FOXO3, (ii) increased unphosphorylated active FOXO3 protein levels, (iii) enhanced ATP2B1 and ATP2A1 expression and (iv) decreased the intracellular $Ca^{2+}$ pools (Fig. 4I). As a consequence, this leads to impairment of the NF-κB inflammatory pathway (also in part mediated by activation of FOXO3) and inhibition of cytokine expression upon treatment with compound PI-7. When and how this action is taking part in infected SARS-CoV-2 cells will be an issue of future investigations.

Of interest, the treatment with compound PI-7 restored both transcription and translation of ATP2B1 and ATP2A1 protein amounts, mostly as a consequence of FOXO3 transcriptional activity (Fig. 4I). As $Ca^{2+}$ transition had been previously reported to be correlated to the endoplasmic reticulum stress (Deniaud et al, 2008), we validated our qPCR data and ATP2A1 expression levels by measuring of endoplasmic reticulum stress. To this aim, high-resolution immunofluorescence analyses were performed using the lattice SIM2 technology (Elyra 7, ZEISS), which showed increased levels of GRP78, as a marker of endoplasmic reticulum stress, in the SARS-CoV-2-infected HEK293T-ACE2 cells upon treatment with compound PI-7, in comparison with vehicle-treated cells ($P < 0.05$, Fig. 5C). These results suggested that treatment with PI-7 decreases intracellular $Ca^{2+}$ levels also by restoring the expression of ATP2A1 (whose levels were decreased in the presence of SARS-CoV-2, Fig. 3K) and promoting the uptake of $Ca^{2+}$ from cytosol to endoplasmic reticulum. This action is in agreement with the translocation of $Ca^{2+}$ from the cytosol to the endoplasmic reticulum lumen as observed by (Minton, 2013). Taking altogether, compound PI-7 is able to reduce the intracellular $Ca^{2+}$ increased during SARS-CoV-2 infection, through PI3K/Akt pathway

inhibition, and FOXO3 transcriptional activation thus enhancing the expression of ATP2B1 and ATP2A1. A more comprehensive action at the transcriptional level of compound PI-7 will be further detailed in the near future research (see model as presented in Fig. 5D).

## PI-7 reduces syncytia formations in SARS-CoV-2 cellular infections

Many enveloped viruses (including SARS-CoV-2) have been shown to cause fusion of the neighboring cells into multinucleated 'syncytia' (Braga et al, 2021). The relevance of syncytia formation as a measure of virus infectivity is still obscure and needs further investigations, although their formations had clinical impact in during COVID-19 pandemic being a peculiar late-stage histological feature found in the majority of the lungs of these patients as sign of distress function. Here by taking advance of this phenomenon, we developed an assay to be used as an end-point evaluation of drug efficacy in vitro post infection. To further investigate the antiviral activity of compound PI-7, we measured its ability to reduce the syncytia formation. Thus, to determine the relative proportions of syncytia, the same HEK293T-ACE2 cells were treated with PI-7 (1 μM) or vehicle, infected with SARS-CoV-2, and fixed for immunofluorescence analyses using an antibody against the ACE2 protein (Fig. 4G). We found decreased syncytia percentages in the cells upon treatment with PI-7 and BAPTA-AM, respectively ($P = 7.9$ E-5; $P = 1.19$ E-19) (Figs. 5E and EV5D,E) upon 72 h of infection. As a further control, lower expression levels of the "syncytia marker" TMEM16 (Braga et al, 2021) and viral Spike in PI-7-treated (1 μM concentration) HEK293T-ACE2 cells upon SARS-CoV-2 infection (see Fig. EV5F,G). Altogether, these data indicate the antiviral activity of compound PI-7 against SARS-CoV-2 infection by lowering the percentage levels of syncytia formation.

## Discussion

$Ca^{2+}$ as an important second messenger controls essential functions including cellular signaling processes and immune responses (Brini et al, 2013). Intracellular and organellar $Ca^{2+}$ concentrations are tightly controlled via different transmembrane proteins, ATPases,

ion channels, and uniporters. Among these pumps, PMCAs and SERCAs are considered an efficient line of defense against abnormal $Ca^{2+}$ rises. Viral entry into the host cells enhances PI3K/Akt signaling pathways, which in turn diminishes the transcriptional nuclear activity of the transcriptional factor FOXO3 by promoting its phosphorylation (Brunet et al, 1999), thus causing a reduction in the transcriptional activation of their target genes, including ATP2B1 and ATP2A1. This, in turns causes an increases the intracellular $Ca^{2+}$ concentration that enhances SARS-CoV-2 replication and propagation and increases the NF-κB inflammatory pathway (Thompson et al, 2015), thus further promoting the cytokine storm induced by SARS-CoV-2. During viral infection, cellular $Ca^{2+}$ dynamics are highly affected, as dysregulation of the host cell signaling cascades is elicited by these infectious agents (Berlansky et al, 2022). Two mechanisms of regulation of $Ca^{2+}$ can be envisioned upon SARS-Cov-2 infection: the first is related to the extracellular virus–host interaction, and the second is linked to an intracellular mode of action acquired once the virus has entered the cell (Shang et al, 2020). Intracellular $Ca^{2+}$ has been reported to enhance SARS-CoV-2 replication by activating downstream processes, such as modifying the host cellular metabolism and accelerating the inflammation process (Serebrovska et al, 2020). Here, we focused on understanding how intracellular $Ca^{2+}$ modulation can influence SARS-CoV-2 infection and replication. Literature data supports that intracellular $Ca^{2+}$ increase in response to virus infection especially because of Golgi, Endoplasmic Reticulum and mitochondria. Indeed SARS-CoV-2 and other viruses at RNA to replicate and survive into the cells needs higher $Ca^{2+}$ levels (Chen et al, 2019; Poggio et al, 2023; Saurav et al, 2021; Zhou et al, 2013). Data from COVID-19 patients show that upon infection the diminishing level of ATP2B1 (Fig. EV1G,H) enhances the pool of intracellular $Ca^{2+}$ potentially excluding its way-out through the membrane by specific protein pumps. Of note, an opposite trend with upregulation was observed by ATP2A2 (a sarcoplasmic/endoplasmic reticulum $Ca^{2+}$ transporting 2 protein) (see Fig. EV1G), thus suggesting in addition a different transcriptional regulation mechanism of the ATP2A2 gene.

Here, gene expression data obtained using HEK293T-ACE2 cells infected with SARS-CoV-2 showed a modulation of $Ca^{2+}$ signaling pathways, mostly mediated by decreased levels of the PMCA and SERCA $Ca^{2+}$ pumps family proteins (see Fig. 1G; Appendix Table S1). Then we focus on ATP2B1, showing a time-dependent decrease in expression in response to SARS-CoV-2 infection. The results presented here suggest two possible hypotheses of ATP2B1 downregulation. The first implies the action of $Ca^{2+}$ signaling and its dynamics, the second is related to PI3K/Akt/FOXO3 pathway and its negative transcription feedback.

Furthermore, an opposite correlation was seen between viral replication (as shown by the viral N protein levels) and ATP2B1 levels of expression in the two different cellular models (i.e., HEK293T-ACE2 cells in Fig. 1I, and human primary epithelial nasal cells in Fig. 2D). Of interest, the overexpression of ATP2B1 is responsible for decrease intracellular $Ca^{2+}$ levels (Fig. 2B) which is of importance for SARS-CoV-2 viral propagation. Here using a new caloxin-derivative molecule (i.e., compound PI-7), we show decreased intracellular $Ca^{2+}$ levels (Fig. 4E,F) and reduced SARS-CoV-2 infection (Fig. 4H,I). Then we presented data that suggest an enhancement of the level of ATP2B1 protein stability exerted by PI-7 by answering the question on how do PI-7 decrease cytoplasmic

level of $Ca^{2+}$. Thus, this longer time of protein stabilization, measured by CHX studies (Fig. EV4H,I), might be responsible in the reduction of intracellular $Ca^{2+}$ (Figs. 4E,F and EV4G) and counteracting SARS-CoV-2 infections (Fig. 4H). Whether the addition of PI-7 results on the direct or indirect binding to ATP2B1 pump to support the exports of $Ca^{2+}$ into the extracellular environment will be aim of future deep investigations. New structural conformational studies and protein–drug crystallography will be needed to address these findings.

Of importance, prophylactic treatment of HEK293T-ACE2 cells before SARS-CoV-2 infection (VOCs Delta and Omicron) exerts antiviral actions on viral replication (as measured by decreased viral E, Orf1a/b, Spike and N proteins and or RNA expressions). The results showing decreased syncytia formation in the PI-7-treated cells, compared to vehicle controls, provide further support here (see Fig. 5E). This is probably a subordinate effect, caused by the impairment of SARS-CoV-2 spike protein which has known fusogenic properties at the level of the cell plasma membrane since described by (Pal, 2021; Rajah et al, 2022). Furthermore, $Ca^{2+}$ is known to be of importance for syncytia generation: indeed, drugs that inhibit TMEM16 activity (a $Ca^{2+}$-activated ion channel; e.g., niclosamide) blunted $Ca^{2+}$ oscillations in Spike-expressing cells, and as a consequence, inhibited Spike-driven syncytia formation (Braga et al, 2021). In our assays, the negative expression of TMEM16 upon PI-7 treatment further confirmed these results (see Fig. EV5F). Of importance, our data also show restoration of ATP2A1 upon PI-7 treatment (SERCA1, see Fig. 4H,I). Results supporting this hypothesis are also presented in a model which shows that upon SARS-CoV-2 infection, there is downregulation of $Ca^{2+}$ pumps on the cell membrane (ATP2B1) and the endoplasmic reticulum (ATP2A1) thus increasing the intracellular $Ca^{2+}$ levels during viral infection and replication (Fig. 5D). How and when $Ca^{2+}$ is mobilized from endoplasmic reticulum, cytoplasm, and plasma membrane during SARS-CoV-2 replication are questions to address in the near future. Recently M protein of Coronaviruses was found to directly interact with ATP2B1, and through this we might envision regulation of the $Ca^{2+}$ channel and its cellular transport (Gordon et al, 2020).

One of the mechanisms responsible for reduction of expression of ATP2B1 and ATP2A1 during SARS-CoV-2 infection involves the FOXO3 transcription factor and its functional regulation. Our study also demonstrates a positive regulation of the ATP2B1 and ATP2A1 locus by FOXO3. Of interest, during SARS-CoV-2 infection, we observed an increased levels of inactive phosphorylated-(Ser253)-FOXO3 and a substantial reduction in its total protein content under these specific cellular conditions (see Fig. 3K). The phosphorylation of FOXO3 has been previously shown to be triggered by the PI3K/Akt pathway, thus causing its exclusion from the nucleus and inhibiting the transcriptional activation of its target genes (Brunet et al, 1999; Manning and Cantley, 2007; Stefanetti et al, 2018) including the newly identified target here (ATP2B1 and ATP2A1). Of note, over-activation of PI3K/Akt pathway during SARS-CoV-2 infection has also been reported (Khezri et al, 2022), and is here confirmed in our in vitro model. Thus, we hypothesized that following the increased FOXO3 phosphorylation, which inactivates its transcriptional regulation function, augments an additional decrease of ATP2B1 and ATP2A1 mRNA/protein levels during SARS-CoV-2 infection. Of interest, activated Akt is also required for intracellular $Ca^{2+}$ release during

other viral infections (e.g., Herpes simplex virus) (Cheshenko et al, 2013). Thus, taken together, activation of ATP2B1 and ATP2A1 by FOXO3 is a mechanism of escape from virus replication of infected cells, with ATP2B1 being responsible for the pumping of $Ca^{2+}$ from the cytoplasm (and consequently out of the cell) together with the action of ATP2A1, which stores the cytoplasmic $Ca^{2+}$ in the endoplasmic reticulum. These two controlled mechanisms of action can explain the impaired viral replication. Future studies will investigate the link between PI3K/Akt and the intracellular $Ca^{2+}$ release during SARS-CoV-2 infection (see model presented in Fig. 5D).

Of importance, in a cohort of infected symptomatic patients (197 patients affected by severe COVID-19 and 370 asymptomatic cases), we identified a rare homozygous (0.03% allele frequency in our analyzed cohort) intron variant in the ATP2B1 locus (i.e., rs111337717) as a novel genetic factor responsible for severe COVID-19 predisposition (Table 1). The data presented here underline the marker identification to stratify those people who retain the C/C variant in ATP2B1 (rs111337717), as they might be subjected to severe COVID-19 following virus infection and replication. SNP in noncoding regions can have detrimental impacts by modulating expression levels and it remains to be fully explained how these regions that contain the polymorphism act as modulators. According to our model, we postulate the presence of this SNP variant would influence negatively on the regulation of the ATP2B1 locus and its expression. An intriguing question was in those patients carrying the SNV what determines the predisposition to severe symptomatology COVID-19? We employed CRISPR/Cas9 technology to develop isogenic cell lines carrying the disease mutation observed in high-risk patients and we found the same molecular pathways altered during SARS-Cov-2 infection (i.e., PI3K/Akt and $Ca^{2+}$ signaling pathways), suggesting that those are probably the key pathways involved in this process, favoring the viral replication in the host cells. To date, variants in the ATP2A1 genes that are responsible for COVID-19 predisposition have not been reported.

Of importance, the antiviral activity of PI-7 molecule (i.e., intracellular $Ca^{2+}$ levels reduction, Figs. 4E,F and EV4G, viral RNAs and proteins decrease, Fig. 4H,I, and syncytia impairment, Fig. 5E) was tested at nontoxic concentrations (i.e., 1 μM), in terms of cell proliferation and caspase dependent apoptosis (Figs. 4C,D and EV4C–F).

In addition, the data suggest that the downregulation of those inflammatory cytokines, whose expression is driven by NF-κB, might be mediated by the activation of FOXO3 upon PI-7 treatment in SARS-CoV-2-infected cells (Fig. 4I). In this regard, literature data have been previously reported a negative cross-talk between FOXO3 and NF-κB (Thompson et al, 2015). Treatment with compound PI-7 increased FOXO3 protein levels, reduced NF-κB, and diminished the levels of inflammatory cytokines belonging to the cytokine storm in COVID-19 patients (see Fig. 5B). We thus envision the use of compound PI-7 in a prophylactic and therapeutic manner, and with this anti-inflammatory action could prevent the cytokine storm during SARS-CoV-2 infection. At this time, it remains to investigate whether additional NPRL3 inflammasome signaling is further affected (Lee et al, 2012; Nieto-Torres et al, 2015).

It has been proposed that an increase in cytosolic $Ca^{2+}$ causes mitochondrial dysfunction leading to NLRP3 inflammasome activation. Moreover, the Golgi complex has a key role as an intracellular $Ca^{2+}$ store, in synergy with the endoplasmic reticulum. A question remains to be answered related to the potential mobilization of $Ca^{2+}$ by PI-7 in Golgi compartment and its cellular homeostasis. This will be studied in future efforts.

In conclusion, we proposed a model (Fig. 5D) in which during SARS-CoV-2 infection, the PI3K/Akt pathway is activated via Akt phosphorylation, thus enhancing the phosphorylation of FOXO3 to exclude it from the nucleus as already discussed by (Manning and Cantley, 2007, Stefanetti et al, 2018). As a consequence, the expression of the FOXO3 targets, including ATP2B1 and ATP2A1, are reduced, thus increasing the intracellular $Ca^{2+}$ levels and further promoting SARS-CoV-2 replication. Furthermore, SARS-CoV-2 infection also leads to NF-κB activation (also due to reduction of active FOXO3; see also Thompson et al, 2015) and in summary, the expression of the inflammatory cytokines (e.g., IL-1β, TNF-α and IL-12) belonging to the cytokine storm resulted increased. In contrast, treatment with the caloxin derivative (compound PI-7) inhibits the viral propagation of SARS-CoV-2 (VOC: Delta and Omicron) by potentially anchoring ATP2B1 on the membrane thus enhancing its activity, thus reducing intracellular $Ca^{2+}$ levels that are necessary for SARS-CoV-2 replication. Compound PI-7 also increases ATP2B1 and ATP2A1 levels (Fig. 4H,I). These increased levels of ATP2B1 and ATP2A1 mostly occurred because of the enhancement of transcriptionally active FOXO3. The levels of active unphosphorylated FOXO3 are also increased due to the inactivation of PI3K/Akt pathway (due to decreased viral infection upon compound PI-7; Fig. 4I). As a consequence, the levels of nuclear active FOXO3 results increased, thus enhancing the transcriptional activation of ATP2B1 and ATP2A1. Treatment with PI-7 also reduces NF-κB activation (mainly due to enhancement of FOXO3) and accordingly, the expression of the inflammatory cytokines belonging to the cytokine storm resulted decreased IL-1β, TNF-α and IL-12. This approach represents a promising strategy that instead of looking for the development of antiviral therapeutic molecules that target viral proteins, looks directly at the host $Ca^{2+}$ channels, resulting in broad applicability for other infection with need of $Ca^{2+}$ for their in cell replication. However, progress to bring such drugs to the clinic faces several important challenges. There are three main reasons for hesitation. First, $Ca^{2+}$ signaling might affect several biological pathways. Indeed, side effects might be observed in patients treated with these types of drugs. Second, genetic loss-of-function studies in mice and in cells suggest incorrect regulation of the inflammatory responses caused by global $Ca^{2+}$ mis-regulation. Third, and more critically, animal studies with these small molecules to sustain ATP2B1 expression have to confirm the absence of risk of overt adverse effects. At this time, the mechanism of action and compound PI-7 represent additional weapons to impair further waves of infections by COVID-19, as new variants, and co-infection with other viruses and new $Ca^{2+}$-dependent viruses as already reported by other investigators (Chow et al, 2023; Swets et al, 2022).

## Methods

### Cell culture

HEK293T cells (CRL-3216, ATCC) and HEK293T stable clones overexpressing human ACE2 (HEK293T-ACE2 cells) were grown in a humidified 37 °C incubator with 5% $CO_2$. The cells were

cultured under feeder-free conditions using Dulbecco's modified Eagle's medium (DMEM; 41966-029; Gibco) with 10% fetal bovine serum (10270-106; Gibco), 2 mM L-glutamine (25030-024; Gibco), and 1% penicillin/streptomycin (P0781; Sigma-Aldrich), with the medium changed daily. Freshly isolated human nasal epithelial cells were collected by nasal brushing of healthy donors (as previously described; Ferrucci et al, 2021). These primary human epithelial cells (EVA—EMBL-EBI; project ID: PRJEB4241; analyses: ERZ1700617) were cultured in PneumaCult (no. 05009; STEM-CELL Technologies) with 2 mM L-glutamine (25030-024, Gibco), and 1% penicillin/streptomycin (P0781, Sigma-Aldrich). The cells were dissociated with Trypsin-EDTA solution (T4049, Sigma-Aldrich) when the cultures reached ~80% confluency. Cell culture are regularly tested by PCR to assure that cell lines are free from Mycoplasma contamination.

## Generation of HEK293T-ACE2 stable clones

HEK293T-ACE2 stable clones were generated as previously described (Ferrucci et al, 2022). Briefly, HEK293T cells (at ~70% confluency) were transfected with 1 µg DNA plasmid pCEP4-myc-ACE2 (#141185, Addgene) using X-tremeGENE 9 DNA Transfection Reagent (06365779001; Sigma-Aldrich) diluted with serum-free DMEM (41966-029; Gibco), to a concentration of 3 µL reagent/100 µL medium (3:1 ratio [µL]). The transfection complex was added to the cells after 15 min of incubation, in a dropwise manner. At 48 h from transfection, the cell culture medium was changed, and the cell clones were selected using 800 mg/mL hygromycin.

## Transient transfections

### ACE2-overexpressing cells
Briefly, HEK293T clones (at ~70% confluency) were transfected with 1 µg DNA plasmid pCEP4-myc-ACE2 (#141185, Addgene) using X-tremeGENE 9 DNA Transfection Reagent (06365779001; Sigma-Aldrich) diluted with serum-free DMEM (41966-029; Gibco), to a concentration of 3 µL reagent/100 µL medium (3:1 ratio [µL]).

### ATP2B1-overexpressing cells
Primary human nasal epithelial cells ($5 \times 10^5$) and HEK293T-ACE2 cells were seeded into six-well culture plates, and after 24 h they were transiently transfected with the plasmid DNA construct pMM2-hATP2B1b (#47758, Addgene). Transient transfections were performed with X-tremeGENE 360 Transfection Reagent (XTG360-RO, #08724105001, Roche), according to the manufacturer's instructions. Briefly, XTG360-RO DNA Transfection Reagent was diluted with serum-free DMEM (41966-029; Gibco) (10 µL reagent/500 µl medium). Then, 5 µg DNA plasmid was added to 500 µL diluted X-tremeGENE 9 DNA Transfection Reagent. The transfection reagent: DNA complex was incubated for 15 min at room temperature. The transfection complex was then added to the cells in a dropwise manner. Twelve hours after transfection, the cell culture medium was changed. At 48 h after transfection started, the cells were used for $Ca^{2+}$ assays and immunoblotting analyses.

### FOXO3-overexpressing cells
HEK293T-ACE2 cells ($3 \times 10^5$) were seeded in a six-well plate and transfected for 48 h with Flag-FOXO3 (Addgene Flag-FOXO3

#153142) plasmid. Transient transfections were performed with X-tremeGENE™ 360 Transfection Reagent (XTG360-RO, #08724105001, Roche), according to the manufacturer's instructions. Briefly, XTG360-RO DNA Transfection Reagent diluted with serum-free Dulbecco's modified Eagle's medium (41966-029; Gibco) (10 µl reagent/500 µl medium). Then, 2 µg per well of DNA plasmid were added to 250 µl of diluted X-tremeGENE 360 DNA Transfection Reagent, the transfection reagent: DNA complex was incubated for 15 min at room temperature and added to the cells in a dropwise manner. Twelve hours after transfection, the cell culture medium was changed. After 48 h from the transfection started, the cells were used for immunoblotting analyses.

## RNA interference

Primary human nasal epithelial cells. Primary human nasal epithelial cells ($5 \times 10^5$) were seeded into six-well culture plates, and after 12 h they were transiently transfected with the ATP2B1 siRNA (sc-42596, Santa-Cruz). A pool of a three siRNAs (sc-37007, sc-44230, sc-44231, Santa-Cruz) was used as the negative control. RNA interference via siRNAs was performed with Lipofectamine RNA-imax (13778-150, Invitrogen), according to the manufacturer's instructions. Briefly, Lipofectamine RNA-imax was diluted with serum-free DMEM (41966-029; Gibco) (9 µL reagent/150 µL medium). The siRNAs were then diluted in serum-free DMEM (3 µL siRNA/150 µL medium) to obtain the 30 pmol concentration. The transfection reagent:RNA complex was then incubated for 5 min at room temperature. The mixture was then added to the cells in a dropwise manner and incubated for 48 h and 72 h for intracellular $Ca^{2+}$ assays (as described below). HEK293T-ACE2 cells. HEK293T-ACE2 cells ($3 \times 10^5$) were seeded into six-well culture plates, and after 24 h they were transiently transfected with the ATP2B1 siRNA (sc-42596, Santa- Cruz). A pool of three siRNAs (sc-37007, sc-44230, sc-44231, Santa-Cruz) was used as the negative control. Transient transfections were performed with Lipofectamine RNA-imax (13778-150, Invitrogen), according to the manufacturer's instructions. Briefly, Lipofectamine RNA-imax was diluted with serum-free DMEM (41966-029; Gibco) (9 µL reagent/150 µL medium). The siRNAs were then diluted in serum-free DMEM (3 µL siRNA/150 µL medium) to obtain the 30 pmol concentration. The transfection reagent:RNA complex was then incubated for 5 min at room temperature. The transfection complex was then added to the cells in a dropwise manner. After 12 h and 24 h, the cells were infected with SARS-CoV-2 viral particles or lysed for immunoblotting analyses.

## Luciferase assay

Oligonucleotides containing specific elements were synthesized and cloned in a pGL4 vector upstream Luciferase gene. Cells were seeded 12 h before transfection in growth medium without antibiotics. Reporter constructs were transfected using X-treme-GENE™ 360 Transfection Reagent (Roche) according to the manufacturer's instructions. Cells were then seeded in 96-well plates, and assays were performed 48 h post-transfection using the Dual-Luciferase® Reporter (DLR™) Assay System (Promega, Madison, WI) was used to measure the Renilla and Firefly luminescence signals, with the value of Renilla luminescence used for normalization. Luciferase intensity were measured with a multimode plate reader (Enspire, 2300, PerkinElmer).

## In vitro treatment with compound PI-7

HEK293T-ACE and primary human nasal epithelial cells ($2.5 \times 10^5$) were plated in six-well plates and treated with 1 μM compound PI-7, or with 0.001% DMSO as the vehicle control. The proteomic analyses were performed after 24 h of treatment. The viral infections were performed after 1 h of treatment.

## Cell proliferation assays (i.e., cell index)

Real-time cell proliferation analysis for the Cell Index (i.e., the cell-sensor impedance was expressed every two minutes as a unit called "Cell Index"). HEK293T-ACE2 cells ($1.5 \times 10^4$) were plated. After 2 h cells were treated with the indicated concentrations of PI-7 and PI-8 (from 200 to 1200 μM or from 0.1 to 10 μM); with vehicle-treated cells were the negative control. Impedance was measured every 2 min over 48 h. The $IC_{50}$ values were calculated through nonlinear regression analysis performed with ($IC_{50}$ value) the RTCA software vs.1.2.1 (XCELLIGENCE ACEA System application) with the Sigmoidal dose–response (Variable slope).

## Intracellular $Ca^{2+}$ assays in human primary epithelial nasal cells overexpressing- or downregulating- ATP2B1 or treated with EGTA and PI-7-treated HEK293T cells

Nontransfected human nasal primary epithelial cells, HEK293T, and those transfected with the ATP2B1 siRNA (for 48 h) or treated with 1 mM EGTA were plated ($3 \times 10^4$ cells/well) into 96-well plates (3917, Costar) previously coated with PureCol (1:30,000; #5005, Advanced BioMatrix). The cells were washed two times with minimum essential medium (MEM; 21090-022, Gibco) without phenol red and fetal bovine serum. They were then incubated for 30 min at room temperature with a solution of MEM without phenol red and fetal bovine serum containing 2.5 μM Flu-3-AM (F1241, Invitrogen) supplemented with Pluronic F-127 (1:1, v-v; P3000MP, Invitrogen) and 2 mM probenecid inhibitor (57-66-9, Invitrogen). The cells were subsequently washed three times to remove any dye that was nonspecifically associated with the cell surface, with MEM containing probenecid. The cells were then further incubated for 20 min at room temperature with 2 mM probenecid inhibitor dissolved in MEM, to allow hydrolysis of the AM ester bond, and then rinsed with MEM containing probenecid. The cells were then treated with 10 mM $CaCl_2$ (C3306-500G, Sigma) in MEM containing probenecid. The relative fluorescent units (RFUs) were immediately acquired (excitation, 506 nm; emission, 526 nm) using a multimode plate reader (Enspire 2300, PerkinElmer).

## Caspase assay

HEK293T-ACE2 ($3 \times 10^4$) were plated in 96-well plate and treated with escalating doses (1 μM, 10 μM, 100 μM) of the drug PI-7 or the drug PI-8. Vehicle-treated (0.001% DMSO) cells were used as negative controls. Staurosporine-treated cells (10 μM) were used as positive controls of the assays. The assay was performed by using Caspase-3 Activity Assay Kit (#5723, Cell Signaling), by following the manufacture's instruction. Briefly, after 18 h from the treatment started, cell medium was removed and 30 μl of cell lysis buffer (PI-7018, Cell Signaling) were added. The plate was then incubated for 5 min on ice. Later, 25 μl of cell lysate were mixed with 200 μl substrate solution B (#11734S, Cell Signaling) in a black 96-well plate. Fluorescence was detected by using a multimode plate reader (Enspire, 2300, PerkinElmer). The positive control AMC (25 μl, #11735S, Cell Signaling) mixed with 200 μl 1× assay buffer A (#11736S, Cell Signaling) was used as a positive control of fluorescence.

## Protein-degradation assay

HEK293T cells were grown to log phase at 37 °C and cycloheximide was added to a final concentration of 100 μg/ml. Cells were then incubated at 37 °C, and at the indicated time points were collected by centrifugation and lysed. Protein concentration was assessed by the DC Protein Assay (BioRad). Equal amounts of protein were resolved by SDS-PAGE. Immunoblots were performed with a mouse anti-actin primary anti-PMCA ATPase (1:500; 5F10; # MA3-914, Invitrogen) and anti-β-actin (1:10,000; A5441; Sigma) as a loading control.

## SARS-CoV-2 isolation

SARS-CoV-2 was isolated from a nasopharyngeal swab obtained from an Italian patient, as previously described (Ferrucci et al, 2021). Briefly, Vero E6 cells ($8 \times 10^5$) were plated in DMEM (41966-029; Gibco) with 2% fetal bovine serum in a T25 flask, to which the clinical specimen was added. The inoculated cultures were grown in a humidified 37 °C incubator with 5% $CO_2$. When cytopathic effects were observed (7 days after infection), the cell monolayers were scraped with the back of a pipette tip, and the cell culture supernatant containing the viral particles was aliquoted and frozen at −80 °C. Viral lysates were used for total nucleic acid extraction for confirmatory testing and sequencing (GISAID accession numbers: VOC Delta, EPI_ISL_3770696; VOC Omicron 2, EPI_ISL_10743523; VOC Omicron 5 EG5.1).

## SARS-CoV-2 infection

HEK293T-ACE2 and human primary epithelial nasal cells treated with 1 μM compound PI-7 or vehicle (0.001% DMSO) were infected with SARS-CoV-2 viral particles. Mock-infected cells, in which medium of the non-infected cells is replaced with a fresh medium without supplementary FBS, at the same volume as those used in the infected cells experiments, were used as the negative control of infection Cells were then incubated for 1 h at 37 °C and 5% $CO_2$, in parallel with infected cells. Afterward, mock medium was removed, cells were washed once with PBS 1× and fresh medium +2% of inactivated FBS was added to the cell culture, as well as in infected cells. After 24, 48, and 72 h of infection, the cells were lysed for qPCR, immunoblotting, and RNA-seq, or fixed for immunofluorescence analyses. These experiments were performed in a BLS3-authorized laboratory.

## RNA extraction and qPCR assays

RNA samples were extracted using TRIzol RNA Isolation Reagent (15596026, Invitrogen), according to the manufacturer's instructions. Reverse transcription was performed with 5× All-In-One RT MasterMix (g592; ABM), according to the manufacturer's

instructions. The reverse transcription products (cDNA) were amplified by qRT-PCR using an RT-PCR system (QuantStudio™ 5 Real-Time PCR System Applied Biosystems, Foster City, CA, USA). The cDNA preparation was through the cycling method, by incubating the complete reaction mix as follows: cDNA reactions: (37 °C for 10 min and 60 °C for 15 min); Heat-inactivation: 95 °C for 3 min; Hold stage: 4 °C. The targets were detected with the SYBR green approach, using BrightGreen BlasTaq 2× PCR MasterMix (G895; ABM). Human ACTB was used as the house-keeping gene to normalize the quantification cycle (Cq) values of the other genes. These runs were performed on a PCR machine (Quantstudio5, Lifetechnologies) with the following thermal protocol: Hold stage: 50 °C for 2 min; Denaturation Step: 95 °C for 10 min; Denaturation and annealing (×45 cycles): 95 °C for 15 s and 60 °C for 60 s; Melt curve stage: 95 °C for 15 s, 60 °C for 1 min, and 95 °C for 15 s. Relative expression of the target genes was determined using the $2 - \Delta\Delta Cq$ method, as the fold increase compared with the controls. The data are presented as means ± SD of the $2 - \Delta\Delta Cq$ values (normalized to human ACTB) of three replicates. Primes sequences are described in Appendix Table S9.

## SARS-CoV-2 gene (N-E-Orf1a/b) quantification qPCR

RNA extraction using nucleic acid extraction kits (T-1728; ref: 1000021043; MGI tech) with automated procedures on a high-throughput automated sample preparation system (MGISP-960; MGI Tech), following the manufacturer's instructions.

RNA samples (5 ng) were used to perform qPCR with the CE-IVD-approved SARS-CoV-2 Viral3 kit (BioMol laboratories) (Zollo et al, 2021) and the IVD-approved RT-25HT COVID-19 HT Screen (Clonit) following the manufacturer's instructions.

## Immunoblotting

Cells were lysed in 20 mM sodium phosphate, pH 7.4, 150 mM NaCl, 10% (v/v) glycerol, 1% (w/v) sodium deoxycholate, 1% (v/v) Triton X-100, supplemented with protease inhibitors (Roche). The cell lysates were cleared by centrifugation at 16,200×g for 30 min at room temperature, and the supernatants were removed and assayed for protein concentrations with Protein Assay Dye Reagent (BioRad). The cell lysates (20 μg) were resolved on 10% SDS-PAGE gels. The proteins were transferred to PVDF membranes (Millipore). After 1 h in blocking solution with 5% (w/v) dry milk fat in Tris-buffered saline containing 0.02% [v/v] Tween-20, the PVDF membranes were incubated with the primary antibody overnight at 4 °C: anti-SARS-CoV-2 N protein (1:250; 35-579; ProSci Inc.); anti-PMCA ATPase (1:500; 5F10; # MA3-914, Invitrogen); anti-β-actin (1:10,000; A5441; Sigma); anti-p65 (1:1000; sc-372, Santa-Cruz Biotechnology); anti-p-(Ser311) p65–NF-κB (1:500; sc-101748. Santa-Cruz Biotechnology); anti-Caspase-3 (1:500, ab49822, Abcam); anti-phosphorylated-Ser473-AKT (1:250, ab81283, Abcam); anti-AKT1 (1:500, #2967, Cell Signaling); anti-p-(Ser253) FOXO3A (1:500, ab47285, Abcam); anti-FOXO3A (1:1000, ab47409, Abcam), ATP2A1 (1:500, ab2819, Abcam), anti-glyceraldehyde-3-phosphate dehydrogenase (1:10,000; sc-365062. Santa-Cruz Biotechnology), anti-α-tubulin (1:3000; Ab15246, Abcam), anti-Flag (1:10,000 SIGMA, A2220). The membranes were then incubated with the required secondary antibodies for 1 h at room temperature, as secondary mouse or

rabbit horseradish-peroxidase-conjugated antibodies (NC 15 27606, ImmunoReagents, Inc.), diluted in 5% (w/v) milk in TBS-Tween. The protein bands were visualized by chemiluminescence detection (Pierce-Thermo Fisher Scientific Inc., IL, USA). Densitometric analysis was performed with the ImageJ software. The peak areas of the bands were measured on the densitometry plots, and the relative proportions (%) were calculated. Then, the density areas of the peaks were normalized with those of the loading controls, and the ratios for the corresponding controls are presented as fold changes. Immunoblotting was performed in triplicate. The densitometric analyses shown were derived from three independent experiments. Membrane and cytoplasmic protein fractions of the cultured cells were obtained with Mem-PER Plus Membrane Protein Extraction kits (89842, Thermo Scientific).

## High-resolution immunofluorescence

SARS-CoV-2-infected HEK293T-ACE2 cells were fixed in 4% paraformaldehyde in phosphate-buffered saline (PBS) for 30 min, washed three times with PBS, and permeabilized for 15 min with 0.1% Triton X-100 (215680010; Acros Organics) diluted in PBS. The cells were then blocked with 3% bovine serum albumin (A9418; Sigma) in PBS for 1 h at room temperature. The samples were incubated with the appropriate primary antibodies overnight at 4 °C: anti-ACE2 (1:1000; ab15348; Abcam); anti-SARS N protein (1:100; 35-579; ProSci Inc.); or anti-GRP78 BiP (1:1000, ab21685, Abcam). After washing with PBS, the samples were incubated with the secondary antibody at room temperature for 1 h, as anti-mouse Alexa Fluor 488 (1:200; ab150113; Abcam) or anti-rabbit Alexa Fluor 647 (1:200; ab150075; Abcam). DNA was stained with DAPI (1:1000; #62254, Thermo Fisher). The slides were washed and mounted with coverslips with 50% glycerol (G5150; Sigma-Aldrich). Microscopy images were obtained using the Elyra 7 (Zeiss) platform with the optical Lattice SIM2 technology (with the ZEN software, blue edition), using the ×63 oil immersion objective.

## Differential proteomics analysis

Differential proteomic analysis was carried out using a shotgun approach. In detail, three biological replicates of HEK293T-ACE2 cells treated for 24 h with compound PI-7 or with the vehicle control (0.001% DMSO) were lysed with lysis buffer (50 mM ammonium bicarbonate, 5% sodium dodecyl sulfate), and the protein extracts were quantified by BCA assay. The equivalent of 50 μg of each protein extract was reduced, alkylated, and digested onto S-Trap filters according to the Protifi protocol (Protifi, Huntington, NY), as previously reported. Peptide mixtures were dried in a SpeedVac system (Thermo Fisher, Waltham, MA) and an aliquot for each replicate was subjected to a clean-up procedure using a C18 zip-tip system (Merck KGaA, Darmstadt, Germany). Desalted peptide mixtures were resuspended in a solution of 0.2% HCOOH in LC-MS grade water (Waters, Milford, MA, USA) and 1 μL of each was analyzed using an LTQ Orbitrap XL (Thermo Scientific, Waltham, MA) coupled to the nanoACQUITY UPLC system (Waters). Samples were initially concentrated onto a C18 capillary reverse-phase pre-column (20 mm, 180 μm, 5 μm) and then fractionated onto a C18 capillary reverse-phase analytical column (250 mm, 75 μm, 1.8 μm), working at a flow rate of 300 nl/

min. Eluents "B" (0.2% formic acid in 95% acetonitrile) and "A" (0.2% formic acid, 2% acetonitrile, in LC-MS/MS grade water, Merck) were used with a linear two-step gradient. The first step went from 5% B to 35% B in 150 min, and the second step from 35% B to 50% B in 10 min. MS/MS analyses were performed using Data-Dependent Acquisition (DDA) mode, after one full scan (mass range from 400 to 1800 $m/z$), with the 10 most abundant ions selected for the MS/MS scan events, and applying a dynamic exclusion window of 40 s. All of the samples were run in technical duplicates. The raw data were analyzed with MaxQuant 1.5.2 integrated with the Andromeda search engine. For the MaxQuant, the following parameters were used: a minimum of four peptides including at least two unique for protein identification; a false discovery rate (FDR) of 0.01 was used. Protein quantification was performed according to the label-free quantification intensities, using at least four unmodified peptides razor + unique.

## Protein–protein interaction networks

To investigate the interactions between the protein products of the top differentially expressed proteins in the HEK293T-ACE2 cells following treatment with compound PI-7, a protein interaction network was generated using the Search Tool for the Retrieval of Interacting Genes/Proteins (STRING) database (https://string-db.org). The nodes consisted of genes, and the edges were derived from experimentally validated protein–protein interactions.

## RNA sequencing (RNA-seq)

### RNA isolation and library construction and sequencing

The total RNA was isolated from the HEK293T-ACE2 cells and different edited clones using TRIzol RNA Isolation Reagent (#15596018; Ambion, Thermo Fisher Scientific). It was then quantified in a NanoDrop One/OneC Microvolume UV–Vis spectrophotometer (Thermo Scientific), checked for purity and integrity, and submitted to Macrogen Europe B.V. for sequencing. Libraries were prepared using TruSeq Stranded mRNA Library Prep kits according to the protocols recommended by the manufacturer (i.e., TruSeq Stranded mRNA Reference Guide # 1000000040498 v00). Trimmed reads were mapped to the reference genome with HISAT2 (https://ccb.jhu.edu/software/hisat2/index.shtml), a splice-aware aligner. After the read mapping, Stringtie (https://ccb.jhu.edu/software/stringtie/) was used for transcript assembly. The expression profile was calculated for each sample and transcript/gene as read counts, FPKM (fragment per kilobase of transcript per million mapped reads), and TPM (transcripts per kilobase million).

### Analysis of differentially expressed genes

Differentially expressed genes analysis was performed on a comparison pair (SARS-CoV-2-infected vs. mock-infected cells and T/T vs. C/C clones). The read count value of the known genes obtained through the -e option of the StringTie was used as the original raw data. During data preprocessing, low-quality transcripts were filtered out. Afterward, Trimmed Mean of M-values (TMM) normalization was performed. Statistical analysis was performed using Fold Change, with exactTest using edgeR per comparison pair (SARS-CoV-2-infected vs. mock-infected cells) and using DESq2 per comparison pairs (T/T vs. C/C clones). For

significant lists, hierarchical clustering analysis (Euclidean method, complete linkage) was performed to group the similar samples and genes. These results are depicted graphically using a heatmap and dendrogram. For the enrichment test, which is based on Gene Ontology (http://geneontology.org/), DB was carried out with a significant gene list using g: Profiler tool (https://biit.cs.ut.ee/gprofiler/). Pathway enrichment analysis was performed using the KEGG database (http://www.genome.jp/kegg/).

## ChIP-seq analysis

ChIP-Seq of FOXO3 and the input control were downloaded from E-MTAB-2701.XC1. Reads were quality checked and filtered with Trimmomatic (Bolger et al, 2014). Alignments to the reference genome (hg19) were performed with BWA aln (Li and Durbin, 2010), using the default parameters. SAMtools rmdup was used to remove potential PCR duplicates. SAMtools sort and index were used to sort and index the bam files. Uniquely mapped reads of the FOXO3 signal were normalized over genomic input log2[FOXO3/Input]) using the bamCompare tool from Deeptools suite (Diaz et al, 2012) with the exactScaling method as the scaling factor. The normalized FOXO3 signal was visualized with UCSC genome browser.

## Phenotype definition

The COVID-19 asymptomatic cohort were selected as previously described [PMID:33815819]. The cohort included individuals from Campania (Italy) screened in May 2020 for SARS-CoV-2 who were positive for SARS-CoV-2 antibodies but without any COVID-19 symptoms in the three previous months, such as hospitalization requirement, fever, cough or at least two symptoms among sore throat, headache, diarrhea, vomiting, asthenia, muscle pain, joint pain, smell or taste loss, and shortness of breath (Lavezzo et al, 2020). The COVID-19 hospitalized cohort was selected as previously described (Russo et al, 2021; Andolfo et al, 2021).

## In silico analysis

The list of 351 coding variants of ATP2B1 was obtained from gnomAD (https://gnomad.broadinstitute.org/). FATHMM scores were used to assess the pathogenicity of the coding variants (Shihab et al, 2013). The GTEx database (Consortium, 2020) was used to select the eQTLs for ATP2B1, setting the level of significance at $P < 1 \times 10^6$ (see par.4,2 supplemental section https://doi.org/10.1126/science.aaz1776). The GWAVA tool was used to assign a functional score to each SNP (Ritchie et al, 2014). RegulomeDB was used to evaluate the effects of SNPs on altering TFBS (Boyle et al, 2012).

## DNA extraction, SNP genotyping

Genomic DNA of cases was extracted from peripheral blood using Maxwell RSC Blood DNA kits (Promega, Madison, WI, USA), and DNA concentrations and purities were evaluated using a NanoDrop 8000 spectrophotometer. The DNA samples were genotyped by the TaqMan SNP Genotyping Assay for the SNPs rs111337717 and rs116858620 (Applied Biosystems by Thermo Fisher Scientific, Waltham, MA, USA).

## In silico analysis of single-cell RNA-sequencing data

Publicly available single-cell RNA-sequencing data for adult human lung from COVID-19 patients were used, from https://singlecell.broadinstitute.org/single_cell/study/SCP1219/columbia-university-nyp-covid-19-lung-atlas (Melms et al, 2021). The proportions of ATP2B1-, ATP2B2-, ATP2B3-, and ATP2B4-positive cells in different lung cell types were calculated. In a similar manner, the expression of ATP2B1, ATP2B2, ATP2B3, ATP2B4, ATP2A1, ATP2A2, ATP2A3, and FOXOs in these lungs from COVID-19 patients and controls were investigated.

## Transfection and electroporation of cell lines

Transfection and electroporation of cell lines HEK293 cells were transfected in 24-well multi-wells with 500–750 ng of Cas- or ABE-encoding plasmids, 250 ng of the desired pUC19-sgRNA plasmid and, in HDR experiments, 200–500 ng of ssODN using TransIT-LT1 (Mirus Bio), according to manufacturer's instructions. The following guides were used:

Guide A: TGTACACATGTACATTATTG
Guide B: GTACACATGTACATTATTGT
Ultramer:CAGATATATCCCAATTAGTTAATGCAAAGTGTGTACACATGCACATTATTGTGTGAATGTAACAGTACAGACAGCAGTGACAAAT.

Clones were single-cell seeded and expanded, and Sanger sequenced.

## In silico off-target analysis

Off-targets for gRNA + 4 were analyzed by Cas-OFFinder online algorithm, by selecting: SpCas9 from Streptococcus pyogenes: 5'-NGG-3', mismatch number ≤ 4, DNA bulge size = 0, RNA bulge size = 0 and as a target genome the Homo sapiens (GRCh38/hg38)—Human.

## Statistical analyses

Statistical testing on each experiment is describing the name of the statistical test used to generate error bars and $P$ values, and the number (n) of independent experiments underlying each data point with biological replicates. Figure legends contain a basic description of n, P and the test applied. A pipeline statistical analyses is applied to time series experiments following (a) ANOVA Single factor (SPSS vs.29.0.2.0 -20), using KOLMOGOROV–SMIRNOV (KS) shortcut function, two-sample test and a "non parametric" KS shortcut function, were applied to two-sample test. (b) Fd ANOVA using a global approach and pair wise analyses for selected groups "Functional data analysis of variance" (Górecki and Smaga, 2019). Then we apply the "unpaired two-tailed Student's $t$ tests" adjusted with Bonferroni method for multiple comparison. All of the data are given as means ± SD or SEM. In each figure, statistical significance is represented as follows: $*P < 0.05$, $**P < 0.01$, and $***P < 0.001$. The $IC_{50}$ values for PI-7 and PI-8 were calculated by nonlinear regression analysis, performed with Graph Pad Prism 9 (using [inhibitor] versus response [three parameters]) and applying the statistical formula related to the Sigmoidal dose–response (Variable slope). Allele and genotype frequencies were compared using chi-square or Armitage tests. Proteomic data were statistically analyzed using the Perseus software (Perseus version 1.6.15.0) with Student's $t$ tests were applied with a false discovery rate cutoff of 0.05 and with a log2FC cutoff of 0.5.

## Biosafety

Experiments involving SARS-CoV-2 were performed in accordance with the Guidance_on_the_Regulation_of_Select_Agent_and_Toxin_Nucleic-Acids as released by the Federal Select Agent Program (FSAP available at www.selectagents.gov, document February 2020). A BSL3 authorized—Risk Group 3 (RG3)—laboratory at CEINGE was used for the working environment with Regulated Genomes at Lower Containment Policy. All cell culture experiments with viruses have been conducted in BSL-3 laboratories using procedures approved by each institution's health and security department authorized by Regione Campania ASL Napoli 1, Ministero della Sanità and Ceinge Biotecnologie avanzate "Franco Salvatore"/Azienda Ospedaliera Universitaria Federico II, September 1, 2023.

## Ethics

Committee approvals for the COVID-19 samples use in this study were as follows: (i) protocol no. 141/20; date: April 10, 2020, CEINGE TaskForce Covid-19; Azienda Ospedaliera Universitaria Federico II, Direzione Sanitaria, protocol no. 000576 of April 10, 2020; (ii) protocol no. 157/20; date: April 22, 2020, GENECOVID, with the experimental procedures for the use of SARS-CoV-2 in a biosafety level 3 (BSL3) laboratory were authorized by Ministero della Sanità and Dipartimento Di Medicina Molecolare e Biotecnologie Mediche, Università degli Studi di Napoli Federico II and Azienda Ospedaliera Universitaria Federico II, Direzione Sanitaria protocol no. 0007133 of May 8, 2020; (iii) protocol no. 18/20; date: June 10, 2020, Genetics CEINGE TaskForce Covid-19; Azienda Ospedaliera Universitaria Federico II, Direzione Sanitaria protocol no. 000576 of April 10, 2020.

# Data availability

Gene expression (RNA-seq) data are deposited on RNA data bank at EBI: https://www.ebi.ac.uk/biostudies/arrayexpress/studies/E-MTAB-11973; https://www.ebi.ac.uk/biostudies/arrayexpress/studies/E-MTAB-13916. The mass spectrometry proteomics data have been deposited to the ProteomeXchange Consortium via the PRIDE: https://proteomecentral.proteomexchange.org/cgi/GetDataset?ID=PXD051059.

The source data of this paper are collected in the following database record: biostudies:S-SCDT-10_1038-S44319-024-00164-z.

# Peer review information

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

## Acknowledgements

The authors thank Prof. P Forestieri (President, CEINGE) and Dr. M Giustino (CEO, CEINGE) for collaborative support of the program within the Regione Campania Covid-19 Taskforce supporting a BSL3 facility at CEINGE. We thank Prof. Claudia Angelini (Consiglio Nazionale delle Ricerche Institute for Applied Mathematics "Mauro Picone" (IAC) Naples, Italy), and Dr. Simon Zollo (Department of Social Sciences and Economics, Sapienza University of Rome,

Italy) for supporting with the statistical analyses and revisions within this effort. We thank further the European School of Molecular Medicine (SEMM) for a doctorate program Fellowship (F.A.) and the Molecular Medicine and Medical Biotechnology University Federico II of Naples Doctorate Program Fellowship (F.B.). The authors thank Haim Holdings Korea for supporting the doctorate research (F.B.) within the discovery of PI-7 and PI-8 small molecules and NRF 2018R1A5A2025079–Korean Ministry of Science and ICT (J-HC). This research was funded by the project 'CEINGE-TASK-FORCE-2022 COVID19', POR Campania FESR 2014/2020, CUP: D63C22000570002, and 'CEINGE TaskForce COVID19', code D64I200003800 by Regione Campania for the fight against COVID-19, DGR no. 140; 17 March 2020. We additionally thank Italian Association for Cancer Research (AIRC) project IG.22129 (MZ), the Italian Ministero dell'Università e della Ricerca Italiana (PRIN) grant (PRIN 2022), No. 2022MXFLMZ (MZ); Ministero dell'Università e della Ricerca Italiana (PRIN) grant (PRIN 2022), No. 2022T59RWR (VF); PNRR "Partenariato di Neuroscienze e Neurofarmacologia (PE12)—A multiscale integrated approach to the study of the nervous system in health and disease" (MNESYS), University Federico II, MUR: CN00000006—CUP: E63C22002170007 (M.Z); PNRR "National Center for Gene Therapy and Drugs based on RNA Technology" (CN3) University Federico II, CUP: E63C22000940007, Code: CN00000041 (M.Z). We also thank the University of Naples 'Federico II- Spin-Off-Elysium Cell Bio ITASRL. (https://www.elysiumcellbioita.com) developing nutraceutical agents against COVID-19 (IP presented: 102023000016338, 1-08-2023; 1020 24000005170, 7.03.2024).

## Author contributions

**Pasqualino de Antonellis**: Investigation; Writing—original draft. **Veronica Ferrucci**: Investigation; Writing—original draft. **Marco Miceli**: Investigation. **Francesca Bibbo**: Investigation. **Fatemeh Asadzadeh**: Investigation. **Francesca Gorini**: Investigation. **Alessia Mattivi**: Resources. **Angelo Boccia**: Formal analysis. **Roberta Russo**: Visualization. **Immacolata Andolfo**: Visualization. **Vito Alessandro Lasorsa**: Formal analysis. **Sueva Cantalupo**: Investigation. **Giovanna Fusco**: Investigation. **Maurizio Viscardi**: Investigation. **Sergio Brandi**: Investigation. **Pellegrino Cerino**: Investigation. **Vittoria Monaco**: Investigation. **Dong-Rac Choi**: Resources. **Jae-Ho Cheong**: Resources. **Achille Iolascon**: Conceptualization. **Stefano Amente**: Resources; Formal analysis. **Maria Monti**: Investigation. **Luca L Fava**: Resources. **Mario Capasso**: Formal analysis. **Hong-Yeoul Kim**: Resources; Investigation. **Massimo Zollo**: Supervision; Writing—review and editing.

Source data underlying figure panels in this paper may have individual authorship assigned. Where available, figure panel/source data authorship is listed in the following database record: biostudies:S-SCDT-10_1038-S44319-024-00164-z.

## Disclosure and competing interests statement

The authors declare no competing interests.

# Expanded View Figures

**Figure EV1. Deregulated Ca$^{2+}$ pumps during SARS-CoV-2 infection, including ATP2B1. Related to Fig. 1.**

(**A**) A representative immunoblotting analysis using antibodies against the CoV-2 N protein from cells treated with escalating concentration of "gossypol-pubChem CID 3503" at 1 and 5 μM and vehicle as control in SARS-CoV-2 (VOC Δ) infected HEK293T-ACE2 cells. β-Actin was used as the loading control. (**B**) Quantification of Cov-2 N gene (2$^{-\Delta\Delta Ct}$) in gossypol-treated HEK293T-ACE2 (5 μM—48 h) and infected as in (**A**). Cells treated with vehicle were used as negative control. Scattered plot shows the individual value and mean as indicated by the horizontal black lines of $N = 3$ biological replicates. Unpaired two-tailed *T* Student tests Bonferroni corrected. *p < 0.05 (vehicle vs. gossypol); the other comparisons are not statistically significant, as expected. (**C**) A representative immunoblotting analysis using antibodies against the Cov-2 N protein on total protein lysates obtained from human primary epithelial nasal cells treated with BAPTA at 20 μM concentration. Vehicle-treated cells were used as control. β-Actin is used as the loading control. (**D**) RNA Sequencing (RNA-seq) analyses was performed in HEK293T-ACE2 cells treated as in Fig. 2E. (**E**) The Gene Set Enrichment Analysis (see GSEA project on https://doi.org/10.1073/pnas.0506580102) is applied for the identification of deregulated key genes and pathways. KEGG pathways analyses by statistical KS global test, *P* < 0.05. KEGG pathway enrichment analysis indicates those significant deregulated genes were highly clustered in calcium signaling pathway (red box). *P* adj: adjusted *P* values. (**F**) In silico analysis of publicly available datasets of single-cell RNA sequencing (https://singlecell.broadinstitute.org) for the expression of the plasma membrane calcium ATPases members (PMCAs or ATP2B1-4) of the large family of type Ca$^{2+}$ion pumps in multiple cell type in the lung parenchyma (including alveolar macrophages and in the alveolar epithelial cells type I and type II). (**G**) Literature public search on available datasets obtained from a single-nuclei RNA-seq (snRNA-seq) on >116,000 nuclei from n.19 COVID-19 autopsy lungs and n.7 pre-pandemic controls (Melms et al, 2021); to verify expression of PMCAs and SERCAs pumps (ATP2B1-4 and ATP2A1-3 genes, respectively). The numbers within the dots are the median percentage level of expression between the two populations tested. (**H**) Expression levels of ATP2B1 in single-nuclei RNA-seq database (snRNA-seq) as described above in (**G**). Source data are available online for this figure.

▶

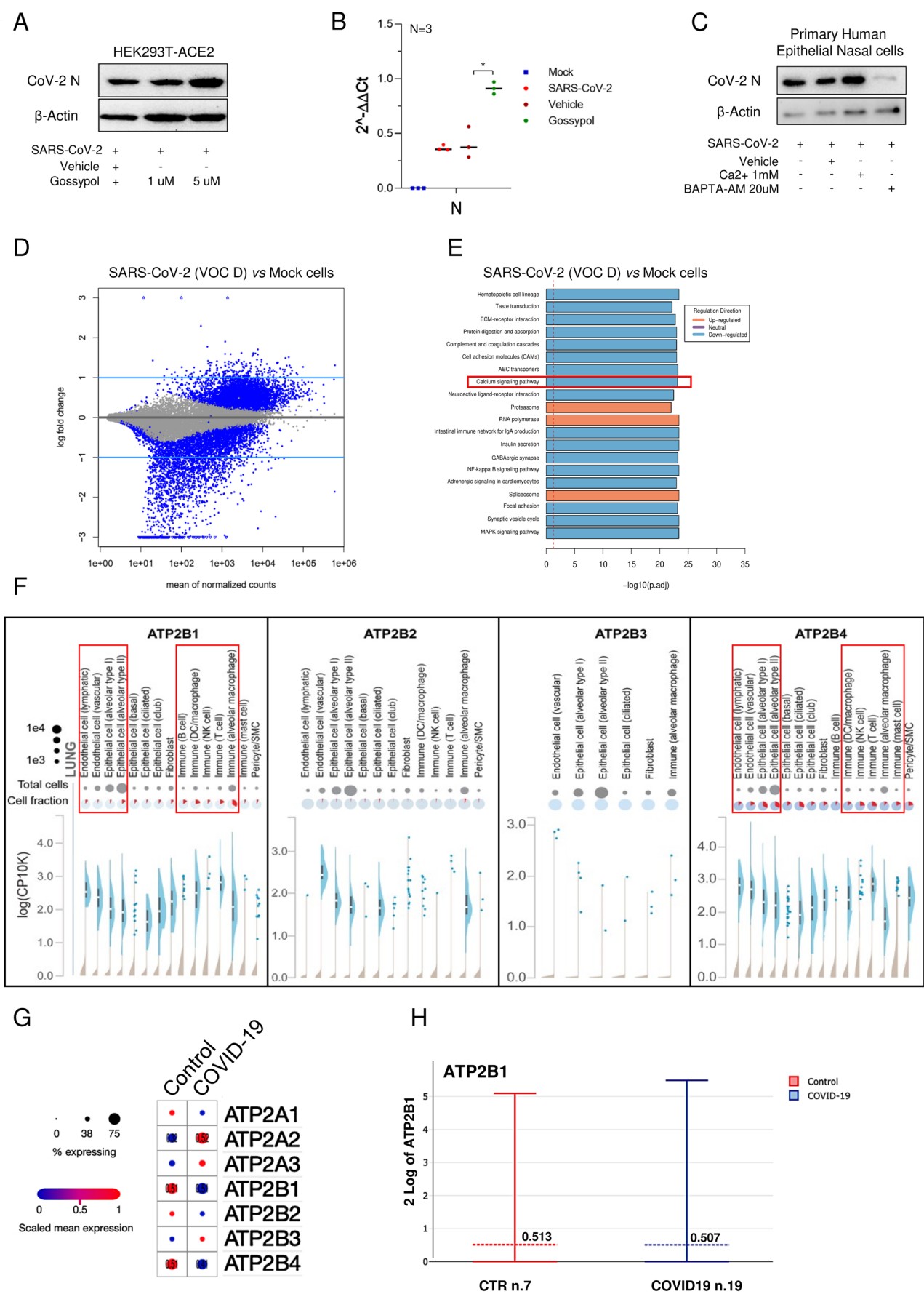

**A**

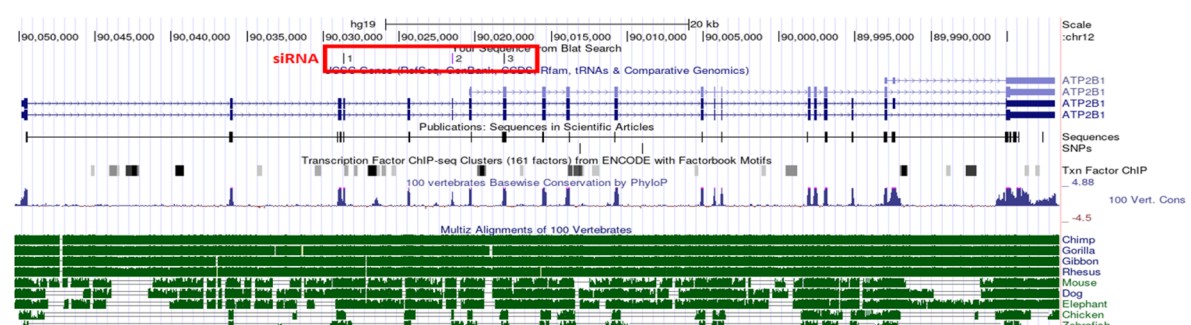

**B**

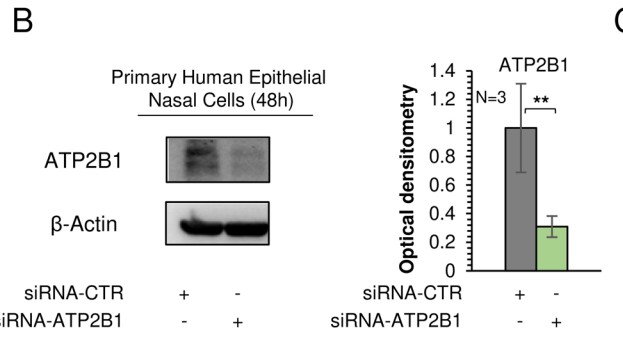

**C**

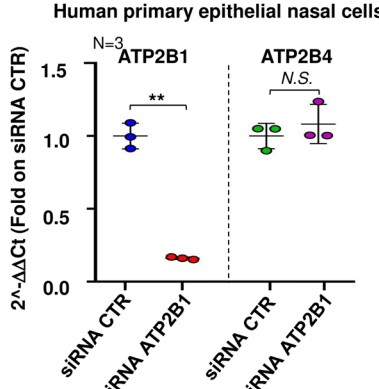

**D**

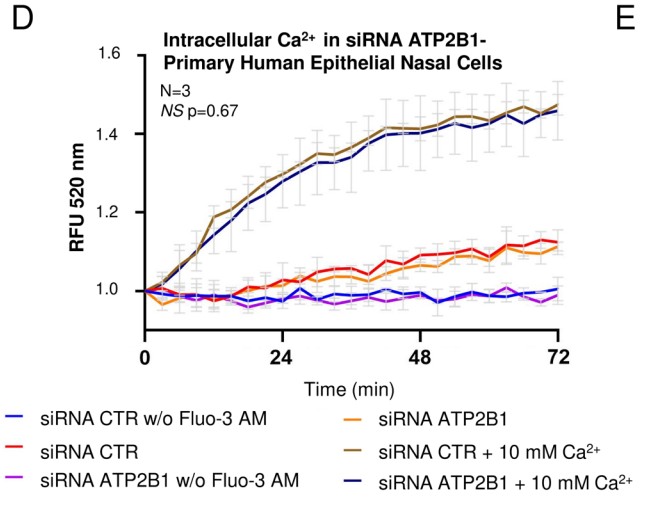

**E**

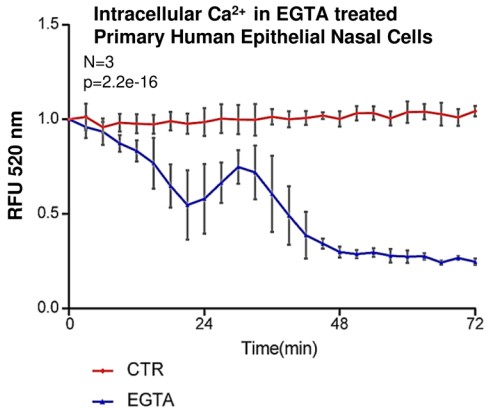

siRNA CTR w/o Fluo-3 AM

siRNA CTR

siRNA ATP2B1 w/o Fluo-3 AM

siRNA ATP2B1

siRNA CTR + 10 mM Ca²⁺

siRNA ATP2B1 + 10 mM Ca²⁺

**F**

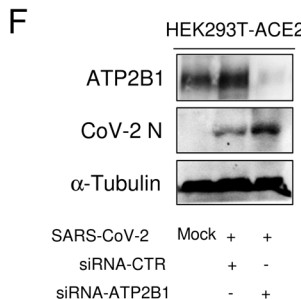

**Figure EV2.   Reduced ATP2B1 protein levels promote SARS-CoV-2 replication. Related to Fig. 2.**

(A) Representation of human ATP2B1 region recognized by siRNA as reported in the UCSC Genome Browser on Human Dec. 2013 (GRCh38/hg38) Assembly (https://genome.ucsc.edu/). At the bottom, the alignment of this genomic region among different species is shown. (B) Left: representative immunoblotting analysis using antibodies against the ATP2B1 protein on human primary epithelial nasal cells transiently treated with siRNA against ATP2B1 (siRNA-ATP2B1) for 48 h. Cells treated with a pool of three unrelated siRNAs (siRNA- CTR) were used as negative controls. β-Actin is used as the loading control. Right: Densitometric analysis of the ATP2B1 band intensities in blots. Data are means ± SD of $N = 3$ biological replicates. Unpaired two-tailed $T$ Student tests, $**P < 0.01$. (C) Quantification of mRNA abundance for ATP2B1 and ATP2B4 ($2^{-\Delta\Delta Ct}$) in cells as treated as in (B). Data are means ± SD of $N = 3$ biological replicates. Unpaired two-tailed $T$ Student $t$ test, $**P < 0.01$; NS not significant. (D) Quantification of relative fluorescence changes of Fluo3-AM as a measure of intracellular $Ca^{2+}$ levels in cells treated as in (B) for up to 72 min. Results are expressed as means ± SEM of $N = 3$ biological replicates. One-way ANOVA and KS test, $P = 0.6748$; NS = not significant between siRNA control (brown) and siRNA-ATP2B1 (dark blue) in presence of 10 mM $Ca^{2+}$. (E) Quantification of relative fluorescence changes of Fluo3-AM as a measure of intracellular $Ca^{2+}$ levels for up to 72 min in human primary epithelial nasal cells treated with EGTA 1 mM. Results are expressed as means ± SEM of $N = 3$ biological replicates. One-way ANOVA and KS tests, $P = 2.2e\text{-}16$. (F) A representative immunoblotting analysis using antibodies against the ATP2B1 protein for total protein lysates obtained from cells treated as described in Fig. 2E. α-Tubulin is used as loading control. Mock-infected cells are used as control. Source data are available online for this figure

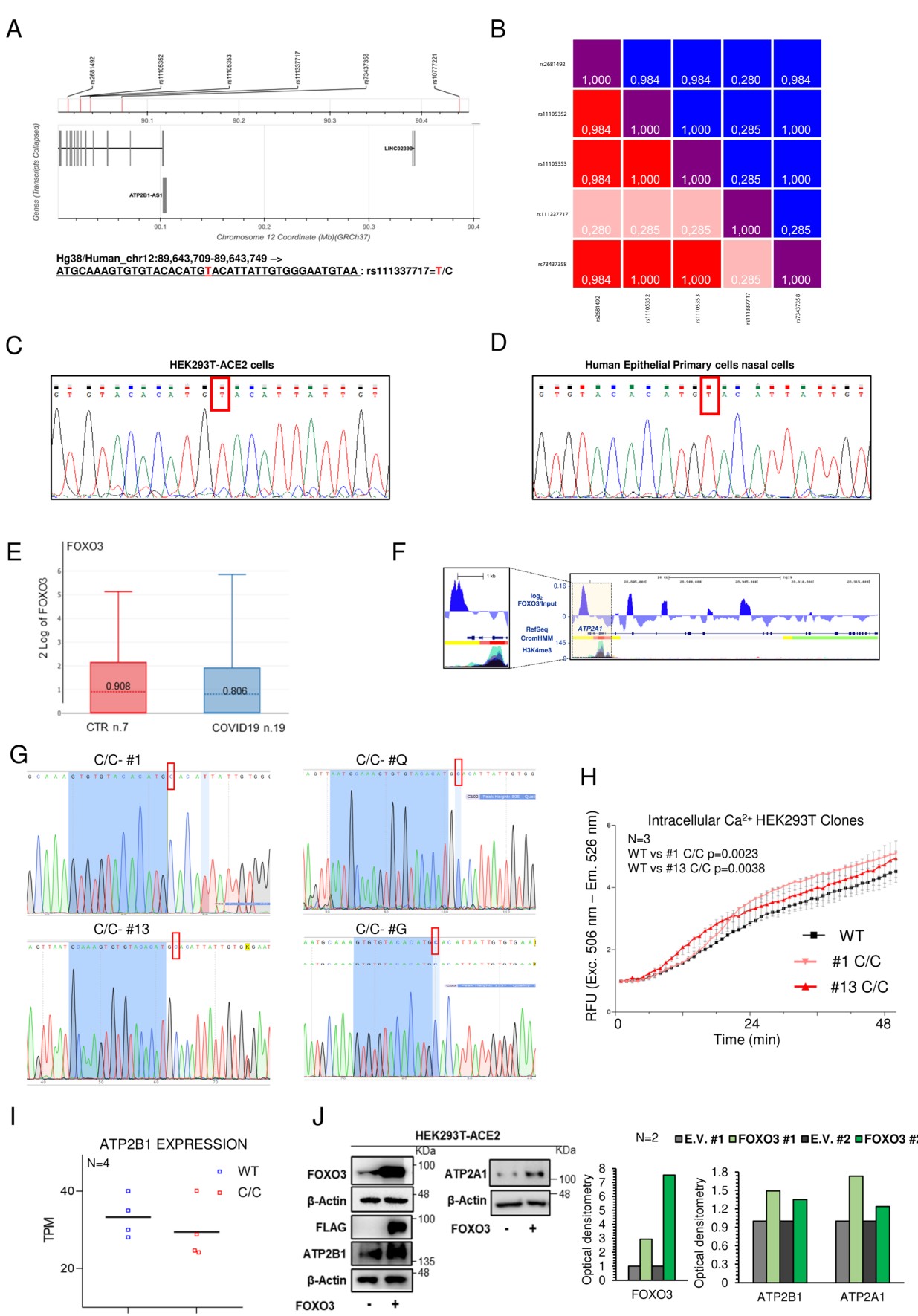

◀ **Figure EV3. The homozygous intronic *ATP2B1* variant rs11337717 is responsible for increased SARS-CoV-2 replication in COVID-19 patients by transcriptional regulation of FOXO3. Related to Fig. 3.**

(A, B) Linkage disequilibrium (LD) analyses on the top n.5 SNPs (rs11105352; rs11105353; rs73437358; rs111337717; rs2681492) in order to select those which are independent. The SNP rs10777221 is excluded from these analyses because located at most 5' region in extragenic ATP2B1 locus region (A). The graph in (B) shows the only SNP not in LD is rs111337717 (black boxes). (C) Sanger DNA sequencing of the genomic region of ATP2B1 locus (chr12:89,643,709-89,643,749) in HEK293T-ACE2 cells to exclude the presence of intronic variance potentially responsible for altered transcriptional levels of ATP2B1 gene. The red box indicates the nucleotide wild type allele "T" for the SNP here studied. (D) Sanger DNA sequencing of the genomic region of ATP2B1 locus (chr12:89,643,709-89,643,749) in human primary epithelial nasal cells to exclude the presence of intronic variants potentially responsible for altered transcriptional levels of ATP2B1 gene. (E) FOXO3 Expression in UMAP by disease ontology labels single-nuclei RNA-seq (snRNA-seq) analyses performed on >116,000 nuclei from n.19 COVID-19 autopsy lungs and n.7 pre-pandemic controls. Data measurements values of $Log^2$ FOX3 mRNA expression: in CTR donors n.7: max=5.856, center= 0.806, min=0; in COVID-19 n.19 affected patients: max=5.128, center=0.908, min=0. (F) Genome browser screenshots showing accumulation of normalized FOXO3 signal, together with CromHMM state segmentation and H3K4me3 signal (ENCODE), along the ATP2A1 gene in human cells. ForCromHMM state segmentation colors indicate: Bright Red—Promoter; Orange and yellow— enhancer; Green— Transcriptional transition. The expanded view of the highlighted region, on the left, shows FOXO3 peaks over ATP2A1 enhancer regions, as marked by yellow region of CromHMM. (G) Sanger DNA sequencing of the genomic region of ATP2B1 locus (chr12:89,643,709-89,643,749) in HEK293T-ACE2 relative to the CRISPR/Cas9 edited clones to show the presence of intronic homozygous variant (C/C) responsible for altered transcriptional levels of ATP2B1 gene. The red box indicates the nucleotide edited for the SNP here studied. (H) Quantification of relative fluorescence changes of Fluo3-AM as a measure of intracellular $Ca^{2+}$ levels for up to 48 min in HEK293T isogenic clones. Results are expressed as means ± SEM of $N = 3$ biological replicates. One-way ANOVA and KS test. In details: WT vs. CC $P = 0.0023$; WT vs. CC $P = 0.0038$. (I) Quantification of ATP2B1 mRNA abundance in HEK293T ($N = 4$ homozygous C/C) isogenic clones compared to ($N = 4$ WT T/T) unedited clones. mRNA levels measured by RNA-seq were plotted using transcript per million (TPM). Scattered plots show individual value and mean as indicated by the horizontal black lines of $N = 4$ biological replicates. Fold change value $+/-2$. Statistical Mobin Wald test NS not significant. (J) Left: representative immunoblotting analysis using antibodies against FOXO3, ATP2B1, ATP2A1 as indicated proteins on total protein lysates obtained from HEK293T-ACE2 cells transiently transfected with the human FOXO3-encoding plasmid (FLAG antibody positive) for 48 h. Empty vector transfected cells were used as negative control. β-Actin is used as the loading control. On the right: Densitometric analysis of FOXO3, ATP2B1 and ATP2A1 from $N = 2$ technical replicates. Source data are available online for this figure

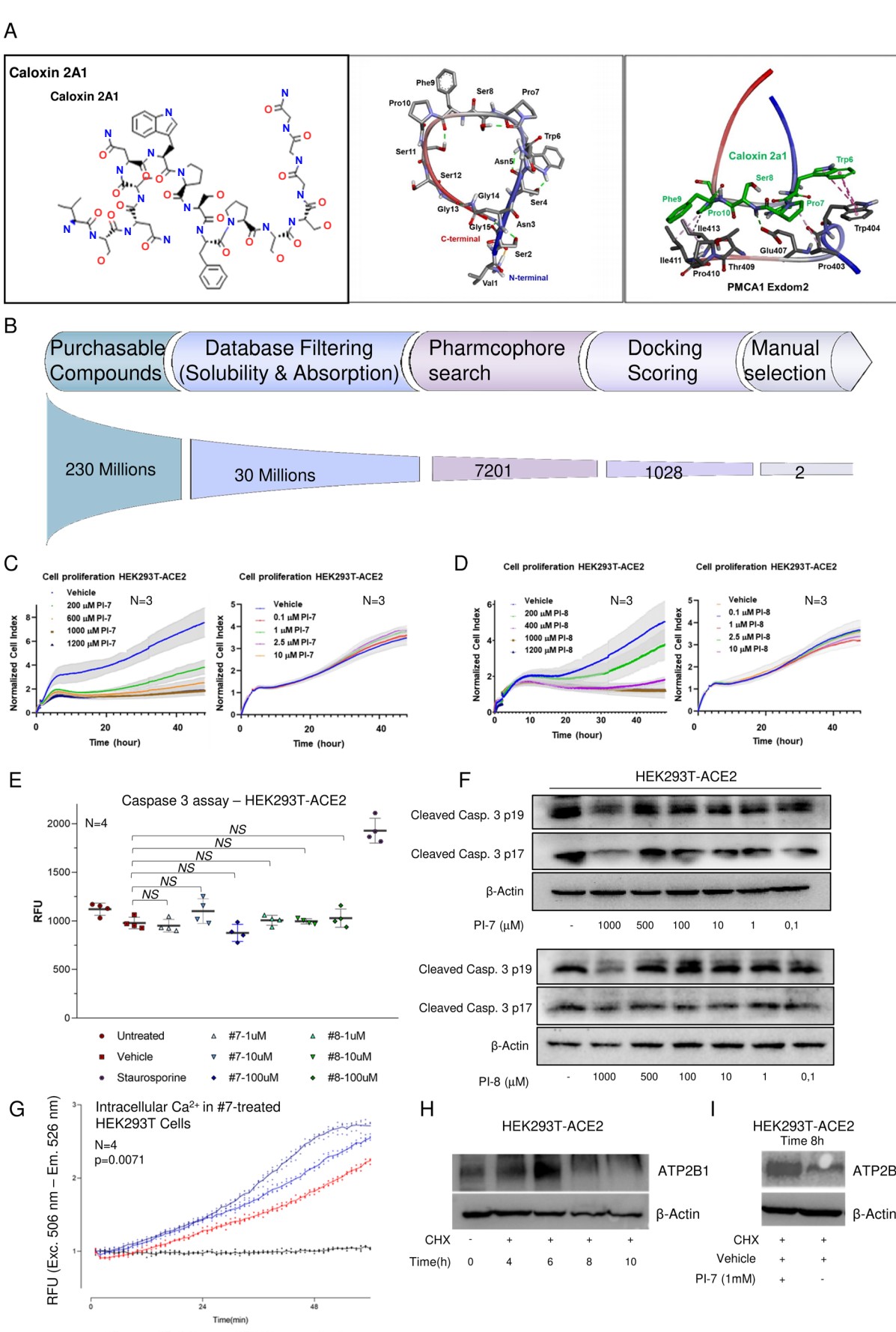

**Figure EV4. ATP2B1 impairment using a nontoxic "caloxin derivative" (compound PI-7) impairs intracellular Ca²⁺ levels. Related to Fig. 4.**

(A) On the left: The sequence of caloxin 2a1 sequence, as peptide, is shown. On the right: The molecular modeling of ATP2B1-caloxin 2a1 structure by docking and energy minimization modeling via artificial intelligence as a drug design computational tool is shown. The pharmacophore model by using the structures ATP2B1–exodom-2 and caloxin 2a1 is also shown. Five pharmacophore features were produced. (B) Pipeline of the drug discovery is shown as described in the manuscript. (C, D) Real-time cell proliferation analyses for the Cell Index (i.e., the cell-sensor impedance was expressed every two minutes as a unit called "Cell Index"). Results are expressed as means ± SEM of $N = 3$ biological replicates. HEK293T-ACE2 treatment described in the Methods are treated with escalating doses of PI-7 (C) or PI-8 (D); with vehicle-treated cells were the negative control. Impedance was measured every 2 min over 48 h. The graphs showing "normalized cell index" were generated using Graph Pad Prism 9. (E) Caspase-3 activity measured in HEK293T-ACE2 cells with increasing concentrations of compound PI-7 and PI-8 for 18 h. Vehicle-treated cells and cells treated with 10 μM staurosporine are used as negative and positive controls, respectively. Data are presented as relative fluorescent units (RFUs; excitation: 380 nm; emission: 460 nm). Results are expressed as means ± SEM of $N = 3$ biological replicates. NS not significant. One-way ANOVA test among multiple groups, Untreated vs. Vehicle $P = 0.1596$, Vehicle vs. PI-7-1 μM p = 0.9993, Vehicle vs. PI-7-10 μM $P = 0.3039$, Vehicle vs. PI-7-100 μM $P = 0.5176$, Vehicle vs. PI-8-1 μM $P = 0.9992$, Vehicle *vs.* PI-8-10 μM $P > 0.9999$, Vehicle vs. PI-8-100 μM $P = 0.9732$), NS not significant. (F) A representative immunoblotting analyses on total protein lysates obtained from HEK293T-ACE2 treated with escalating doses of PI-7 (top) and PI-8 (bottom) molecules using antibodies against Cleaved Caspase-3 fragments (17–19 kDa). β-Actin is used as the loading control. Vehicle-treated cells are used as a negative control of the experiment. (G) Quantification of relative fluorescence changes of Fluo3-AM as a measure of intracellular Ca²⁺ levels for up to 48 min in HEK293T cells treated with 10 μM of PI-7, vehicle-treated cells were used as negative control. Results are expressed as means ± SEM of $N = 4$ biological replicates. Fd ANOVA global - CH test, $P = 0.007$. (H) A cycloheximide (CHX) chase assay, representative immunoblotting analyses on total protein lysates obtained from HEK293T-ACE2 treated with CHX at different time point (from T = 0 to T = 10 h) using antibodies against ATP2B1 and β-Actin used as the loading control. Vehicle-treated cells (i.e., 0.001% DMSO) are used as negative control of the experiment. (I) A Representative immunoblotting analyses on total protein lysates obtained from HEK293T-ACE2 treated with CHX and PI-7 for 8 h, using antibodies against ATP2B1. β-Actin is used as loading control. Vehicle-treated cells are used as negative control. Source data are available online for this figure

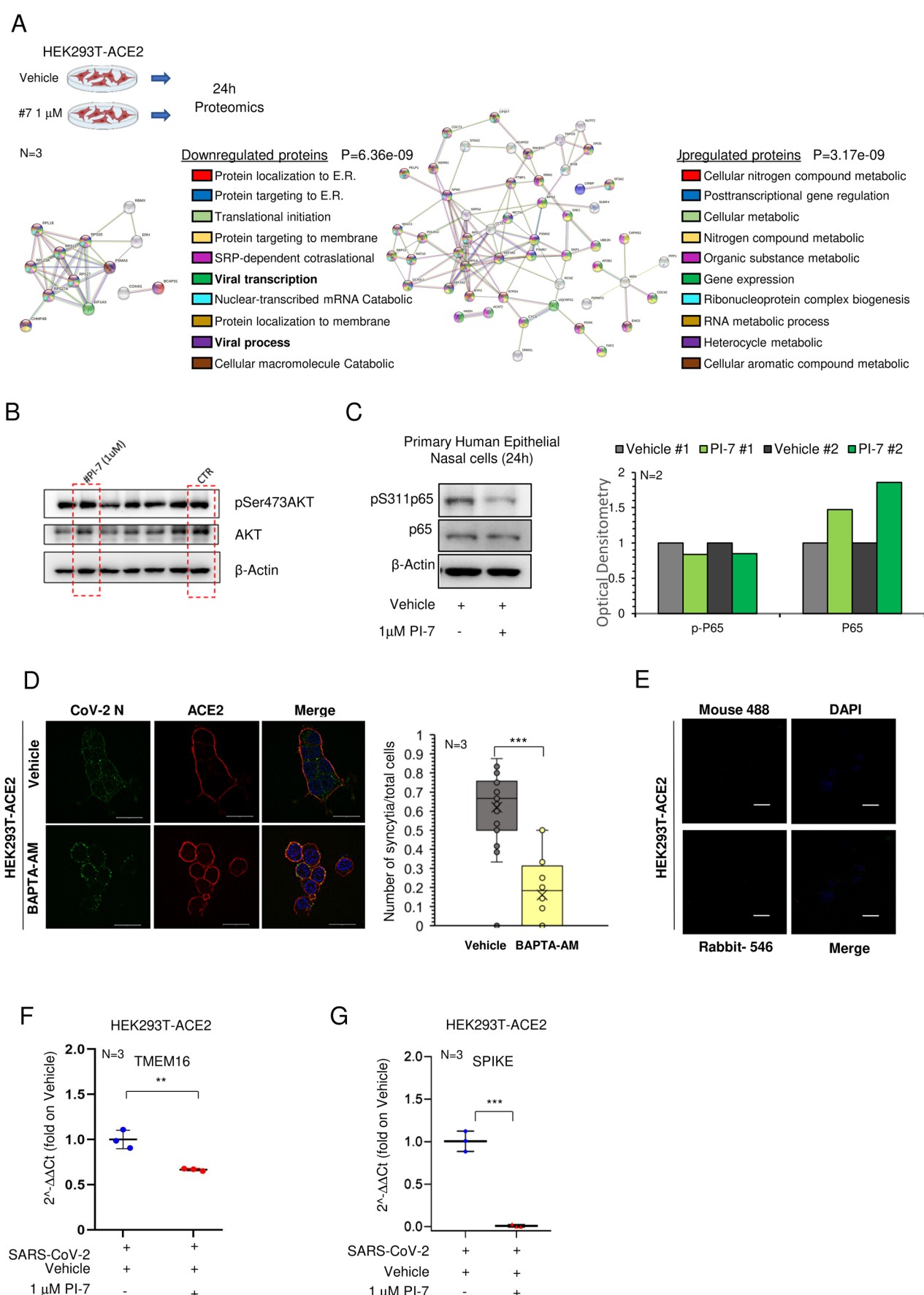

**Figure EV5.  Compound PI-7 diminishes SARS-CoV-2 replication by affecting viral processes, syncytia formation and inflammatory pathways. Data related to Fig. 5.**

(A) Top panel: a proteomic assay based on LC-MS/MS approach performed on HEK293T-ACE2 cells treated with PI-7 molecule (1 μM) for 24 h, $N = 3$ biological replicates. Bottom left: A protein interaction network was generated using the Search Tool for the Retrieval of Interacting Genes/ Proteins (STRING) database (https://string-db.org), using only those proteins that were downregulated in PI-7- treated cells within "viral process" and "viral transcription" category functions, in bold (i.e., n.18 downregulated proteins, Appendix 7). One-way ANOVA, $P = 6.36$ e09. $N = 3$ biological replicates. Bottom right: A protein interaction network was generated using STRING database (https://string-db.org) by using only those proteins found upregulated in PI-7-treated cells (i.e., n.66 upregulated protein, with different category functions, Appendix 8). One-way ANOVA test, $P = 3.17$e09. (B) A representative immunoblotting analysis on uninfected cells using antibodies against the pSer-473AKT and AKT proteins on total protein lysates obtained from HEK293T-ACE2 treated with compound PI-7 (1 μM-dashed lines) or vehicle-treated (CTR dashed lines) for 24 h. β-Actin is used as the loading control. (C) A representative immunoblotting analysis using antibodies against the pS311–p65 and p65 proteins on total protein lysates obtained from human primary epithelial nasal cells treated with compound PI-7 (1 μM) for 24 h. β-Actin is used as the loading control. Densitometric analysis from $N = 2$ technical replicates. (D) IF with an antibody against viral CoV-2 N (green) and human ACE2 (red) proteins in HEK293T-ACE2 cells treated with BAPTA-AM (20 μM) and infected with SARS-CoV-2 for 72 h (i.e., treated as in Fig. 4G). Right: The graph showing the intensity of fluorescence is shown on the left. Data are means ± SD of $N = 3$ biological replicates. Unpaired two-tailed T Student test, ***$P < 0.001$. Data measurements values: vehicle-treated min=0.333, max=0.875, center=0.667, bounds of box= 0.5–0.757 and whiskers = 0, percentiles= 0.064 ($K = 0.01$) − 1.122 ($K = 0.99$); BAPTA-AM treated: min=0.0, max=0.5, center=0.183, bounds of box=0–0.312, whiskers=none, percentiles= 0 ($K = 0.01$) − 0.5 ($K = 0.99$). The SIM² image are acquired with Elyra 7 (Zeiss) and processed with Zeiss ZEN software (blue edition). Magnification, ×63. Scale bar, 20 μm. (E) A representative IF staining with secondary antibodies anti-rabbit Alexa Fluor 546 (1:200; #A10040, Thermo Fisher Scientific) or anti-mouse Alexa Fluor 488 (1:200, ab150113, Abcam) on HEK293T-ACE2 cells. DAPI is used for nuclear staining (blue). The image was acquired with Elyra 7 (Zeiss). Magnification: 40×; Scale bar, 20 μm. (F) QPCR of mRNA abundance relative to that in control (CTR) cells ($2^{-\Delta\Delta Ct}$) for human TMEM16 gene. RNA extracted from HEK293T-ACE2 cells treated as in Fig. 4G). Scattered plots show the individual values ad mean as indicated by the horizontal black lines of $N = 3$ biological replicates. Unpaired two-tailed T Student test and Bonferroni corrected **$P < 0.01$. (G) QPCR of mRNA abundance relative to that in control (CTR) cells ($2^{-\Delta\Delta Ct}$) for SARS-CoV-2 SPIKE gene from HEK293T-ACE2 cells treated as in Fig. 4G. Scattered plots show the individual values ad mean as indicated by the horizontal black lines of $N = 3$ biological replicates. Unpaired two-tailed T Student test ***$P < 0.001$. Source data are available online for this figure

