## [Peer Review File · EMBO Reports]

Targeting ATP2B1 impairs PI3K/Akt/FOXO signaling and reduces SARS-COV-2 infection and replication.

Pasqualino de Antonellis, Veronica Ferrucci, Marco Miceli, Francesca Bibbo, Fatemeh Asadzadeh, Francesca Gorini, Alessia Mattivi, Angelo Boccia, Roberta Russo, Immacolata Andolfo, Vito Alessandro Lasorsa, Sueva Cantalupo, Giovanna Fusco, Maurizio Viscardi, Sergio Brandi, Pellegrino Cerino, Vittoria Monaco, Dong-Rac Choi, Jae-Ho Cheong, Achille Iolascon, Stefano Amente, Maria Monti, Luca L. Fava, Mario Capasso, Hong-Yeoul Kim, and Massimo Zollo

Corresponding author(s): Massimo Zollo (massimo.zollo@unina.it)

Review Timeline:

Submission Date:	7th Sep 22
Editorial Decision:	27th Oct 22
Appeal Received:	8th Mar 24
Editorial Decision:	22nd Mar 24
Revision Received:	9th Apr 24
Editorial Decision:	23rd Apr 24
Revision Received:	2nd May 24
Accepted:	7th May 24

Editor: Achim Breiling

Transaction Report:

Dear Dr. Zollo,

Thank you for the submission of your research manuscript to EMBO reports. I have now received the reports from the three referees that were asked to assess the manuscript that are copied below. This took significantly longer than usual, but we depend on our referees that in this case took much more time for the review than originally agreed on.

I am sorry to say that the decision on your manuscript is not a positive one. As you will see, referees #1 and #2 indicate that the data presently are rather premature and that they do not sufficiently support the conclusions. Referee #1 in particular indicates that the physiological relevance of the findings remains unclear and also indicates that the molecular explanation of the action of identified inhibitor(s) is not convincing or sufficient (a point that was also raised by referee #2). Moreover, both referees note deficiencies in the experimental approach and the data presentation, and also indicate missing controls and other technical shortcomings. Referee #3 is more positive, but also has several concerns. As the reports are below, I will not further detail them here.

Given the comments of the referees, the amount of work required to address them, and the fact that EMBO reports can only invite revision of papers that receive enthusiastic support from all the referees upon initial assessment, I cannot offer to publish your manuscript.

I am sorry to have to disappoint you this time. I nevertheless hope that the referee comments will be helpful in your continued work in this area, and I thank you once more for your interest in our journal.

Yours sincerely

Referee #1:

In this manuscript, Antonellis and colleagues investigated the involvement of the ATP2B1 membrane Ca²⁺ pump in SARS-CoV-2 infection. The authors demonstrated that knockdown of ATP2B1 attenuated SARS-CoV-2 infection (based on the amount of SARS-CoV-2 N protein), and SARS-CoV-2 infection downregulated ATP2B1 expression in primary human nasal epithelial cells. The authors further developed a new drug that inhibits ATP2B1 function and SARS-CoV-2 replication in vitro. It would be appreciated that the authors explored the role of the membrane Ca²⁺ pump in virus infection; however, this study is too descriptive, molecular mechanisms are essentially still ambiguous, and the presented data do not sufficiently support the authors' conclusions. Therefore, it is unclear how broadly significant the findings are, and this reviewer deems that the manuscript might be more suitable for a specialized journal.

Major concerns:

- 1 It is essential to demonstrate the involvement of ATP2B1 in viral infection in vivo. Therefore, the authors need to prove the effect of PI-7 treatment on SARS-CoV-2 infection in animal models.
- 2 It is a superb discovery that the authors identified an SNP in the ATP2B1 gene frequently observed in severe cases of COVID-19. This finding led to identifying FOXO3 as a transcriptional regulator of ATP2B1. However, it is mandatory to perform promoter assays to check whether the regulation of ATP1B1 expression by FOXO3 is affected by the SNP to support the authors' conclusion.
- 3 The method and data for quantification of viral replication are not convincing.
 - 3.1 Viral replication should be examined by the qPCR method, which ensures the quantification, rather than immunoblot analysis of SARS-CoV-2 N protein (Figures 1H, 2E, 3E, and 5B).
 - 3.2 Additional appropriate controls should be provided in some experiments.
 - a. Figure 1H: mock-infected and untransfected controls are needed because N protein expression in the siControl-transfected cells appears to be substantially low compared to that in virus-infected (untransfected) cells in Figure 1C. Does siControl inhibit virus infection?
 - b. Figure 5B: Additional validation in mock-infected cells is needed.
 - c. Figures 5D and 5E: The authors need to include a negative control for immunofluorescence, including mock-infected cells, to guarantee the specificity of the antibody against the N protein.
- 4 The methodology and data/interpretation for measuring extracellular Ca²⁺ concentrations are unconvincing.
 - 4.1 Fluo 3-AM, an acetoxymethyl derivative of Fluo 3, is an inactive form as it is and becomes active to measure Ca²⁺ only after hydrolysis by intracellular esterase. If one wants to measure extracellular Ca²⁺ concentration, Fluo 3, but not its AM derivative, must be used.
 - 4.2 Even if the AM derivative was converted to Fluo 3 by serum constituents in the culture medium, the property of Fluo 3 (K_d = 400 nM) is not suited for measuring extracellular Ca²⁺, the concentration of which is mM level. Once a decrease in fluorescence

intensity is detected by Fluo 3, the extracellular Ca²⁺ concentration drops by 1/10,000-fold! Such a robust decrease in Ca²⁺ in the medium (which is present in a large quantity) cannot be accounted for by changes in intracellular Ca²⁺ (marginal of the amount). Furthermore, such extreme Ca²⁺ deprivation will result in apparent cytotoxicity (which contradicts the authors' claim).
5 Although PI-7 was identified as an ATP2B1 inhibitor, an underlying, precise molecular mechanism of ATP2B1 inhibition remains to be elucidated.

5.1 The established function of ATP2B1 is to pump Ca²⁺ from the cytosol to the outside. Nevertheless, PI-7 treatment decreased intracellular Ca²⁺ concentration (Figure 4F).

5.2 Alternatively, this reagent upregulated the expression of ATP2B1 (Figure 5C), which might be more likely to lead to a decrease in intracellular Ca²⁺ concentration. In fact, the treatment of PI-7 also affects the expression and phosphorylation levels of Akt and FOXO3, signaling factors of the pathway that the authors identified as a regulator of ATP2B1 expression during viral infection. In this case, another(s) target of this reagent should exist other than ATP2B1.

6 The relationship among intracellular calcium levels, ATP2B1 expression, and viral infection needs to be further examined in detail and appropriately.

6.1 The authors speculated that knockdown of ATP2B1 did not alter intracellular Ca²⁺ concentration (Figure 2C) due to complementary upregulation of other Ca²⁺ pump paralogues. If this is true, a decrease in ATP2B1 levels by SARS-CoV-2 infection may also not change intracellular Ca²⁺ concentration or virus infection through the complementary expression of such pumps, which contradicts the promotion of the infection in ATP2B1 knockdown cells (Figure 2E). Hence, additional appropriate experiments to elucidate the mechanism of this discrepancy are required.

6.2 It is necessary to verify whether SARS-CoV-2 infection indeed increases intracellular Ca²⁺ concentration. In addition, intracellular Ca²⁺ concentration after SARS-CoV-2 infection should be measured in ATP2B1-knockdown or -overexpressing cells.

6.3 Reduced intracellular Ca²⁺ concentration by PI-7 was only examined at 100 μM (Figure 4). Thus, it is unclear whether the Ca²⁺ concentration in the cells treated with 1 μM PI-7 was indeed reduced in Figures 5 and 6. The intracellular Ca²⁺ in the cells treated with 1 μM PI-7 should be analyzed to confirm a decreased intracellular Ca²⁺ concentration.

General concerns:

1 Texts in the Figures cannot be distinguished due to low image resolution (Figures 1D, 3C, S2A, S3A, and S6A)

2 Please be advised to consult with an expert in statistics. For example, the authors utilized unpaired two-tailed Student's t-tests; however, this test compares the difference between the means of two data sets, but not the difference among multiple conditions (Figure 1A, 4E, 4F, and 6B). In addition, the authors should perform statistical tests and provide the p values in the time course experiments (Figures 2B and 2C).

Minor comments:

1 Page15, line 22: HEK193T might be a typo.

2 "∞" in Figures 4C, 4D, 5D, and 5E should be replaced by "μ"?

3 Figures 5D and 5E: The fluorescence signal derived from the N protein is too faint. In addition, these lack scale bars and the information on the arrows in the figures.

4 Some references are cited inappropriately.

4.1 Page7, line 8: Neither Zhou's nor Chen's paper describes SARS-CoV-2 infection. Note that they were published in 2019 or earlier.

4.2 Page 19, lines 13-14: Although the authors cite the paper by Shang et al. to explain the requirement of Ca²⁺ for the binding of SARS-CoV-2 to ACE2, the article does not provide such data.

Referee #2:

The work describes the role of ATP2B1 and ATP2A1 in SARS-CoV-2 infection. The choice of studying ATP2B1 is mostly due to a demonstrated reciprocal relationship between its expression and virus replication, which has merit. The work is extensive and experiments are executed and reported in good detail. However, interpretation of the data may be premature and it is recommended that many of the assertions of what is being seen be revised.

The work begins by showing that the ATP2B1 calcium channel is down-regulated upon SARS-CoV-2 infection. The change in ATP2B1 expression is 5-fold and significant while the control, b-actin remains constant. This is good but it is important to reflect that positive sense viruses like SARS-CoV-2 express their viral mRNAs through a process of cap snatching. This results in degradation of many cell mRNAs. Indeed, it is indicated that many calcium channels and other proteins are down-regulated and likely reflects this process of mRNA destabilization. B-actin may not be the best control in this situation as the protein has a low turnover. Instead, it would be better to use a house keeping gene with better turnover such as GAPDH.

The relationship of ATP2B1 expression, calcium levels in cells and infection is then evaluated using calcium sensitive dyes. This work appears well done with good controls and obtains large differences in outcomes, supporting the hypothesis.

Next, SNP analysis is used to look at natural differences in ATP2B1 genes in the population. However, the gene sequence is highly conserved with only one change being present in an intron. The relevance of this change is not explored, so while an interesting side note, this information is not useful for understanding the role of the protein in infection by SARS-CoV-2 and should be removed or minimized and put in the discussion.

ATP2B1 gene regulation is then assessed. FOXO3 is indicated to control ATP2B1 expression. This is shown by over-expression of FOXO3, which results in more ATP2B1 and A1 expression and appears a strong outcome.

Next, two small molecules are made. PI-7 and PI-8. These were built based on a known pharmacophore and docking approaches onto ATP2B1. It is shown that it is not toxic below 200 μM . Of note, all the figures have μ (u in μM) incorrectly shown as an infinity symbol - please correct. When cells are treated with 100 μM of each, a modest reduction (~20%) in calcium release into the extracellular fluid or seen intracellularly. This difference increases when low glucose medium is used (30% change). We are not shown how lower concentrations of each compound affects calcium levels. A control for this experiment should be BAPTA and BAPTA-AM. This is a widely accepted chelator of calcium and will help to better understand the baseline of the assays used. A dose curve for PI-7 is important since the next experiments use PI-7 at 100-fold lower dose, at 1 μM . It is important to tell the reader why this much lower amount is used and to provide a dose response curve for calcium mobilization. Also, all the measurements are indirect using whole cells and a fluorescent dye readout. A patch clamp of the ATP2B1 protein would be better to directly measure the impact of PI-7 on calcium flux.

In the next experiment 1 μM PI-7 is used to show the potential inhibition of ATP2B1 has effects on expression of ATP2B1, FOXO3 and PI3K activity. ATP2A1 expression is also elevated, presumably to compensate for loss of ATP2B1 function. The confusing aspect is the authors argue that loss of ATP2B1 function is due to PI-7 inhibition and this somehow results in FOXO3 and PI3K changes. This seems to be the wrong way around as the protein expression would be controlled by the transcription factors and not the other way. A feedback model would provide a better explanation of this outcome but would need to be shown. Syncytia formation are also looked at. However, the relevance of this as a measure of virus infectivity is obscure as syncytia are often the product after infection of a cell and protein expression. A drop in spike protein expression, would result in less syncytia. The authors need to better describe the relevance of this work.

Lastly, PI-7 treatment effects on ATP2B1, A1 and production of 3 cytokines after infection. It is interesting that PI-7 at 1 μM is able to restore cell functionality back to uninfected levels in infected cells. This is an interesting result but because PI-7 is not well characterized for direct effects on ATP2B1, the mechanism, while suggested, remains uncertain.

Overall, this interesting body of work is diminished by poor characterization of PI-7 and how it interacts with ATP2B1. Patch clamp or other biochemical type assay that can measure direct interaction with ATP2B1 should be attempted and allow the experiments to be more easily interpreted.

Referee #3:

The manuscript by Antonellis et al deals with the Ca^{2+} -dependency of SARS-CoV-2 infection and replication. ATP2B1 is a known Ca^{2+} regulator of cellular export and homeostasis and the authors found that a new nontoxic caloxin-derivate (PI-7) inhibits ATP2B1 which leads to diminished extra- and intracellular Ca^{2+} levels and the impairment of SARS-CoV-2 replication. Therefore, it is proposed to use PI-7 as prophylactic therapy as it reduces intracellular Ca^{2+} levels.

Furthermore, the authors discovered a FOXO3 transcriptional site with a rare intronic variant which is associated with severity of COVID19. During infection, the PI3K/Akt signalling pathway is activated, FOXO3 inactivated and in turn the transcriptional control of Ca^{2+} pumps like SERCA inhibited.

The manuscript is well written (although I strongly recommend passing it to a native speaker for cross-check). The findings are novel, and the experiments shown were carried out thoroughly. To my opinion, the findings represent important milestones that can advance research on SARS-CoV-2.

I do have some minor comments that can improve the quality of the manuscript:

- How frequent is the intronic variant on the FOXO3 transcriptional site within the population? This should be stated in the manuscript.
- Figure 1 shows gene expression analysis of infected cells compared to non-infected ones. These experiments were carried out in HEK293T cells. To my knowledge, HEK cells also endogenously express ACE2 - how can the authors be sure that they did not measure artifacts?
- Ca^{2+} homeostasis within cells is also dependent on CRAC channels. Did the authors take the possibility into account that these types of channels also play a role during SARS-CoV-2 infection/replication?
- All Western Blots in the figures are just cut-outs. Whole blots have to be shown in appendix.
- The fluorescent images in figure 5 are quite dark - is there a better resolution achievable? In addition, the arrows in figure 5 are too big and cover the spots that should be seen.

** As a service to authors, EMBO Press provides authors with the ability to transfer a manuscript that one journal cannot offer to publish to another journal, without the author having to upload the manuscript data again. To transfer your manuscript to another EMBO Press journal using this service, please click on
Link Not Available

Naples, 08 March 2024:

EMBOR-2022-56072V1: POINT-BY POINT RESPONSE

We would like to thank the reviewers and the Editor for the constructive comments. Here below, we provide a point-by-point response to the concerns raised.

Referee #1:

In this manuscript, Antonellis and colleagues investigated the involvement of the ATP2B1 membrane Ca²⁺ pump in SARS-CoV-2 infection. The authors demonstrated that knockdown of ATP2B1 attenuated SARS-CoV-2 infection (based on the amount of SARS-CoV-2 N protein), and SARS-CoV-2 infection downregulated ATP2B1 expression in primary human nasal epithelial cells. The authors further developed a new drug that inhibits ATP2B1 function and SARS-CoV-2 replication in vitro.

It would be appreciated that the authors explored the role of the membrane Ca²⁺ pump in virus infection; however, this study is too descriptive, molecular mechanisms are essentially still ambiguous, and the presented data do not sufficiently support the authors' conclusions. Therefore, it is unclear how broadly significant the findings are, and this reviewer deems that the manuscript might be more suitable for a specialized journal.

Major concerns:

1- It is essential to demonstrate the involvement of ATP2B1 in viral infection in vivo. Therefore, the authors need to prove the effect of PI-7 treatment on SARS-CoV-2 infection in animal models.

R1-A1) We agree with the reviewer that it would be very interesting to study the role of ATP2B1 in viral infection in vivo in animal models. However, at this time it is outside of the scope of this study.

Here we have focused our analysis in vitro to better dissect molecular mechanisms through the identification of an intronic variant “SNP”, within the ATP2B1 gene locus, with a genetic functional correlation being a predisposition locus to support the SARS-CoV-2 infection and identify this as a marker for those patients at risk of SARS-CoV2 infection and disease progression.

Additional experiments as further suggested by other reviewers have been conducted to further characterize the Ca²⁺ role during SARS-CoV-2 infection. Using a known cell-permeant chelator of Ca²⁺ (BAPTA-AM). BAPTA-AM treatment has been performed to further validate our studies and results are now part in the manuscript (see now “Figure 1 D and “Figure S1B”). Results presented showed that BAPTA-AM treatment of SARS-CoV-2 infected cells generates an impairment of virus replication as measured by Nucleoprotein (N) expression, see immunoblot staining data in both HEK293T-ACE (Figure 1 D) and in human epithelial primary nasal cells (Figure S1B).

This thus confirms our assumption that SARS-CoV-2 need Ca²⁺ to survive and replicate in cell (this was already observed by others scientists (see articles presented by Shang et al., 2020; Serebrovska et al., 2020; Sascha Berlansky et al., 2022). In addition, and most importantly, the use of BAPTA-AM

influence negatively the formation of syncytia of HEK293T-ACE2 cells as further described in supplemental Figure 5C. Below the data as presented on the new Figures.

R1-Q2 It is a superb discovery that the authors identified an SNP in the ATP2B1 gene frequently observed in severe cases of COVID-19. This finding led to identifying FOXO3 as a transcriptional regulator of ATP2B1. However, it is mandatory to perform promoter assays to check whether the regulation of ATP1B1 expression by FOXO3 is affected by the SNP to support the authors' conclusion.

R1-A2) We appreciate the reviewer's valuable of our findings and his suggestions.

Following his request, we have now obtained data through the use of a classic luciferase reporter assay (see figure below), thus showing a decreased reporter activity in the presence of the intronic polymorphism rs111337717 of ATP2B1 gene (i.e., C/C), as here identified. This data is presented now in the manuscript as Figures 3E and further support the hypothesis that this polymorphism is involved in the regulation of ATP2B1 expression and consequently affecting the SARS-CoV-2 infection and replication in human in those carrying this polymorphism.

We also tested in our reporter assay, the ectopic expression of FOXO3, as identified here and predicted to be a motif target of the fork-head transcription factor family. The data presented now, show that FOXO3 induces transcription within the use of the luciferase reporter assay (see Figure 3 F), to a greater extent in the T/T genotype as compared to the C/C genotype (these analyses were performed using the existing sequence in the intronic polymorphism rs111337717 of ATP2B1 gene). This strongly indicates that the intronic polymorphism in patients carrying the C/C genotype would affect the severity outcome of SARS-CoV-2 infected patients. Here we draw a model as presented in Figure 3G.

In conclusion, our data show that rs111337717 polymorphism is responsible for increased viral replication thus causing severe COVID-19 in patients carrying the homozygous allele (C/C), correlating these findings to the altered transcriptional regulation of ATP2B1. We have updated the text in the results section to describe these findings.

Additionally, to functionally validate this genetic polymorphism, we employed the Cas9 system as previously described (Shao Y., et al., 2014) to introduce a C/C point mutation in the human locus corresponding to the rs11337717 in HEK-293T cells (both HEK293T-ACE2 and human epithelial primary nasopharyngeal cells which are known to retain the T/T genotype in the SNPs at the ATP2B1 locus (Supplemental Figure 3C-D). N. 4 clones were further studied following a strategy of selection as depicted in Figure 3G. Although the C/C point mutation is intronic in the ATP2B1 gene, its mRNA level, in these selected edited clones, was found somewhat decreased as compared to WT cells and unedited clones. This further suggest that the T versus C mutation influence the mRNA levels of ATP2B1 in our identified clones (data presented in Supplemental Figure 3I). Further measuring the intracellular Ca²⁺ level, in the steady state conditions, it was further observed in these clones an increase of the amount of Ca²⁺ (see now Supplemental Figure 3H), and its description in figure legend.

In conclusion by RNA-seq data in all the selected C/C clones, we show that PI3K/AKT and Ca²⁺ signaling pathways were mainly affected. This is presented in results section and in Figure 3H-I and legend descriptions.

Taken altogether these new data further strength the role of the SNP variant in the ATP2B1 gene regulating the level of its expression, resulting as both occurring the enhanced Ca²⁺ level in the cell together with an unbalanced activation of the PI3K/AKT pathway. This is in agreement with SARS-CoV-2 infection in human. We refer this in Discussion section referencing the most relevant literature data (Khezri et al., 2022, Soheila Fattahi et al., 2022, Maria Sofia Basile et al., 2021). Below new Figures data as appearing in the revised manuscript.

R1-Q3 The method and data for quantification of viral replication are not convincing. Viral replication should be examined by the qPCR method, which ensures the quantification, rather than immunoblot analysis of SARS-CoV-2 N protein (Figures 1H, 2E, 3E, and 5B).

R1-A3) We thank the reviewer the its critical comments.

As general comment we previously show immunoblot data to confirm further the mRNA level of expression of our infected cells (measuring the virus MOI and the viral replication of the time). All the data are now presented through qPCR as requested and presented in Supplemental Table 2 (as representing data of Figure 2C, 2E, 3K, 4H). There are now new set of qPCR data representing results in Figures 1B, 1I, 5D, 5G.

R1-Q3.1 Additional appropriate controls should be provided in some experiments. **Figure 1H:** mock-infected and untransfected controls are needed because N protein expression in the siControl-transfected cells appears to be substantially low compared to that in virus-infected (untransfected) cells in Figure 1C.

R1-A3.1) We thank the reviewer for the opportunity to comment and clarify this issue.

The experiment using a non-virulent strain (the mock-infected) in the SARS-CoV-2 experiment is not available and literature presented in the SARS-CoV-2 field of experiments and it is known that the "mock infected" means "un-infected cells data". For your reference SARS-CoV-2 literature data is indicating that this approach is the main procedure of description for the infections and report here some examples.

-SARS-CoV-2 Disrupts Splicing, Translation, and Protein Trafficking to Suppress Host Defenses - Cell - <https://doi.org/10.1016/j.cell.2020.10.004>;

-Integrated multi-omics analyses identify anti-viral host factors and pathways controlling SARS-CoV-2 infection Nat Commun 15, 109 (2024). <https://doi.org/10.1038/s41467-023-44175-1>;

-SARS-CoV-2 infection in hamsters and humans results in lasting and unique systemic perturbations after recovery SCIENCE TRANSLATIONAL MEDICINE 7 Jun 2022 Vol 14, Issue 664 DOI: 10.1126/scitranslmed.abq3059 ;

-SARS-CoV-2 airway infection results in the development of somatosensory abnormalities in a hamster model SCIENCE SIGNALING 9 May 2023 Vol 16, Issue 784 DOI: 10.1126/scisignal.ade4984).

In our experiments following these articles we have used “mock control uninfected cells” as guide lines for control definitions. In more details, the mock data is an identical parallel treatment, in which medium of the non-infected cells is replaced with a fresh medium without supplementary FBS, at the same volume as those used in the infected cells experiments.

For the above reasons: we comment in Methods section page 26” Cells were then incubated for 1 h at 37 °C and 5% CO₂, in parallel with infected cells. Afterwards, mock medium was removed, cells were washed once with PBS 1x and fresh medium +2% of inactivated FBS was added to the cell culture, as well as in infected cells”.

A more precise nomenclature of our controls following reviewer suggestion is now referred through whole manuscript changing “non-infected cells” to “mock-infected”.

Does siControl inhibit virus infection? Figure 5B: Additional validation in mock-infected cells is needed.

Here we show the original blot in which we load the mock-infected (see lane 1, and Figure below) where saw the level of ATP2B1 protein is substantially higher that those evaluated in the infected cells. To this end we might claim that our siControl inhibit virus infection. At his time we imagine that is the reduced amount of ATP2B1 that increase the nucleoprotein N and this is described in Figure 2C- E and in results section at page 10. Figure below show the data as supportive data images of our immunoblots.

R1-Q3.2 c. Figures 5D and 5E: The authors need to include a negative control for immunofluorescence, including mock-infected cells, to guarantee the specificity of the antibody against the N protein.

R1-A3.2) We thank for the reviewer's suggestions. We have added negative control for immunofluorescence, including high-resolution images, see Figures 5 as shown below. The additional figures in high resolution are deposited on our CEINGE Institutional cloud. These images are available at this link: <https://cloud.ceinge.unina.it/s/GDjGfSq7gNqESxXc>.

R1-Q4 The methodology and data/interpretation for measuring extracellular Ca²⁺ concentrations are unconvincing.

R1-A4) This is an important and relevant consideration, and we thank the reviewer for this comment and apologize for not explaining better our results in more details. However, we agree with the reviewer that the measurements of the extracellular calcium were non properly performed and now eliminated from this revised version of the manuscript. In the revision manuscript we focused on the analysis of intracellular Ca²⁺ because we believe that are more relevant during SARS-CoV2 infections than measuring the Ca²⁺ level outside the cell. This indeed improves the message and clarify the mechanism here identified and presented.

R1-Q4.1 Fluo 3-AM, an acetoxymethyl derivative of Fluo 3, is an inactive form as it is and becomes active to measure Ca²⁺ only after hydrolysis by intracellular esterase. If one wants to

measure extracellular Ca^{2+} concentration, Fluo 3, but not its AM derivative, must be used. Even if the AM derivative was converted to Fluo 3 by serum constituents in the culture medium, the property of Fluo 3 ($K_d = 400 \text{ nM}$) is not suited for measuring extracellular Ca^{2+} , the concentration of which is mM level. Once a decrease in fluorescence intensity is detected by Fluo 3, the extracellular Ca^{2+} concentration drops by 1/10,000-fold! Such a robust decrease in Ca^{2+} in the medium (which is present in a large quantity) cannot be accounted for by changes in intracellular Ca^{2+} (marginal of the amount). Furthermore, such extreme Ca^{2+} deprivation will result in apparent cytotoxicity (which contradicts the authors' claim).

R1-A4.1) We thank the reviewer for raising this concern and apologize for causing excessive confusion. We have now clarified these comments removing these set of experiments from the revised manuscript.

R1-Q5 Although PI-7 was identified as an ATP2B1 inhibitor, an underlying, precise molecular mechanism of ATP2B1 inhibition remains to be elucidated. The established function of ATP2B1 is to pump Ca^{2+} from the cytosol to the outside. Nevertheless, PI-7 treatment decreased intracellular Ca^{2+} concentration (Figure 4F). Alternatively, this reagent upregulated the expression of ATP2B1 (Figure 5C), which might be more likely to lead to a decrease in intracellular Ca^{2+} concentration. In fact, the treatment of PI-7 also affects the expression and phosphorylation levels of Akt and FOXO3, signaling factors of the pathway that the authors identified as a regulator of ATP2B1 expression during viral infection. In this case, another(s) target of this reagent should exist other than ATP2B1.

*R1-A5) We thank the reviewer for this comment which helped us to clarify the mechanism. As described in the text (page 13), the screening to identify PI-7 was conducted starting from a known interactor of ATP2B1 Caloxin 2A1. Caloxin-*a1* is known to be an inhibitor of ATP2A1. In our experiments we have analyzed both PI-7 and PI-8 raised with a pipeline screening methodology described in the manuscript (see Supplementary Figure S4, A-B-C-D), showing that PI-8 has no effect on the amount of intracellular Ca^{2+} , while PI-7 reduce it when used at 100mM in Primary Human Epithelial Nasal Cells. Results are presented in Figure 4E showing that PI-7 reduce the Ca^{2+} at 100 micromolar concentration.*

Our additional data we produced which explains, the effect of PI-7 on ATP2B1 is showing during a Cycloheximide assays (CHX- which visualizes protein degradation kinetics upon CHX treatment over different timepoints) indicating in HEK-293 that while ATP2B1 protein half-life is within 6h of CHX treatment, the time of ATP2B1 protein degradation increases to 8h (see immunoblot data analysis). These are presented now in Figure Supplemental 4 H-I.

Answering the question on how do PI-7 decrease cytoplasmic Ca^{2+} we think we have sufficient data (as shown in Supplemental Figure 4) that computational analysis identifies PI-7 as a potential binder of the Caloxin 2A1-ATP2B1 interactive structure, whose action is not related to the Ca^{2+} binding, but through the data presented, enhancing – upregulating the level of ATP2B1 protein expression (by inhibiting its degradation) upon infections of SARS-CoV-2 (Supplemental Figure 4 A-B). At this time the requested directed binding experiment is not performed.

In conclusion the data presented suggests that we PI-7 through its potential binding with ATP2B1 impairs its time on protein degradation. Further studies using molecular and structural analyses will

map in the near future the interaction of PI-7 to ATP2B1 protein (see comment in the result section page 14 and Discussion section page 20). The Figure below is presented on Figure 4E, and Supplemental Figure 4H-I

R1-Q6 The relationship among intracellular calcium levels, ATP2B1 expression, and viral infection needs to be further examined in detail and appropriately. The authors speculated that knockdown of ATP2B1 did not alter intracellular Ca²⁺ concentration (Figure 2C) due to complementary upregulation of other Ca²⁺ pump paralogues. If this is true, a decrease in ATP2B1 levels by SARS-CoV-2 infection may also not change intracellular Ca²⁺ concentration or virus infection through the complementary expression of such pumps, which contradicts the promotion of the infection in ATP2B1 knockdown cells (Figure 2E). Hence, additional appropriate experiments to elucidate the mechanism of this discrepancy are required.

R1-A6.1 We thank the reviewer for the opportunity to comment this issue. The knock-down experiments were transiently conducted using siRNA against ATP2B1 and siRNA-unrelated CTR do not show a significant Ca²⁺ change and this can be due to several technical issues. Our hypothesis corresponds to both “efficacy of transfection” and “time of the appropriate Ca²⁺ detection” within the window of the maximum silencing effect during transduction. We must underline that these assays

are performed in the peculiar cellular model challenging for transfection as the primary nasal epithelial cells. For these reasons we present the data in Supplemental Figure 2C and discussed in the Result section page 9

R1-Q6.2 It is necessary to verify whether SARS-CoV-2 infection indeed increases intracellular Ca²⁺ concentration. In addition, intracellular Ca²⁺ concentration after SARS-CoV-2 infection should be measured in ATP2B1-knockdown or -overexpressing cells.

R1-A6.2 The experiments proposed here to answer review specific request were not performed. The experiments proposed are very problematic using continuous SARS-CoV-2 infection. Anyhow currently, there is sufficient data in literature supporting intracellular Ca²⁺ increase especially because of Golgi, ER and mitochondria response to virus infection. Indeed SARS-CoV-2 and other virus at RNA to replicate and survive into the cells needs higher Ca²⁺ levels (see literature, indeed it was firstly described since 2014 by Zhou, Y., Xue, S., Yang, J.J., 2013. Calcium and viruses. Encyclopedia of Metalloproteins. Springer, New York, pp. 415–424. https://doi.org/10.1007/978-1-4614-1533-6_58; then a review by Xingjuan Chen et al., 2020 which describe the most important literature data as published on Cells 2020, 9(1), 94; <https://doi.org/10.3390/cells9010094>; then by Suman Saurav et al., 2021, Molecular Aspects of Medicine 81 (2021) 101004, Dysregulation of host cell calcium signaling during viral infections: Emerging paradigm with high clinical relevance. <https://doi.org/10.1016/j.mam.2021.101004>; and by Poggio et al., Perturbation of the host cell Ca²⁺ homeostasis and ER-mitochondria contact sites by the SARS-CoV-2 structural proteins E and M, Cell Death and Disease (2023) 14:297). We mention these articles in Discussion section page n. 18.

In addition to strength the above observations, we show here the data in lungs from patients with progressive SARS-CoV-2 disease as infected vs controls as verified in the single-nuclei RNA-seq for >116,000 nuclei sequenced from 19 COVID19 autopsy lungs and seven pre-pandemic controls (Melms et al., 2021) (at the SC RNA-seq DB available at https://singlecell.broadinstitute.org/single_cell/study/SCP1052/covid-19-lung-autopsy-samples). We observe a substantial downregulation of ATP2B1 and ATP2B4 (Ca²⁺ cell membrane proteins) while opposite trend with up-regulation was observed by ATP2A2 (a sarcoplasmic/endoplasmic reticulum Ca²⁺ transporting 2 protein) (see now Supplemental Figure S1-F).

Thus we think that Ca²⁺ is essential for virus entry, viral gene replication, virion maturation, and its release, then upon virus infection there is a substantial increases of Ca²⁺ as activating by production in ER, Golgi and Mitochondria resulting in a positively increase of the intracellular Ca²⁺ level, thus we found in our cellular model upon infection the diminishing level of ATP2B1 that in turns will enhance this phenomena being the Ca²⁺ not able anymore to go out side the cell if using this specific membrane pump. See Results section page 8 and discussion section page 18

R1-Q6.3 Reduced intracellular Ca²⁺ concentration by PI-7 was only examined at 100 μM (Figure 4). Thus, it is unclear whether the Ca²⁺ concentration in the cells treated with 1 μM PI-7 was indeed reduced in Figures 5 and 6. The intracellular Ca²⁺ in the cells treated with 1 μM PI-7 should be analyzed to confirm a decreased intracellular Ca²⁺ concentration.

R1-A6.3 We thank the reviewer for rising this point. Like we described above, we started our experiment with 100mM to firstly define weather and how these molecules affect intracellular Ca^{2+} being their binding predicted only in-silico through a screening with a pipeline methodology starting from a known interactor of ATP2B1 Caloxin 2A1 (see pipeline methodology to discover PI-7, (see now Supplemental Figure S4 A-B, and figure legend).

Caloxin 2A1 is known to be an inhibitor of ATP2A1. In our experiments we have analyzed both PI-7 and PI-8, showing that PI-8 has no effect on the amount of intracellular Ca^{2+} , while treating with PI-7 we found a reduced intracellular Ca^{2+} level when used at 100mM, this thus occurs in primary human epithelial nasal cells.

We have here presented additional data related to the treatment with decrease doses of PI-7 HEK293T-ACE2 cells, to define the minimal dose capable of affect intracellular Ca^{2+} . We found that 1 mM in HEK-293T-ACE2 was the lowest amount of PI-7 capable of impairs intracellular Ca^{2+} levels (see Figure 4F) and accordingly, we used this concentration (1 mM of PI-7) in the SARS-CoV-2 viral infections experiments. These adjusted modifications were accordingly revised in all the experiments here presented.

General concerns:

1 Texts in the Figures cannot be distinguished due to low image resolution (Figures 1D, 3C, S2A, S3A, and S6A).

We believe that this issue might be caused in the PDF generating process, in any case we had Higher resolution images can be downloaded at the following link:

<https://cloud.ceinge.unina.it/s/GDjGfSq7gNqESxC>.

2 Please be advised to consult with an expert in statistics. For example, the authors utilized unpaired two-tailed Student's t-tests; however, this test compares the difference between the means of two data sets, but not the difference among multiple conditions (**Figure 1A, 4E, 4F, and 6B**).

In addition, the authors should perform statistical tests and provide the p values in the time course experiments (**Figures 2B and 2C**).

Stats Expert??

Minor comments:

1 Page15, line 22: HEK193T might be a typo.

2 " α " in Figures 4C, 4D, 5D, and 5E should be replaced by " μ "?

3 Figures 5D and 5E: The fluorescence signal derived from the N protein is too faint. In addition, these lack scale bars and the information on the arrows in the figures.

4 Some references are cited inappropriately.

4.1 Page7, line 8: Neither Zhou's nor Chen's paper describes SARS-CoV-2 infection. Note that they were published in 2019 or earlier.

4.2 Page 19, lines 13-14: Although the authors cite the paper by Shang et al. to explain the requirement of Ca^{2+} for the binding of SARS-CoV-2 to ACE2, the article does not provide such data.

Response:

We apologize for errors and misleading description. We thank the Reviewer for the suggestion. With the help of a native English teacher, specialized in scientific English, and a Statistical expert we carefully revised all the suggestions including the language of the manuscript and all the data analyses. In the revised text these errors were fixed all.

We greatly thank Reviewer for the helpful suggestions, that allowed us to significantly improve the quality of figures/table and the overall quality of our manuscript.

Referee #2:

The work describes the role of ATP2B1 and ATP2A1 in SARS-CoV-2 infection. The choice of studying ATP2B1 is mostly due to a demonstrated reciprocal relationship between its expression and virus replication, which has merit. The work is extensive and experiments are executed and reported in good detail. However, interpretation of the data may be premature and it is recommended that many of the assertions of what is being seen be revised.

We thank the reviewer for reviewing our work and recognizing the value in it.

R2-Q1) The work begins by showing that the ATP2B1 calcium channel is down-regulated upon SARS-CoV-2 infection. The change in ATP2B1 expression is 5-fold and significant while the control, b-actin remains constant. This is good but it is important to reflect that positive sense viruses like SARS-CoV-2 express their viral mRNAs through a process of cap snatching. This results in degradation of many cell mRNAs. Indeed, it is indicated that many calcium channels and other proteins are down-regulated and likely reflects this process of mRNA destabilization. B-actin may not be the best control in this situation as the protein has a low turnover. Instead, it would be better to use a house keeping gene with better turnover such as GAPDH.

R2-A1) We thank reviewer for his suggestions and positive comments about our work. Following reviewer suggestions our RNAseq data (here newly presented) as performed in human HEK293T-ACE2 overexpressing cells, show no differences between ACTB (beta-Actin) and GAPDH mRNA transcripts upon SARS-CoV-2 infection (log2 fold change: ACTB: 0.87, $P=1,1E-16$; GAPDH, 0.87, $P= 5,9E-04$; see extended data 1, extra table). We also checked at protein level potential discrepancies and we found that ACTB or GAPDH show the same trend of protein expression in our immunoblot analyses (Data presented now on Figure 1C, clearly indicated that can be normalized by b-actin antibody).

For this reason, we believe that ACTB could be considered a good housekeeping gene in our experiments, thus this data indicates that beta-Actin (ACTB) can be considered as good as GAPDH.

R2-Q2) The relationship of ATP2B1 expression, calcium levels in cells and infection is then evaluated using calcium sensitive dyes. This work appears well done with good controls and obtains large differences in outcomes, supporting the hypothesis.

R2-A2) We thank reviewer for his positive feedback. We performed additional experiments and requested by reviewers and now presented on Figure 2B, 3E, 3F, S2C, S2D, S3H, S4G. See figure legends.

R2-Q3) Next, SNP analysis is used to look at natural differences in ATP2B1 genes in the population. However, the gene sequence is highly conserved with only one change being present in an intron. The relevance of this change is not explored, so while an interesting side note, this information is not useful for understanding the role of the protein in infection by SARS-CoV-2 and should be removed or minimized and put in the discussion.

ATP2B1 gene regulation is then assessed. FOXO3 is indicated to control ATP2B1 expression. This is shown by over-expression of FOXO3, which results in more ATP2B1 and A1 expression and appears a strong outcome.

R2-A3) Following this reviewer suggestion and the similar question raised by Reviewer #1 we have now performed the following new assays.

Following their requests, we have now obtained data through a classic luciferase reporter assay (see figure below), thus showing a decreased reporter activity in the presence of the intronic polymorphism rs111337717 of ATP2B1 gene (i.e., C/C) here identified. This data is presented into the manuscript as Figures 3E and further support the hypothesis that this polymorphism is involved in the regulation of ATP2B1 expression and consequently affecting the SARS-CoV-2 infection and replication in human.

We also tested in our reporter assay, the ectopic expression of FOXO3, as identified here and predicted to be a motif target of the fork-head transcription factor family. The data presented now, show that FOXO3 induces transcription within the use of the luciferase reporter assay (see Figure 3 F), to a greater extent in the T/T genotype as compared to the C/C genotype (from the sequence present in the intronic polymorphism rs111337717 of ATP2B1 gene). This strongly indicates that the intronic polymorphism C/C genotype would affect the outcome of SARS-CoV-2 infected patients. Here we draw a model as presented in Figure 3G.

In conclusion, our data show that rs111337717 polymorphism is responsible for increased viral replication thus causing severe COVID-19 in patients carrying homozygous, correlating these findings to the altered transcriptional regulation of ATP2B1. We have updated the text in the results section page 12 to describe these findings.

Additionally, to functionally validate the genetic polymorphism, we employed the CRISPR/Cas9 system as previously described (Shao Y., et al., 2014) to introduce a C/C point mutation in the human locus corresponding to the rs11337717 in HEK-293T cells (of note both HEK293T-ACE2 and human epithelial primary nasopharyngeal cells retain the T/T genotype in the SNPs at the ATP2B1 locus (Supplemental Figure 3C-D). N. 4 clones were further studied following a strategy of selection as depicted in Figure 3G. Although the C/C point mutation is intronic in the ATP2B1 gene, its mRNA level, in these selected edited clones, was found slightly decreased as compared to WT cells and unedited clones. This further suggest that the T versus C mutation influence the mRNA levels of ATP2B1 expression in our identified clones (data presented in Supplemental Figure 3I). Measuring the intracellular Ca²⁺ level, in the steady state conditions, an increase of the amount of Ca²⁺ was further observed in these clones (see now Supplemental Figure 3H), and its description in figure legend.

In conclusion by RNA-seq data in all the selected C/C clones, we show that PI3K/AKT and Ca²⁺ signaling pathways were mainly affected. This is presented in results section and in Figure 3H-I and legend descriptions.

Taken altogether these new data further strength the role of the SNP variant in the ATP2B1 gene regulating the level of its expression, consequently both the enhanced Ca²⁺ level in the cell together with an unbalanced activation of the PI3K/AKT pathway are occurring. This is in agreement with SARS-CoV-2 infection in human. We refer this in Discussion section referencing the most relevant literature data (Khezri et al., 2022, Soheila Fattahi et al., 2022, Maria Sofia Basile et al., 2021). Below new Figures data as appearing in the revised manuscript.

R2-Q4) Next, two small molecules are made. PI-7 and PI-8. These were built based on a known pharmacophore and docking approaches onto ATP2B1. It is shown that it is not toxic below 200 uM.

- Of note, all the figures have mu (u in uM) incorrectly shown as an infinity symbol - please correct.
- (b)When cells are treated with 100 uM of each, a modest reduction (~20%) in calcium release into the extracellular fluid or seen intracellularly. This difference increases when

- low glucose medium is used (30% change). We are not shown how lower concentrations of each compound affects calcium levels. A control for this experiment should be BAPTA-AM and BAPTA-AM-AM. This is a widely accepted chelator of calcium and will help to better understand the baseline of the assays used.**
- c) A dose curve for PI-7 is important since the next experiments use PI-7 at 100-fold lower dose, at 1 μ M. It is important to tell the reader why this much lower amount is used and to provide a dose response curve for calcium mobilization.**
 - d) Also, all the measurements are indirect using whole cells and a fluorescent dye readout. A patch clamp of the ATP2B1 protein would be better to directly measure the impact of PI-7 on calcium flux.**

R2-A5) We thank reviewer for his comments. All the typos (e.g., mM) have been corrected throughout the manuscript.

- (a) Thank you very much for launching this question, which helped answer the reviewer #1. Thanks to the suggested experiments, we saw that the substantial value of our results was substantially improved.*
- (b) Additional experiments as further suggested by the other reviewer have been conducted to further characterize the Ca^{2+} role during SARS-CoV-2 infection. Using a cell-permeant chelator of Ca^{2+} , BAPTA-AM treatment has been performed to further validate our studies and results are added to the revised manuscript (see Figure 1 D and Figure S1B). Results presented showed that BAPTA-AM treatment of SARS-CoV-2 infected cells generates an impairment of virus replication as measured by Nucleoprotein (N), see immunoblot staining in both HEK293T-ACE (Figure 1 D) and in human epithelial primary nasal cells (Figure S1B). Thus, this confirms our hypothesis that SARS-CoV-2 need Ca^{2+} to survive and replicate in cell (this was already observed by others; see articles presented by Shang et al., 2020; Serebrovska et al., 2020; Sascha Berlansky et al., 2022). In addition, and most importantly, the use of BAPTA AM influence negatively the formation of syncytia of HEK293T-ACE2 cells as described in supplemental Figure 5C. Below the data as presented on the new Figures.*

(c) Following reviewer point we have now performed a dose curve of PI-7. We used PI-7a 100-fold lower dose (i.e., 1 μ M) with no observations of the antiproliferative effects in order to show the antiviral action. This reduced dose is able to impair SARS-CoV-2 replication, by restoring Ca^{2+} levels. We then investigated the effect of decreasing PI-7 concentrations on modulation of Ca^{2+} intracellularly (Figure 4F and S4G). We notice that the lowest PI-7 concentration, corresponding to 1 μ M, substantially decreases intracellular Ca^{2+} in an additional cellular model (i.e., HEK-293T cells). Thus, we consider this concentration as representing a lower limit for obtaining sizable physiological effects. This latest condition is responsible for decreased intracellular Ca^{2+} levels (see now Figure S5D-E on the left) and further required to affect SARS-CoV-2 infection and replication. This is presented now at page 14. For the above reasons, we decide to use 1 μ M in all the followed experiments being the lower dose able to decrease the intracellular Ca^{2+} levels. See Figure 4F and Supplemental Figure 4G and figure legends. See further discussion at page 14 and Figure legend description.

R2-Q6) In the next experiment 1 μM PI-7 is used to show the potential inhibition of ATP2B1 has effects on expression of ATP2B1, FOXO3 and PI3K activity. ATP2A1 expression is also elevated, presumably to compensate for loss of ATP2B1 function.

- The confusing aspect is the authors argue that loss of ATP2B1 function is due to PI-7 inhibition and this somehow results in FOXO3 and PI3K changes. This seems to be the wrong way around as the protein expression would be controlled by the transcription factors and not the other way. A feedback model would provide a better explanation of this outcome but would need to be shown.
- Syncytia formation are also looked at. However, the relevance of this as a measure of virus infectivity is obscure as syncytia are often the product after infection of a cell and protein expression. A drop in spike protein expression, would result in less syncytia. The authors need to better describe the relevance of this work.

R2-A6) We thank reviewer for his positive comments.

- We take here the opportunity to better explain the bases for our conclusions on PI-7 mechanism of action as follow:

“SARS-CoV-2 infection results in Ca^{2+} pumps down-regulation, including ATP2B1, leading to increase of intracellular Ca^{2+} (Figure 1F, H). The treatment with the PI-7 by enhancing the ATP2B1 stability reduce the intracellular Ca^{2+} levels (Figure 4F) which results on the inhibition of SARS-CoV-2 replication (low intracellular Ca^{2+}). In this context, FOXO3 nuclear translocation and transcriptional activity is restored (Brunet et al., 1999; Khezri, M. R., et al., 2022), as shown by the increased expression of its targets ATP2B1, ATP2A1.

See Figure 4H-4I, figure legend and discussion at page 20, and model of action Figure 5E.

Further we saw effect of PI-7 on impairing of SARS-CoV-2 infection on two established cell (non-tumoral lines) HEK293T-ACE2 and primary nasal epithelial cells. Our data show that FOXO3 and PI3K changes are consequence of the reduced viral replication upon PI-7 treatment (Figure 3 K and Figure 4H, respectively). Immunoblotting data demonstrate that PI-7 treatment, in absence of SARS-CoV-2 infection, has no effect on PI3K/AKT signaling pathway in those non-tumorigenic cells (see Supplemental Figure 5C). This further support the notion that PI3K/AKT signaling pathway is modulated only in those SARS-COV-2 infected

cells and in those that are tumorigenic cells. Overall, these data reinforce our model of action as presented in Figure 5E. This was further discussed at page 19

- b) *On answering reviewer point we unfortunately disagree with his comments especially because the analyses of syncytia formation and inhibition have a clinical impact in Covid19 pandemia and this has a substantial relevance of our work. At this time a number of articles provided compelling evidences of SARS-CoV-2-induced syncytia. Here we used this phenomenon for the evaluation of drug efficacy in vitro at the end point of the infection.*

For your reference here are some findings:

-Buchrieser, J., et al., 2021;

-Hoffmann, M. et al., 2020;

-Zhang, Z., et al., 2021;

Thus, further the dependence of Ca²⁺ for SARS-CoV S-mediated virus entry and fusion it has been demonstrated and increasing calcium concentrations enhance the membrane-ordering capacity of SARS-CoV S protein by Lai, A.L., et al., 2017.

Finally, a reduced expression of TMEM16, as a gene involved in syncytia generation [Braga, L., et al., 2021], was recently observed and this was used as control on our experiments.

Thus, in attempt to better describe the relevance of our work we explain our findings in relation to syncytia inhibition by PI-7.

We have also evaluated together with TMEM16, Spike expression in PI-7-treated cells upon SARS-CoV-2 infection (see Figure now S5E-D).

The data overall presented showed that the treatment with PI-7 reduce the expression of Spike gene, TMEM16 and syncytia formation. See discussion section at page 17-20.

R2-Q7) Lastly, PI-7 treatment effects on ATP2B1, A1 and production of 3 cytokines after infection. It is interesting that PI-7 at 1 μM is able to restore cell functionality back to uninfected levels in infected cells. This is an interesting result but because PI-7 is not well characterized for direct effects on ATP2B1, the mechanism, while suggested, remains uncertain.

R2-A7) We thank reviewer for his comment. We rephrased our results with better explanation as follow: “Here, our data suggest that the downregulation of those inflammatory cytokines, whose expression is driven by NF-κB, might be mediated by the activation of FOXO3 upon PI-7 treatment in SARS-CoV-2 infected cells. In this regard, literature data have been previously reported a negative cross-talk between FOXO3 and NF-κB (Thompson T, et al., 2015)”. This sentence is now on page 22 in Discussion section.

In details, the treatment with compound PI-7 by restoring FOXO3 activity, reduced the activation of NF-κB, thus further diminishing the levels of inflammatory cytokines with a known role in the cytokine storm in COVID19 patients (see Figure 5D and Figure S6C and figure legends). Further studies will address these findings in the near future. See page 17 in results section and in discussion page 20.

R2-Q8) Overall, this interesting body of work is diminished by poor characterization of PI-7 and how it interacts with ATP2B1. Patch clamp or other biochemical type assay that can measure direct interaction with ATP2B1 should be attempted and allow the experiments to be more easily interpreted.

R2-A8) We thank reviewer for his suggestion as raised additionally by reviewer # 1. We skip the patch clamp experiments and use biochemical assays.

Similarly asked by Reviewer# 1 in R1-A5) We thank the reviewer for this comment which helped us to clarify the mechanism.

As described in the text (page 13), the screening to identify PI-7 was conducted starting from a known interactor of ATP2B1 Caloxin 2A1. Caloxin-*a1* is known to be an inhibitor of ATP2A1. In our experiments we have analyzed both PI-7 and PI-8 raised with a pipeline screening methodology described in the manuscript (see Supplementary Figure S4, A-B-C-D), showing that PI-8 has no effect on the amount of intracellular Ca^{2+} , while PI-7 reduce it when used at 100mM in Primary Human Epithelial Nasal Cells. Results are presented in Figure 4E showing that PI-7 reduce the Ca^{2+} at 100 micromolar concentration.

Our additional data we produced which explains, the effect of PI-7 on ATP2B1 is showing during a Cycloheximide assays (CHX- which visualizes protein degradation kinetics upon CHX treatment over different timepoints) indicating in HEK-293 that while ATP2B1 protein half-life is within 6h of CHX treatment, the time of ATP2B1 protein degradation increases to 8h (see immunoblot data analysis). These are presented now in Figure Supplemental 4 H-I.

Answering the question on how do PI-7 decrease cytoplasmic Ca^{2+} we think we have sufficient data (as shown in Supplemental Figure 4) that computational analysis identifies PI-7 as a potential binder of the Caloxin 2A1-ATP2B1 interactive structure, whose action is not related to the Ca^{2+} binding, but through the data presented, enhancing – upregulating the level of ATP2B1 protein expression (by inhibiting its degradation) upon infections of SARS-CoV-2 (Supplemental Figure 4 A-B). At this time the requested directed binding experiment is not performed.

In conclusion the data presented suggests that we PI-7 through its potential binding with ATP2B1 impairs its time on protein degradation. Further studies using molecular and structural analyses will map in the near future the interaction of PI-7 to ATP2B1 protein (see comment in the result section page 14 and Discussion section page 20). The Figure below is presented on Figure 4E, and Supplemental Figure 4H-I.

H

I

Referee #3:

The manuscript by Antonellis et al deals with the Ca²⁺-dependency of SARS-CoV-2 infection and replication. ATP2B1 is a known Ca²⁺ regulator of cellular export and homeostasis and the authors found that a new nontoxic caloxin-derivate (PI-7) inhibits ATP2B1 which leads to diminished extra- and intracellular Ca²⁺ levels and the impairment of SARS-CoV-2 replication. Therefore, it is proposed to use PI-7 as prophylactic therapy as it reduces intracellular Ca²⁺ levels. Furthermore, the authors discovered a FOXO3 transcriptional site with a rare intronic variant which is associated with severity of COVID19. During infection, the PI3K/Akt signalling pathway is activated, FOXO3 inactivated and in turn the transcriptional control of Ca²⁺ pumps like SERCA inhibited.

R3-Q1) The manuscript is well written (although I strongly recommend passing it to a native speaker for cross-check). The findings are novel, and the experiments shown were carried out thoroughly. To my opinion, the findings represent important milestones that can advance research on SARS-CoV-2.

R3-A1) We thank reviewer for his positive comments. Following his suggestion, the new version of manuscript have been passed to a native speaker for further cross-check.

I do have some minor comments that can improve the quality of the manuscript:

R3-Q2) How frequent is the intronic variant on the FOXO3 transcriptional site within the population? This should be stated in the manuscript.

R3-A2) We thank reviewer for his comment. Our manuscript identifies a rare homozygous intronic variant in the ATP2B1 locus (rs11337717; chr12:89643729, T>C). The total allele frequency in global population is 0.04 (from gnomAD v3.1.2, https://gnomad.broadinstitute.org/region/12-89588049-89709300?dataset=gnomad_r3).

This information is now in the manuscript in Results Section at page 11.

R3-Q3) Figure 1 shows gene expression analysis of infected cells compared to non-infected ones. These experiments were carried out in HEK293T cells. To my knowledge, HEK cells also endogenously express ACE2 - how can the authors be sure that they did not measure artifacts?

R3-A3) We thank reviewer for his additional comment.

Literature data have shown a low rate of infection by SARS-CoV-2 in HEK293T cells due to low level expression of human angiotensin-converting enzyme 2 (hACE2) that mediate SARS-CoV-2 cellular entry. For this reason, we have used HEK293T hACE2 in order to establish a robust infection with very low MOI of SARS-CoV-2 viral particles (Gordon, D. E. et al. (2020); Gordon, D. E. et al., (2020). Sun, et al. (2021). Gillot C, et al., 2022).

Furthermore, our data also have shown no alteration in ATP2B1 intracellular compartmentalization (see Figure 1A and Figure legends) in HEK293T overexpressing ACE2. These comments are now underlined into the manuscript in the Results section page 7.

R3-Q4) Ca²⁺ homeostasis within cells is also dependent on CRAC channels. Did the authors take the possibility into account that these types of channels also play a role during SARS-CoV-2 infection/replication?

*R3-A4) We thank reviewer for raising this issue. Calcium release-activated channels (CRAC) are plasma membrane Ca²⁺ ion channels formed from the Orai proteins (Orai1–3). Activation of Orai1 occurs in response to binding of the ER Ca²⁺ sensing proteins, *STIM1* and *STIM2* (Scott M. Emrich et al., 2022). Our transcriptomic data (see extended data 1) show the following log₂FoldChange values in of mitochondrial level of expression of ion transporter Ca²⁺ proteins in response to SARS-CoV-2 infection: *ORAI1*:0.43, P=0.002; *ORAI2*: 0.30, P=0.007; *ORAI3*: -0.002, P= 0.980; *STIM1*: 0.088, P=0.302; *STIM2*: -0.039, P=0.768).*

Moreover, the data obtained in HEK293T-ACE2 cells infected with SARS-CoV-2 for 24 hours do not show a substantial down-regulation of CRAC channels. At this time, involvement of CRAC channels in SARS-CoV-2 infection cannot be excluded.

Because the comprehensive data here reported to further dissect the other Ca²⁺ responsive gene/proteins (including those Ca²⁺ release-activated channels) and their deregulations levels upon SARS-CoV-2 infection would be a too broad picture, we will be exploited there results in near future studies.

We are adding this information in the Result section on page 8

R3-Q5) All Western Blots in the figures are just cut-outs. Whole blots have to be shown in appendix.

R3-A5) We thank reviewer for this comment. Extended data containing the whole immunoblots are included in the manuscript.

R3-Q6) The fluorescent images in figure 5 are quite dark - is there a better resolution achievable? In addition, the arrows in figure 5 are too big and cover the spots that should be seen.

R3-A6) We thank reviewer for this suggestion. High resolution immunofluorescence figures are included in the manuscript and can be downloaded at the following link : <https://cloud.ceinge.unina.it/s/GDjGfSq7gNqESxC>

Dear Prof. Zollo,

Thank you for the re-submission of your revised manuscript to our editorial offices. I have now received the reports from the two referees that I asked to re-evaluate your study, you will find below. Original referee #2 has also been invited to re-assess the study, but remained completely unresponsive to our messages. Nevertheless, going through your point-by-point response, I consider her/his points as adequately addressed.

As you will see, the other two referees now support the publication of the study in EMBO reports. However, referee #1 has some remaining concerns and suggestions to improve the manuscript, I ask you to address in a final revised manuscript. Please also provide a final p-b-p-response to the remaining points of referee #1.

We now request the publication of original source data with the aim of making primary data more accessible and transparent to the reader. Our source data coordinator will contact you to discuss which figure panels we would need source data for and will also provide you with helpful tips on how to upload and organize the files.

- Please provide a more comprehensive title with not more than 100 characters including spaces.
- Please shorten the abstract to not more than 175 words.
- Please remove the sections 'In brief' and 'Highlights', and the short title from the manuscript text file.
- We updated our journal's competing interests policy in January 2022 and request authors to consider both actual and perceived competing interests. Please review the policy <https://www.embopress.org/competing-interests> and update your competing interests if necessary. Please name this section 'Disclosure and Competing Interests Statement' and put it after the Acknowledgements section.
- Please reduce the keywords to five and order the manuscript sections like this, only using these names: Title page - Abstract - Keywords - Introduction - Results - Discussion - Methods - Data availability section - Acknowledgements - Disclosure and Competing Interests Statement - References - Figure legends - Expanded View Figure legends
- Please move the 'ethics' section to the methods section and add a paragraph titled 'Biosafety' to the methods section gathering all information on where and how biosafety-relevant experiments with viruses were performed and that these were approved, and by whom (institution, government).
- Please move the funding information to the acknowledgements section. Moreover, please make sure that all the funding information is also entered into the online submission system and that it is complete and similar to the one in the acknowledgement section of the manuscript text file.
- We now use CRediT to specify the contributions of each author in the journal submission system. CRediT replaces the author contribution section. Please use the free text box to provide more detailed descriptions and do NOT provide your final manuscript text file with an author contributions section. See also our guide to authors: <https://www.embopress.org/page/journal/14693178/authorguide#authorshipguidelines>
- Please provide individual production quality figure files as .eps, .tif, .jpg (one file per figure), of main figures and EV figures. Please upload these as separate, individual files upon re-submission.

- We request that primary datasets produced in this study (e.g. RNA-seq, ChIP-seq, structural and array data) are deposited in an appropriate public database.

The accession numbers and database should be listed in a formal "Data Availability" section (placed after Methods) that follows the model below. This is now mandatory (like the COI statement). Please note that the Data Availability Section is restricted to new primary data that are part of this study. This section is mandatory. As indicated above, if no primary datasets have been deposited, please state this in this section

Data availability

- Our journal encourages inclusion of *data citations in the reference list* to directly cite datasets that were re-used and obtained from public databases. Data citations in the article text are distinct from normal bibliographical citations and should directly link to the database records from which the data can be accessed. In the main text, data citations are formatted as follows: "Data ref: Smith et al, 2001" or "Data ref: NCBI Sequence Read Archive PRJNA342805, 2017". In the Reference list, data citations must be labeled with "[DATASET]". A data reference must provide the database name, accession number/identifiers and a resolvable link to the landing page from which the data can be accessed at the end of the reference. Further instructions are available at: <http://www.embopress.org/page/journal/14693178/authorguide#referencesformat>

- Regarding data quantification and statistics, please make sure that the number "n" for how many independent experiments were performed, their nature (biological versus technical replicates), the bars and error bars (e.g. SEM, SD) and the test used to calculate p-values is indicated in the respective figure legends (main and EV figures). Please also check that all the p-values are explained in the legend, and that these fit to those shown in the figure. Please provide statistical testing where applicable. Please avoid the phrase 'independent experiment', but clearly state if these were biological or technical replicates. Please also indicate (e.g. with n.s.) if testing was performed, but the differences are not significant. In case n=2, please show the data as separate datapoints without error bars and statistics. See also:

<http://www.embopress.org/page/journal/14693178/authorguide#statisticalanalysis>

- Please add to each legend (main and EV figures) a 'Data Information' section explaining the statistics used or providing information regarding replicates and scales.

- Please use our reference format:

- Please upload a complete author checklist with your final submission, which you can download from our author guidelines (<https://www.embopress.org/page/journal/14693178/authorguide>). Please insert page numbers in the checklist to indicate where the requested information can be found in the manuscript. The completed author checklist will also be part of the RPF.

Please also follow our guidelines for any use of living organisms, and the respective reporting guidelines:

- Tables S1-S8 are Datasets. Please upload these as dataset files, named Dataset EV1-EV6, and update their callouts accordingly. Please put a title and the legend on the first TAB of the excel files. Please also change the callouts for these in the manuscript text file using the nomenclature Dataset EVx.

- There are further 6 files uploaded as 'data sets', but not called out in the text (it seems).

If these datasets are actually source data, please include them into the source data (see above) and to not upload them as datasets.

- Please add scale bars of similar style and thickness to microscopic images, using clearly visible black or white bars (depending on the background). Please place these in the lower right corner of the images themselves. Please do not write on or near the bars in the image but define the size in the respective figure legend.

- We would encourage you to use 'Structured Methods', our new Materials and Methods format. According to this format, the Materials and Methods section should include a Reagents and Tools Table (listing key reagents, experimental models, software, and relevant equipment and including their sources and relevant identifiers), uploaded as separate file, followed by a Methods and Protocols section in which we encourage the authors to describe their methods using a step-by-step protocol format with bullet points, to facilitate the adoption of the methodologies across labs. More information on how to adhere to this format as well as downloadable templates (.doc or .xls) for the Reagents and Tools Table can be found in our author guidelines (section 'Structured Methods'):

In addition, I would need from you:

Yours sincerely,

Referee #1:

The authors performed additional experiments and revised the manuscript according to the reviewers' comments, which improved the manuscript significantly. Therefore, this study should be published after several remaining issues are addressed.

1 The authors stated that "studies of the role of ATP2B1 in viral infection in vivo using animal models are outside the scope of this study at this time" and did not perform in vivo validation. In that case, the claims of in vivo experiments and drug development should be toned down because the authors did not assess the pharmacokinetics of PI-7 or its antiviral activity in vivo.

1.1 Abstract: "PI-7 can be used prophylactically as a therapeutic agent against COVID-19."

1.2 Highlights: "A new drug and its lack of toxicity "compound PI-7" thus envisioning both preventive and therapeutic applications in patients with COVID-19."

2 The authors claim that they have consulted with statistical experts and carefully reviewed all data analyses; however, issues remain to be fixed.

2.1 A "unpaired two-tailed Student's t-test" is still used for multiple comparisons (Figures 1D, 3F, S1B, S4E, and S5G).

2.2 As this Reviewer had previously pointed out, the authors should perform statistical tests and provide p-values in time-course experiments (Figures 2B, 4E, 4F, S2C, S2D, S3H, S4C, S4D, and S4G).

2.3 For statistical analysis, at least three independent experiments are required. In the following figures, the number of experiments might be two (N=2; Figures 1G, 2A, 2C, 2E, 3K, and 4H).

3 It is suitable to normalize the band intensities of the CoV-2 N protein in the uninfected sample (Mock) to "0," but not "1", as in Figures 1D and 1G (Figures 2C and 2H).

4 A mock-infected sample is needed (Figure 2E).

5 Figures 5C and 5D still lack negative control samples, although the authors claimed, "We have added negative control for immunofluorescence." Hence, mock-infected samples should be included therein.

Referee #3:

The authors have carefully addressed all my comments. To my opinion the quality of the manuscript has significantly improved. Therefore, I recommend publication in EMBO reports.

Naples, 27 March 2024:

EMBOR-2022-56072V2-Q -Decision Letter

Naples, 9 April 2024:

POINT-BY POINT RESPONSE

We would like to thank the reviewers and the Editor for the constructive comments. Here below, we provide a point-by-point response to the concerns raised by Review #1.

Referee #1:

The authors performed additional experiments and revised the manuscript according to the reviewers' comments, which improved the manuscript significantly. Therefore, this study should be published after several remaining issues are addressed.

We thank reviewer for acknowledgement of the improvement of our revised manuscript.

1 The authors stated that "studies of the role of ATP2B1 in viral infection in vivo using animal models are outside the scope of this study at this time" and did not perform in vivo validation. In that case, the claims of in vivo experiments and drug development should be toned down because the authors did not assess the pharmacokinetics of PI-7 or its antiviral activity in vivo.

1.1 Abstract: "PI-7 can be used prophylactically as a therapeutic agent against COVID-19."

1.2 Highlights: "A new drug and its lack of toxicity "compound PI-7" thus envisioning both preventive and therapeutic applications in patients with COVID-19."

A.1-A1.1-A.1.2

We agree with reviewer point. The overstatement was accordingly modified using a "mild wording" both in the abstract and in Highlights section. In abstract "As compound PI-7 shows a lack of toxicity in vitro, its prophylactic use as a therapy against the COVID19 is envisioned here". The Section Highlights as specifically request by the Editor was accordingly removed.

2 The authors claim that they have consulted with statistical experts and carefully reviewed all data analyses; however, issues remain to be fixed.

2.1 A "unpaired two-tailed Student's t-test" is still used for multiple comparisons (Figures 1D, 3F, S1B, S4E, and S5G).

A2.1

We thank reviewer for this note. Through the statistical expert suggestion, the Bonferroni correction has been applied to all the multiple T-Test comparison analyses (qPCR analyses, immunoblotting densitometry), as presented in Figures 1D ($p=0.043$ by unpaired two-samples t-test Bonferroni corrected), S1B now EV1B ($p=0.013278$ by unpaired two-samples t-test Bonferroni corrected), and S5G now EV5F ($p=0.0004$ by unpaired two-samples t-test Bonferroni corrected). Related to the Caspase assay data (S4E now EV4E), we have used the Anova test (ordinary one way multiple

comparison) for each compound-treated group (i.e., PI-7 and PI-8, not statistic [NS] vs vehicle). For Luciferase assay (Figure 3F), we have applied the Anova test (ordinary one way multiple comparison corrected by Sidak's multiple comparisons test) for each FOXO3-transfected group (i.e., Empty Vector T/T vs FOXO3-transfected T/T, $p < 0.0001$; Empty Vector C/C vs FOXO3-transfected C/C, $p = 0.0053$; FOXO3-transfected T/T vs FOXO3-transfected C/C, $p < 0.0001$).

Within this approach all the p-values here in the manuscript have been accordingly corrected.

2.2 As this Reviewer had previously pointed out, the authors should perform statistical tests and provide p-values in time-course experiments (Figures 2B, 4E, 4F, S2C, S2D, S3H, S4C, S4D and S4G).

A.2.2

We thank reviewer for this note. The statistical expert suggests (Dr.s Claudia Angelini and Dr. Simon Zollo both acknowledged in the manuscript) to use within the time-course experiments Anova Single factor (SPSS vs.29.0.2.0 -20) (using KS shortcut function, two sample test) and a “non parametric” KOLMOGOROV-SMIRNOV (or using KS shortcut function, two sample test) and in those not found statistically relevant, we use the Fd Anova which analyse the data using a global approach and pair wise analyses for selected groups “Functional data analysis of variance” (Gorecki T and Smaga L. fdANOVA: an R software package for analysis of variance for univariate and multivariate functional data. Computational Statistics (2019) 34:571–597; <https://doi.org/10.1007/s00180-018-0842-7>).

In general about “time-course” experiments we used a pipeline of statistical analyses “One Way Anova single test” to measure the “one to one” media ratio and variances. Then we used SPSS vs.29.0.2.0 (20) (using KS shortcut function, two sample test) with KOLMOGOROV-SMIRNOV which is a “non parametric” test to define whether two different empiric distributions (ECDF) are statistically relevant. The p values define whether differences between the two distributions are sufficiently high between the two ECDF series, thus indicating that are from the same distributions. The “null” hypothesis confirm that the two series are derived by the same distributions. If p values $P < 0.05$, the hypothesis is rejected and we conclude that the two series are statistically different. If p value $P > 0.05$, the “null” hypothesis is accepted, indicating that the two series are having similar distributions. In this regards we compare the median values of the individual set of experiments for each group of results (see colors data and curves as used for pair-wise analyses in addendum Statistical data). Within these two statistical tests above described we analyze the Figures: 4E, 4F, EV 2C, EV 2D, EV 3H.

In addition, using data presented in Figure EV 4G and Figure 2B, we use the KOLMOGOROV-SMIRNOV TEST and the Fd Anova “Functional data analysis of variance” implemented in R packages.

In details:

In Figure 2B we compare Empty Vector Ca^{++} (grey column) vs ATP2B1 Ca^{++} at 10mM concentration (green column) showing $p = 0.001$ by KOLMOGOROV-SMIRNOV test; $p = 0.0365$ by Fd Anova-global CH test; $p = 0,0012$ by F-type boot strap test; pair-wise = 0,0074 FN test; $p = 0,0034$ FB test.

In Figure EV 4G, we compare WT vehicle treated (blue column) and WT- treated with PI#7 (red column), showing $p = 0.1918$ by KOLMOGOROV-SMIRNOV test; $p = 0.0071$ by Fd Anova-global CH test; $p = 0,021$ by FP test - permutation test ; and $p = 0,0015$ by F-type boot strap test.

In Figure EV 4C and Figure EV 4D we use to calculate the half maximal inhibitory concentration (IC50 value) the RTCA software vs.1.2.1 (XCELLIGENCE ACEA System application) with a statistical formula related to the Sigmoidal dose-response (Variable slope), PI-7 IC50 500 μ M R² 0,9; PI-8 IC50 336 μ M R² 0.9.

We add these results now in Figure legends (2B, 4E, 4F, EV2C, EV 2D, EV 3H, EV 4C, EV 4D, and EV 4G), in the “addendum statistical data”, and in the Statistical section of the manuscript.

2.3 For statistical analysis, at least three independent experiments are required. In the following figures, the number of experiments might be two (N=2; Figures 1G, 2A, 2C, 2E, 3K, and 4H).

A.2.3

We thank reviewer to raise this comment. All the experiments are now provided with three independent assays as requested (N=3; Figures 1G, 2A, 2C, 2E, 3K, and 4H).

3. It is suitable to normalize the band intensities of the CoV-2 N protein in the uninfected sample (Mock) to "0," but not "1", as in Figures 1D and 1G (Figures 2C and 2H).

A.3. As previously reported the data related to COV-2 N protein in Mock-infected cells (which results obtained using the same media but with “NO” virus infection) is now presented similarly in all Figures (1D; 1G; 2C and 2H).

4 A mock-infected sample is needed (Figure 2E).

A.4. To please the reviewer specific request we have add an additional control Mock now presented in EV 2F.

5 Figures 5C and 5D still lack negative control samples, although the authors claimed, "We have added negative control for immunofluorescence." Hence, mock-infected samples should be included therein.

A.5 All the controls as requested by reviewer are now in the Expanded view 5C Figure

Referee #3:

3. 1 The authors have carefully addressed all my comments. To my opinion the quality of the manuscript has significantly improved. Therefore, I recommend publication in EMBO reports.

A.3.1 We would like to thank reviewer #3 for the nice comments related to our manuscript findings.

Dear Prof. Zollo,

Thank you for the submission of your further revised manuscript to our editorial offices. I now went through this and your p-b-p-response and consider the remaining points of referee #1 as adequately addressed. Before proceeding with formal acceptance, I have these editorial requests I ask you to address in a final revised manuscript:

- There are author name discrepancies. Fathemeh Asadzadeh, Dong-Rac Choi and Hong-Yeoul Kim in the manuscript text file vs. Fatemeh Asadzadeh, Choi Dong-Rac and Hong-Yeoul Kim in the submission system. Please check.
- Please make sure that the number "n" for how many independent experiments were performed, their nature (biological versus technical replicates), the bars and error bars (e.g. SEM, SD) and the test used to calculate p-values is indicated in the respective figure legends (also for potential EV figures and all those in the final Appendix). Please also check that all the p-values are explained in the legend, and that these fit to those shown in the figure. Please provide statistical testing where applicable. Please avoid the phrase 'independent experiment', but clearly state if these were biological or technical replicates. Please also indicate (e.g. with n.s.) if testing was performed, but the differences are not significant. In case n=2, please show the data as separate datapoints without error bars and statistics. See also:
<http://www.embopress.org/page/journal/14693178/authorguide#statisticalanalysis>

If $n < 5$, please show single datapoints for diagrams. Moreover:

- Please define the annotated p values *** in the legend of figure 5e as appropriate.
- Please indicate the statistical test used for data analysis in the legends of figures 3a, i; EV 1e.
- Please note that in figures 1b, g; 2g; 3e, j-k, 4h-i; EV 5f-g; there is a mismatch between the annotated p values in the figure legend and the annotated p values in the figure file that should be corrected.
- Please note that the box plots need to be defined in terms of minima, maxima, centre, bounds of box and whiskers, and percentile in the legends of figures 5c, e; EV 3f; EV 5b.
- Please note that information related to n is missing in the legends of figures 3j; EV 3i.
- Please note that the white arrows are not defined in the legend of figure 5e. This needs to be rectified.
- The figure legends are presently very long. Please try to shorten these and do not provide methods information in the legends and describe results.
- Please add scale bars of similar style and thickness to all microscopic images, using clearly visible black or white bars (depending on the background). Please place these in the lower right corner of the images themselves. Please do not write on or near the bars in the image but define the size in the respective figure legend. Presently, the scale bars hard to see and seem to have text nearby. Please improve this.
- Please use the name 'Data Availability Section' and remove the subheadings 'Materials availability' and 'Data and code availability'. Please only provide access information to deposited datasets here (and also remove the sentence 'Any additional information required to reanalyze the data reported in this paper is available from the lead contact upon request.'). Please provide specific URLs for the E-MTAB-11973, E-MTAB-13916 and PXD051059 datasets and make sure that data a public latest upon online publication of the manuscript.
- Please remove the section 'Lead contact' from the manuscript. It is sufficient to indicate the corresponding author on the title page.
- Please provide one ethics statement as part of the methods section. Presently, there are two, partly overlapping statements ('ETHICAL COMMITTEE APPROVAL' and 'Ethics').
- Please provide an Appendix file as pdf, containing all the 8 Appendix tables. The Appendix should have page numbers and needs to include a table of content on the first page (with page numbers) and legends for all content. Please follow the nomenclature Appendix Table Sx etc. throughout the text, and also label the tables according to this nomenclature.
- I would also suggest to provide the primer sequence information as a table in the Appendix. Then, please remove these from the Methods section and add appropriate callouts.
- Please make sure that all figure panels are called out separately and sequentially. Presently, there seem to be no callouts for Figure 2G and Appendix Table S8. Please check.
- Please change the titles of the EV figure in their legends to "Figure EVx" (not "Expanded View x").
- Please use our reference format:
<http://www.embopress.org/page/journal/14693178/authorguide#referencesformat>

- Please make sure that all the funding information is also entered into the online submission system and that it is complete and similar to the one in the acknowledgement section of the manuscript text file. Presently, grants from the European School of Molecular Medicine for a doctorate program Fellowship (F.A.) and the Molecular Medicine Doctorate Program Fellowship are missing from the submission system.

- Thanks for providing the source data. Please upload the source data for the main figures as one folder per figure. The SD for the EV figures can stay grouped together in one folder.

In addition, I would need from you:

- a short, two-sentence summary of the manuscript (not more than 35 words).
- two to four short (!) bullet points highlighting the key findings of your study (two lines each).

All editorial and formatting issues were resolved by the authors.

Prof. Massimo Zollo
University of Naples Federico II
Molecular Medicine and Medical Biotechnology DMMBM
Via Pansini 5
Naples, Italy 80131
Italy

Dear Prof. Zollo,

I am very pleased to accept your manuscript for publication in the next available issue of EMBO reports. Thank you for your contribution to our journal.

Yours sincerely,
